# A systematic search for RNA structural switches across the human transcriptome

Matvei Khoroshkin[1,2,3,4], Daniel Asarnow[1,12], Shaopu Zhou[1,2,3,4], Albertas Navickas [1,2,3,4,13], Aidan Winters[2,3,4,5,6,7], Jackson Goudreau[1,2,3,4], Simon K. Zhou [8,9], Johnny Yu[1,2,3,4], Christina Palka[10], Lisa Fish[1,2,3,4], Ashir Borah[1,2,3,4], Kian Yousefi[1,2,3,4], Christopher Carpenter[1,2,3,4], K. Mark Ansel [8,9], Yifan Cheng [1,11], Luke A. Gilbert [2,3,6,7] & Hani Goodarzi [1,2,3,4,7] ✉

RNA structural switches are key regulators of gene expression in bacteria, but their characterization in Metazoa remains limited. Here, we present SwitchSeeker, a comprehensive computational and experimental approach for systematic identification of functional RNA structural switches. We applied SwitchSeeker to the human transcriptome and identified 245 putative RNA switches. To validate our approach, we characterized a previously unknown RNA switch in the 3′ untranslated region of the RORC (RAR-related orphan receptor C) transcript. In vivo dimethyl sulfate (DMS) mutational profiling with sequencing (DMS-MaPseq), coupled with cryogenic electron microscopy, confirmed its existence as two alternative structural conformations. Furthermore, we used genome-scale CRISPR screens to identify *trans* factors that regulate gene expression through this RNA structural switch. We found that nonsense-mediated messenger RNA decay acts on this element in a conformation-specific manner. SwitchSeeker provides an unbiased, experimentally driven method for discovering RNA structural switches that shape the eukaryotic gene expression landscape.

Gene expression is regulated at the RNA level in all kingdoms of life. Some of the oldest groups of RNA-based regulatory mechanisms are ribozymes (catalytically active RNA molecules) and RNA structural switches (elements that adopt two mutually exclusive conformations, each leading to different gene-regulatory outcomes)[1–3]. In bacteria, a subset of RNA switches, termed riboswitches, control gene expression by binding small molecule ligands that induce RNA conformational changes[4,5]. The discovery of RNA switches in eukaryotes, however, has been more challenging. While a number of thiamine pyrophosphate-sensing riboswitches have been identified in plants and fungi[6], only two human RNA switches are known: the protein-dependent RNA switch in vascular endothelial growth factor-A (VEGFA), and m6A modification-based switches[7,8]. Therefore, the overall impact of RNA switches on gene expression in higher eukaryotes remains unclear, despite their ubiquity in other domains of life. Here, we introduce SwitchSeeker, a systematic computational

[1]Department of Biochemistry and Biophysics, University of California, San Francisco, San Francisco, CA, USA. [2]Department of Urology, University of California, San Francisco, San Francisco, CA, USA. [3]Helen Diller Family Comprehensive Cancer Center, University of California, San Francisco, San Francisco, CA, USA. [4]Bakar Computational Health Sciences Institute, University of California, San Francisco, San Francisco, CA, USA. [5]Department of Biological and Medical Informatics, University of California, San Francisco, San Francisco, CA, USA. [6]Department of Cellular and Molecular Pharmacology, University of California, San Francisco, San Francisco, CA, USA. [7]Arc Institute, Palo Alto, CA, USA. [8]Sandler Asthma Basic Research Center, University of California, San Francisco, San Francisco, CA, USA. [9]Department of Microbiology and Immunology, University of California, San Francisco, San Francisco, CA, USA. [10]Gladstone Institute of Data Science and Biotechnology, San Francisco, CA, USA. [11]Howard Hughes Medical Institute, University of California San Francisco, San Francisco, CA, USA. [12]Present address: Department of Biochemistry, University of Washington, Seattle, WA, USA. [13]Present address: Institut Curie, UMR3348 CNRS, U1278 Inserm, Orsay, France. ✉e-mail: hani.goodarzi@ucsf.edu

and experimental framework for unbiased discovery of RNA structural switches in any transcriptome.

While several RNA switch detection software packages have been developed, most identify new switch sequences based on their homology to one of the 40 known RNA switch families[9]. The small minority of tools enabling de novo prediction of RNA switches lack experimental verification of RNA structure and function[10,11]. Therefore, there is an unmet need for scalable methods of detecting eukaryotic RNA switches and assessing the extent to which they carry out regulatory functions in gene expression control. The approach we introduce here relies on integrating multiple computational and experimental methods: RNA switches are first predicted in silico, then structurally and functionally characterized in vivo, which in turn informs the next iteration of in silico predictions. First, we developed a computational model called SwitchFinder for de novo RNA switch detection, and showed that it identifies RNA switches from novel families with higher accuracy than existing models. Combining SwitchFinder with a set of high-throughput experimental techniques, we set up an end-to-end iterative predict-and-validate platform that we term SwitchSeeker. We applied SwitchSeeker to the human transcriptome to identify putative RNA switches, which we then characterized structurally and functionally using massively parallel assays in vivo. By iteratively improving the SwitchFinder predictions with experimental data, we ultimately report 245 high-confidence and functional RNA structural switches.

Finally, we selected the top scoring switch, located in the 3′ untranslated region (3′UTR) of the *RORC* (RAR-related orphan receptor C) transcript, for further analysis. We used dimethyl sulfate (DMS) mutational profiling with sequencing (DMS-MaPseq) structural probing and single-particle cryogenic electron microscopy (cryo-EM) to confirm that the predicted switch populates alternate molecular conformations. We then performed genome-scale CRISPR-interference (CRISPRi) screens, which showed that one of the two conformations reduces RORC gene expression through activation of the noncanonical nonsense-mediated decay (NMD) pathway. Taken together, our framework provides new insights into the role of RNA structural switches in shaping the human transcriptome, and outlines a broader approach for future comprehensive characterization of RNA switches regulating eukaryotic gene expression across cell types and organisms.

## Results

### Systematic annotation of human RNA structural switches

We define RNA structural switches as regulatory RNA elements that affect the expression of the host RNA molecule through conformational shifts. To discover new eukaryotic RNA switch families, we devised an approach called SwitchFinder that, unlike most existing methods[12–17], does not depend on known sequence motifs. Instead, SwitchFinder uses the RNA sequence to generate an ensemble of secondary structures and their corresponding energy landscape using a Boltzmann equilibrium probability distribution[18]. It prioritizes the sequences that show RNA switch-like features, such as having two local minima in close proximity with a relatively small barrier in between (Fig. 1a and Extended Data Fig. 1a,b). This approach ensures that RNA switches are identified in a generalizable and family-agnostic way, which we validated by demonstrating its high performance on held-out Rfam families (Fig. 1b and Extended Data Fig. 1c). We compared the performance of SwitchFinder to SwiSpot, the state-of-the-art method for family-agnostic riboswitch prediction[10], and observed a performance improvement of 44% on average across all RNA switch families except cyclic di-GMP-II (Fig. 1c). By relying on biophysical features of the folding energy landscape, SwitchFinder captures a wider variety of RNA switches compared with the existing methods.

To confirm that SwitchFinder is not overly tailored to bacterial riboswitches, we tested it on eukaryotic and synthetic riboswitches, including those sensing theophylline[19] and specific RNA-binding proteins[20]. Additionally, we applied SwitchFinder to ribosomal RNAs

to ensure its ability to distinguish RNA switches from nonswitching but highly structured RNAs. This analysis showed that SwitchFinder could distinguish true riboswitches from shuffled controls much more effectively than it could do so with ribosomal RNAs, and that it performed even better on eukaryotic and synthetic riboswitches than it did on bacterial riboswitches (Fig. 1d). Altogether, these benchmarking results gave us high confidence that SwitchFinder could nominate new eukaryotic RNA switches that would expand our understanding of RNA structural switching in gene regulation.

### Discovery of RNA switches with regulatory function in the human transcriptome

Messenger RNA secondary structure in the cell is highly dynamic[21–23] and compartment dependent[24]; therefore, we reasoned that the SwitchFinder predictions may be greatly improved with experimental measurements of RNA secondary structure from living cells. To counteract the limitations of in silico RNA folding predictions in complex eukaryotic transcriptomes[25], we enhanced SwitchFinder by allowing the incorporation of in vivo RNA secondary structure probing data to refine the model's energy terms, resulting in an iterative cycle of computational prediction and experimental validation that we name SwitchSeeker. First, we applied the SwitchFinder model using naive in silico folding to the entirety of the 3′UTRs of the human transcriptome, and chose the 3,750 top candidate switches (of length ≤186 nucleotides) as putative switch elements. To identify the RNA switches that are both functional and structurally bi-stable in the cell, we independently performed two high-throughput in vivo screens: a 'structure screen' that differentiates RNAs that exist as an ensemble of two mutually exclusive conformations from those that exist only in a single conformation, and a 'functional screen' that measures the effect of candidate RNA switches on the expression of a reporter gene.

For the structure screen, we performed an in vivo DMS-MaPseq assay on HEK293 cells expressing a library of the 3,750 candidate RNA switches in a reporter gene context to identify bi-stable RNA structures in the initial pool of 3,750 candidate switches (Extended Data Fig. 2b,c)[26,27]. The accessibility of a single nucleotide in the DMS-MaPseq data is measured as a population average of multiple RNA molecules that represent different minima in the Gibbs free energy landscape. If one conformation dominates the landscape, it dominates the DMS-MaPseq reactivity profile; however, if multiple conformations coexist, they all contribute to the reactivity profile[28,29]. SwitchSeeker exploits this distinction in nucleotide accessibility to find RNA switches that coexist in a balanced state between two conformations in vivo.

For the functional screen, we implemented a massively parallel reporter assay (MPRA)[30] to functionally interrogate RNA switches in HEK293 cells. We cloned the library of 3,750 candidate RNA switch sequences or cognate scrambled control sequences into a dual enhanced green fluorescent protein (eGFP)–mCherry fluorescent reporter, directly downstream of the eGFP open reading frame (ORF; Extended Data Fig. 2d). This enabled us to use eGFP fluorescence to measure the effect of candidate RNA switches on gene expression while using the unaffected mCherry fluorescence as an endogenous control. We transduced HEK293 cells with this synthetic library and sequenced DNA and RNA derived from eight bins of cells sorted by flow cytometry according to their eGFP : mCherry expression ratio (Extended Data Fig. 2e, see Methods). Of the candidate RNA switches tested, 536 (14%) caused significant downregulation of eGFP relative to their scrambled control, and 538 (14%) showed a significant upregulation (Fig. 2b). While our study focused on characterizing the RNA switches that act in the context of 3′UTRs, the SwitchSeeker framework can be readily applied to the study of other types of RNA switches with the use of appropriate reporter constructs.

In the second iteration of SwitchSeeker, guided by in vivo RNA structure data, we refined our predictions, eliminating false positives and focusing on switches with consistent structural configurations

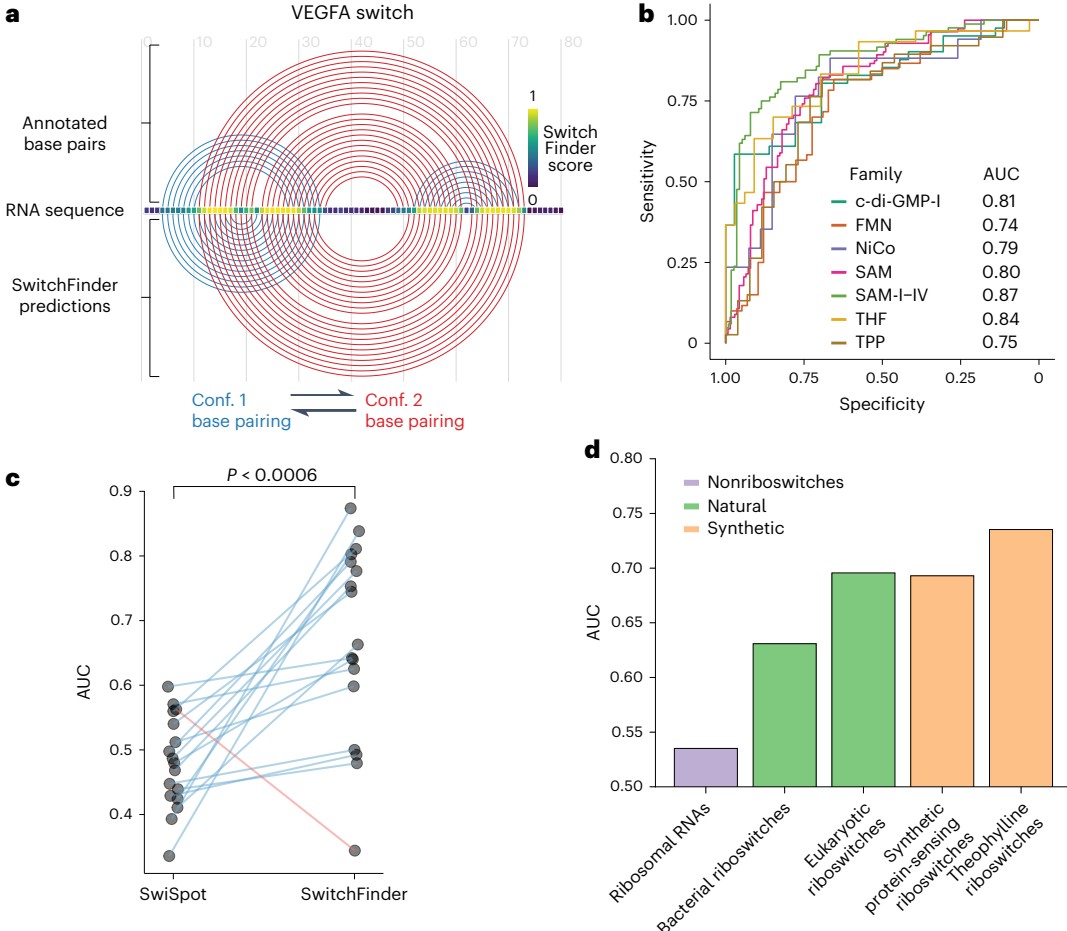

**Fig. 1 | SwitchFinder identifies candidate RNA switches in the human genome. a**, Example of SwitchFinder locating the RNA switch in the VEGFA mRNA sequence. **b**, Receiver operating characteristic (ROC) curves of SwitchFinder predictions of RNA switches from the common Rfam families. SwitchFinder was applied to a mix of real sequences and their shuffled counterparts (with preserved dinucleotide content). ROC curves measure its ability to correctly select the real sequences. AUC, area under the ROC curve; riboswitch families, c-di-GMP-I (Cyclic di-GMP); FMN, flavin mononucleotide; NiCo, nickel or cobalt ions; SAM, S-adenosyl-l-methionine; THF, tetrahydrofolate; TPP, thiamine pyrophosphate. **c**, AUCs of RNA switch predictions across the Rfam families for two models: SwitchFinder and SwiSpot[10]. Each dot represents one Rfam family. The lines show the change in accuracy between the two models. The families that have higher AUCs for SwitchFinder are shown with blue lines; the ones that have higher AUCs for SwiSpot are shown in red. *P* value calculated with the paired two-sided *t*-test ($P = 0.00056$). **d**, AUCs of RNA switch predictions across various groups of natural and synthetic riboswitches, calculated as in **b**.

in vivo. Comparing outcomes of this iteration with the first iteration, we found a significant increase in the proportion of regulatory active switches ($P = 1 \times 10^{-6}$, Extended Data Fig. 2f), validating the enhanced accuracy through in vivo data integration. This process prioritized 1,454 putative RNA switches that occupy two alternative conformational minima and are regulatory active in vivo.

Having identified a large set of candidate RNA switches that affect gene expression, we aimed to assess the degree to which the two stable conformations show divergent regulatory function. For this, we extended our MPRA to include targeted mutations designed to shift the equilibrium between the two conformations of each candidate RNA switch. This was achieved by either disrupting or strengthening conformation-specific stem loops by introducing either individual mutations or reciprocal mutation pairs (Fig. 2c). This additional screen enabled us to identify bona fide RNA switches with strong conformation-dependent activity. We found 245 RNA switches that differentially regulated reporter gene expression when locked in a specific structural conformation. An example candidate switch (located in the 3′UTR of *TCF7* (transcription factor 7)) is shown in Fig. 2d: the TCF7 RNA switch landscape has two local minima, corresponding to two alternative conformations supported by in vivo DMS-MaPseq data (Fig. 2d, bottom). Two mutations in different parts of the switch sequence that

favor conformation 1 resulted in lower expression of the eGFP reporter (top). Conversely, two mutations that favor conformation 2 increased eGFP expression. This observation indicates that the two conformations of the TCF7 RNA switch elicit divergent regulatory functions.

## A bi-stable RNA switch in the 3′UTR of RORC

To demonstrate the validity of SwitchSeeker's predictions, we aimed to biochemically characterize one of the identified RNA switches. We selected the switch that had the most pronounced difference in regulatory functions between its two conformations: a 186 nucleotide element located in the 3′UTR of the *RORC* mRNA. Based on the predicted secondary structures, we designated the three regions involved in the base pairing as 'Box 1' (61–69 nucleotides), 'Box 2' (73–81 nucleotides) and 'Box 3' (116–123 nucleotides). Our data indicate that Box 1 can form base pairs either with Box 2 or with Box 3, resulting in two mutually exclusive conformations that each exert distinct effects on gene expression (Fig. 3a). To confirm that the RORC RNA switch exists as an ensemble of two stable conformations, we designed mutation–rescue pairs of sequences that first shift the equilibrium towards one conformation (mutation), and then shift it towards the other conformation (rescue) (Fig. 3b and Supplementary Data Files), and used in vitro RNA SHAPE (selective 2′-hydroxyl acylation analyzed by primer extension)[31] to

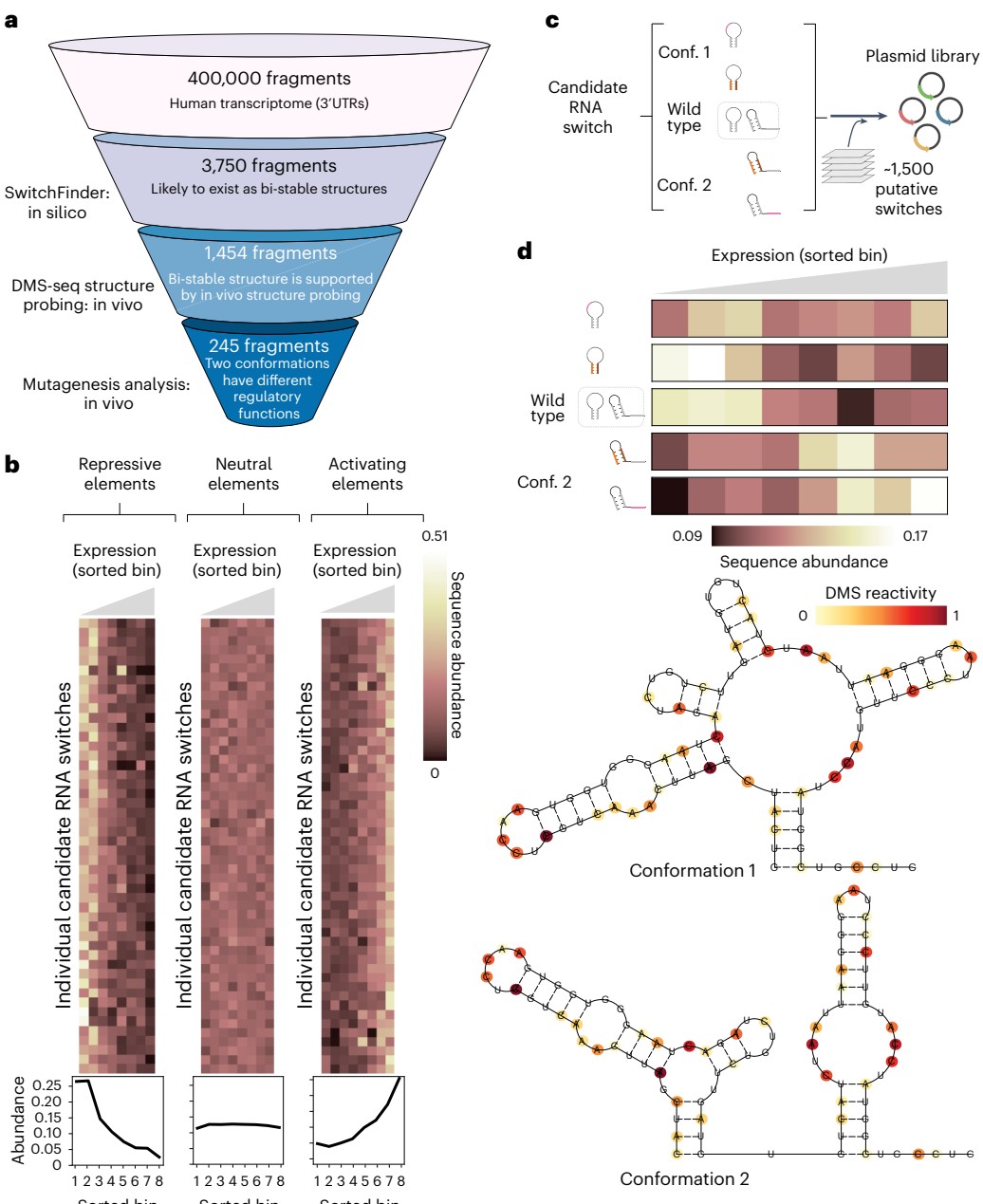

**Fig. 2 | MPRA captures the functional difference between the conformations of candidate RNA switches. a**, Overview of SwitchSeeker, the platform for RNA switch identification, applied to the 3′UTRs of the human transcriptome. **b**, Examples of regulatory elements identified by the functional screen. Each row represents a single candidate RNA switch, each column represents a single bin defined by the reporter gene expression (eGFP fluorescence, normalized by mCherry fluorescence). Bin 1 corresponds to the cells with the lowest eGFP fluorescence, bin 8 corresponds to the highest. The value in each cell is the relative abundance of the given RNA switch in the given bin, normalized across the eight bins. The three plots show examples of candidate switches with repressive, neutral and activating effects on gene expression. The plots below show cumulative sequence abundances across all of the candidate switches in each group. **c**, The set-up of the massively parallel mutagenesis analysis. For each candidate RNA switch, we design four mutated sequence variants. Two of them

lock the switch into conformation 1, and the other two lock it into conformation 2. A sequence library is then generated (Extended Data Fig. 2d), in which each candidate RNA switch is represented by the four mutated sequence variants, along with the reference sequence. **d**, Example of a high-confidence candidate RNA switch identified using the massively parallel mutagenesis analysis. Bottom: Two alternative conformations as predicted by SwitchSeeker. The RNA secondary structure probing data collected with the Structure Screen is shown in color. The Gibbs free energy difference between the two predicted conformations is 2.4 kcal per mol. Top: The effect of the candidate RNA switch locked in one or another conformation on reporter gene expression. Each row corresponds to a single sequence variation that locks the RNA switch into one of the two conformations. Each column represents a single bin defined by the reporter gene expression. The value in each cell is the relative abundance of the given RNA switch in the given bin, normalized across the eight bins.

monitor the resultant RNA structures. We found that mutating Box 3 (117-AC) reduced the reactivity of the Box 2 region (Fig. 3c), supporting the idea that Box 1 would switch its contacts from Box 3 to Box 2, thereby stabilizing conformation 2. Introducing the rescue mutation

(65-GT,117-AC) into Box 1 restored the original reactivity profile of the element. Complementary experiments using the mutation (77-GA) to stabilize conformation 1, and the rescue mutation (63-TC,77-GA) to stabilize conformation 2, had a similar outcome. Even though we did

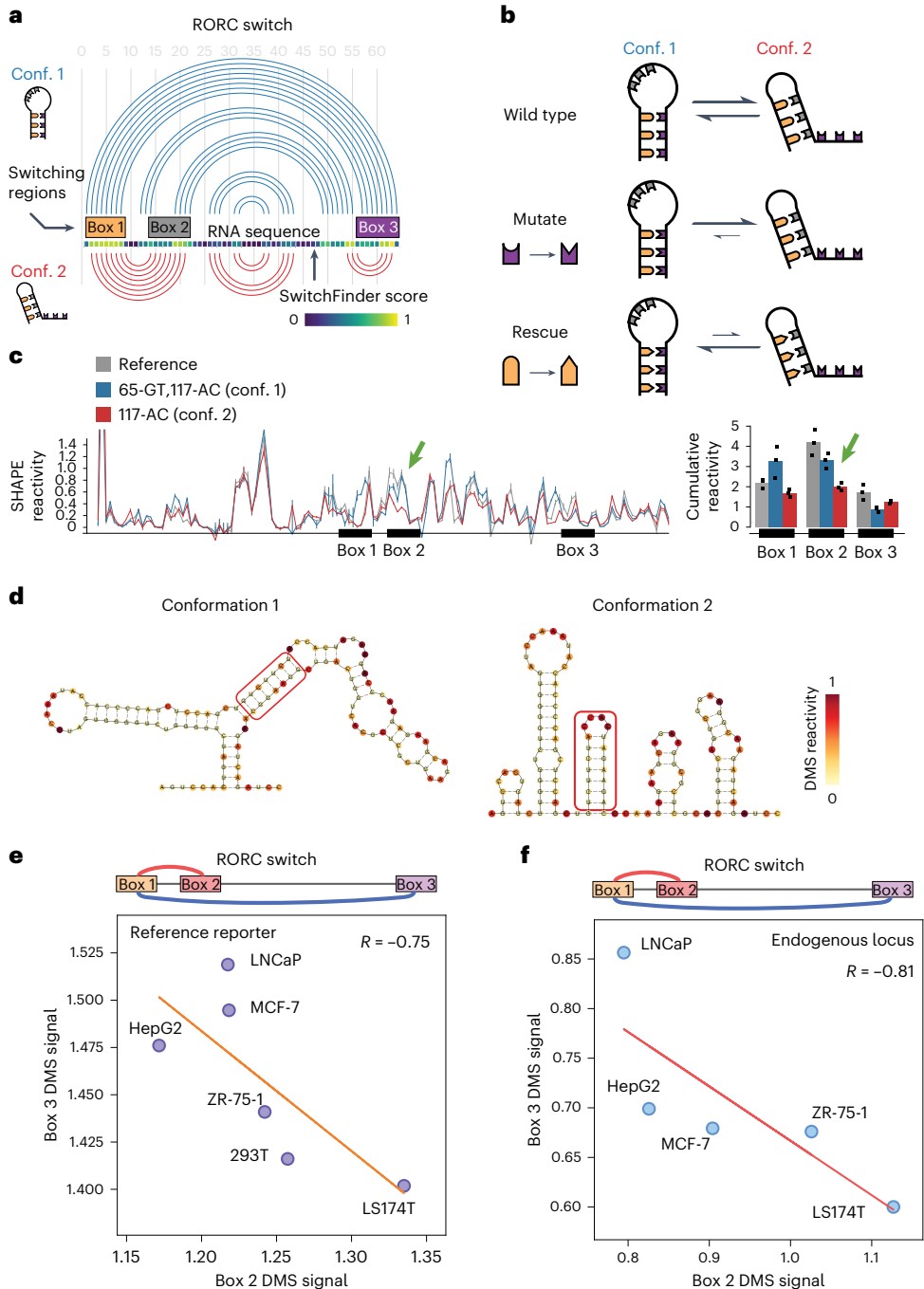

**Fig. 3 | A fragment of RORC 3′UTR forms an ensemble of two alternative structures. a**, Arc representation of the two alternative conformations of the RORC RNA switch as predicted by SwitchSeeker. The two conformations are shown in blue and red, respectively. Left: The schematic representations of the two conformations, as used throughout the article. **b**, The set-up of mutation–rescue experiments. The switching regions are color coded as in **a**. A-U and C-G base pairing is shown with compatible shapes (triangle and half-circle). The two conformations of the switch reside in the equilibrium state. Mutation of the Box 3 region disrupts the base pairing between the Box 1 and the Box 3 regions. This causes a shift of the equilibrium towards conformation 2. Rescue mutation of the Box 1 switching region restores the base pairing between Box 1 and Box 3, but at the same time it disrupts the base pairing between Box 1 and Box 2. Therefore, the equilibrium shifts towards conformation 1. **c**, In vitro SHAPE reactivity of the RORC RNA switch sequence in vitro. Left: SHAPE reactivity profiles for the reference sequence (in gray) and for the mutation–rescue pair of sequences (blue, 65-GT,117-AC; red, 117-AC). Shown is the average for three replicates with

the respective error bars (s.d.). The SHAPE reactivity changes in the nonmutated regions are highlighted with bold arrows. Right: Barplots of cumulative SHAPE reactivity in the switching regions. **d**, Secondary structures of the two conformations of RORC RNA switch predicted by the RNAstructure algorithm[56] guided by the DMS reactivity data. The base pairing of Box 1 with either Box 3 (conformation 1) or Box 2 (conformation 2) is highlighted by a red frame. The two clusters were identified using the DRACO unsupervised deconvolution algorithm[28]. **e**, Accessibility of the Box 2 (x axis) and Box 3 (y axis) regions of the RORC element across cell lines, as measured with DMS-MaPseq (normalized reactivity, see Methods). The cell lines were engineered to express a GFP reporter containing the RORC switch sequence in the 3′UTR, and the accessibility of the reporter mRNA was measured with DMS-MaPseq. Linear regression is shown with an orange line. **f**, Accessibility of the Box 2 (x axis) and Box 3 (y axis) regions of the RORC element in the endogenous RORC mRNA, as measured with DMS-MaPseq (normalized reactivity, see Methods).

not observe a substantial decrease in reactivity of Box 3 upon the 77-GA mutation, the rescue significantly increased its reactivity (Extended Data Fig. 3a,b). These findings support the role of the three highlighted regions in forming an ensemble of states in which Box 2 and Box 3 compete for base pairing to Box 1.

To extend our in vitro observations to living cells, we performed high-coverage DMS-MaPseq of the RORC switch in vivo in the reporter context (Extended Data Fig. 3c). Using a DMS concentration sufficient to cause multiple modifications to the same RNA molecule, we implemented the DRACO computational approach[28], which identified two distinct clusters in both biological replicates, representing the two conformations, at relative proportions of 27% to 73% (Fig. 3d and Extended Data Fig. 3e). The profiles of these clusters were distinct ($P = 0.18$ and $P = 0.72$ in replicates 1 and 2, respectively) but showed high correlation within each cluster across replicates (Extended Data Fig. 3d). To ascertain whether sequence mutations similarly influence the conformational equilibrium in vivo, we conducted DMS-MaPseq on the two rescue mutant sequences (Extended Data Fig. 3f). This analysis corroborated our SHAPE findings: the (63-TC,77-GA) mutation stabilized conformation 2, while the (65-GT,117-AC) mutation favored conformation 1. The alignment of in vitro SHAPE and in vivo DMS-MaPseq results reinforces the notion that the RORC switch consistently exhibits its conformational dynamics across both experimental settings.

To determine whether the RORC element functions as a dynamic RNA switch or simply represents a static equilibrium of two conformations, we investigated whether the proportions of its alternative conformations change inside cells. To this end, we introduced a reporter containing the RORC sequence into five cell lines representing diverse genetic backgrounds: LNCaP (prostate), MCF-7 (breast), HepG2 (liver), ZR-75-1 (breast), 293T (kidney) and LS174T (colon). Using DMS-MaPseq, we assessed the conformational dynamics of the RORC switch in these cell lines. Our findings confirm not only that the relative proportions of the two conformations vary among these cell lines but they also demonstrate a strong anticorrelation in the accessibility of Boxes 2 and 3 ($R = -0.75$) (Fig. 3e). This anticorrelation supports the hypothesis of their competitive base pairing with Box 1, further suggesting dynamic switching behavior.

To extend our analysis from the reporter to the endogenous context, we performed DMS-MaPseq targeting the endogenous RORC mRNA across the same five cell lines. This approach yielded similar observations: a strong anticorrelation in accessibility ($R = -0.81$, Fig. 3f) and variability in the relative proportions of the two conformations. Importantly, the conformational ratios across cell lines were highly correlated between the reporter and endogenous contexts ($R = 0.93$, Extended Data Fig. 3g), demonstrating the high relevance of the reporter screening approach to understanding the behavior of RNA switches in the context of their endogenous mRNA. These data strongly support the hypothesis that the RORC element functions as an RNA switch, adopting two alternative conformations, the balance of which is influenced by the cellular landscape.

Finally, we used single-particle cryo-EM to investigate the tertiary structures of the two RORC RNA switch conformations that we identified using SHAPE and DMS-MaPseq. Micrographs of the reference RORC RNA switch contain a mixture of compact and extended particles, with features suggestive of RNA secondary structure (Fig. 4a and Extended Data Fig. 4a–c), including elongated tertiary features consistent with A-form helices, as well as bends and junctions consistent with complex RNA folding (Extended Data Fig. 4d–f). Strikingly, particles of the conformation 1 mutant (77-GA) appear more extended, while those of the conformation 2 mutant (117-AC) are mostly compact (Fig. 4a). Cryo-EM image processing shows that reference RORC RNA can be classified into three structural classes (Classes A, B, and C), with the Class B structure absent in the (77-GA) mutant and Class A absent in the (117-AC) mutant (Fig. 4b). This analysis suggests that Class A can be assigned to the more extended conformation 1, and Class B to the

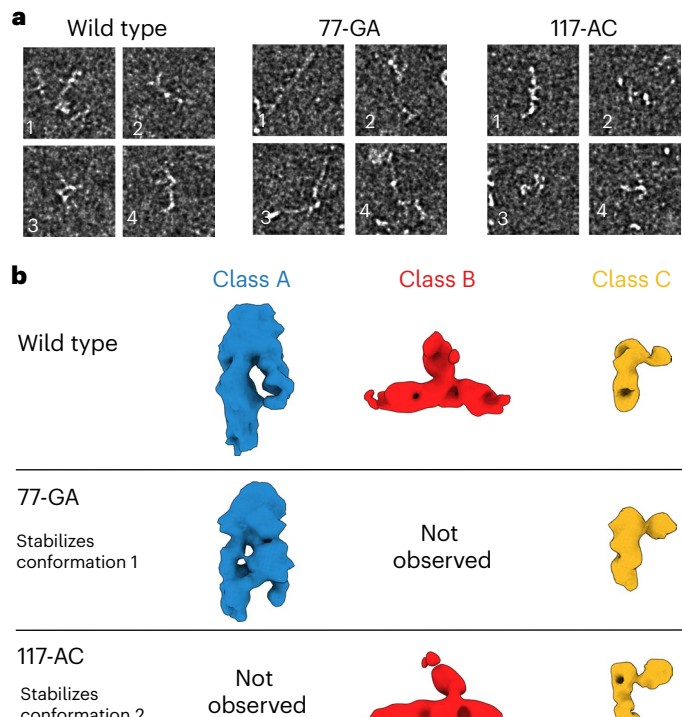

**Fig. 4 | Cryo-EM of RORC 3′ mRNA is consistent with dynamic exchange in a shallow energy landscape. a**, Cryo-EM of wild-type RORC mRNA, 77-GA mutant and 117-AC mutant, as representative examples of qualitatively different compact and extended RNA-like particles. Different morphologies are indicated by numbered labels. Source micrographs were phase-flipped, Gaussian filtered and contrast inverted for display (see Extended Data Fig. 5). Scale bars, 50 nm. **b**, Three structural classes of the refolded RORC 3′mRNA element as determined on cryo-EM processing, with RNA-like features (top). Further cryo-EM imaging and 3D classification of the 77-GA mutant (middle) and 117-AC mutant (bottom) indicate that Class A is present in wild-type and 77-GA samples but absent from the 117-AC sample, and Class B is conversely present in wild-type and 117-AC samples but absent from the 77-GA mutant. Class C is common to all three samples. We thus assign Class A as the conformation 1 state, and Class B as the conformation 2 state. We propose Class 3 to represent a partly folded intermediate that is not disrupted in the mutated constructs.

compact conformation 2 (Fig. 4b). We propose that Class C, which is present in all three datasets, represents a folding intermediate lacking the tertiary interactions made by either Boxes 2 or 3. Although the extreme flexibility of the RNA limits the resolution of the reconstructions to approximately 10 Å (Extended Data Fig. 5g–i), it is sufficient for discrimination of these different RNA folds. These results confirm that the RORC RNA switch indeed adopts distinct tertiary structures in solution and that the designed mutations heavily bias toward one conformation or the other.

## Alternative conformations of the *RORC* RNA switch play divergent roles in gene regulation

Having validated that the RORC RNA switch can adopt two stable conformations, we next explored the distinct regulatory activities of each conformation. We engineered HEK293 cell lines to express eGFP reporters carrying RORC switch variants in the 3′UTR and assessed eGFP expression changes using flow cytometry. To specifically lock the switch in each conformation, we implemented two parallel strategies: for conformation 1, one strategy involved mutating Box 2 to prevent its pairing with Box 1 (mutant '73-CCCTATGA'), and another introduced mutations into both Boxes 1 and 3 to disrupt their interaction with Box 2 (mutant '61-TATATAA,116-TTATATA'). Remarkably, both strategies, despite modifying different parts of the sequence, induced

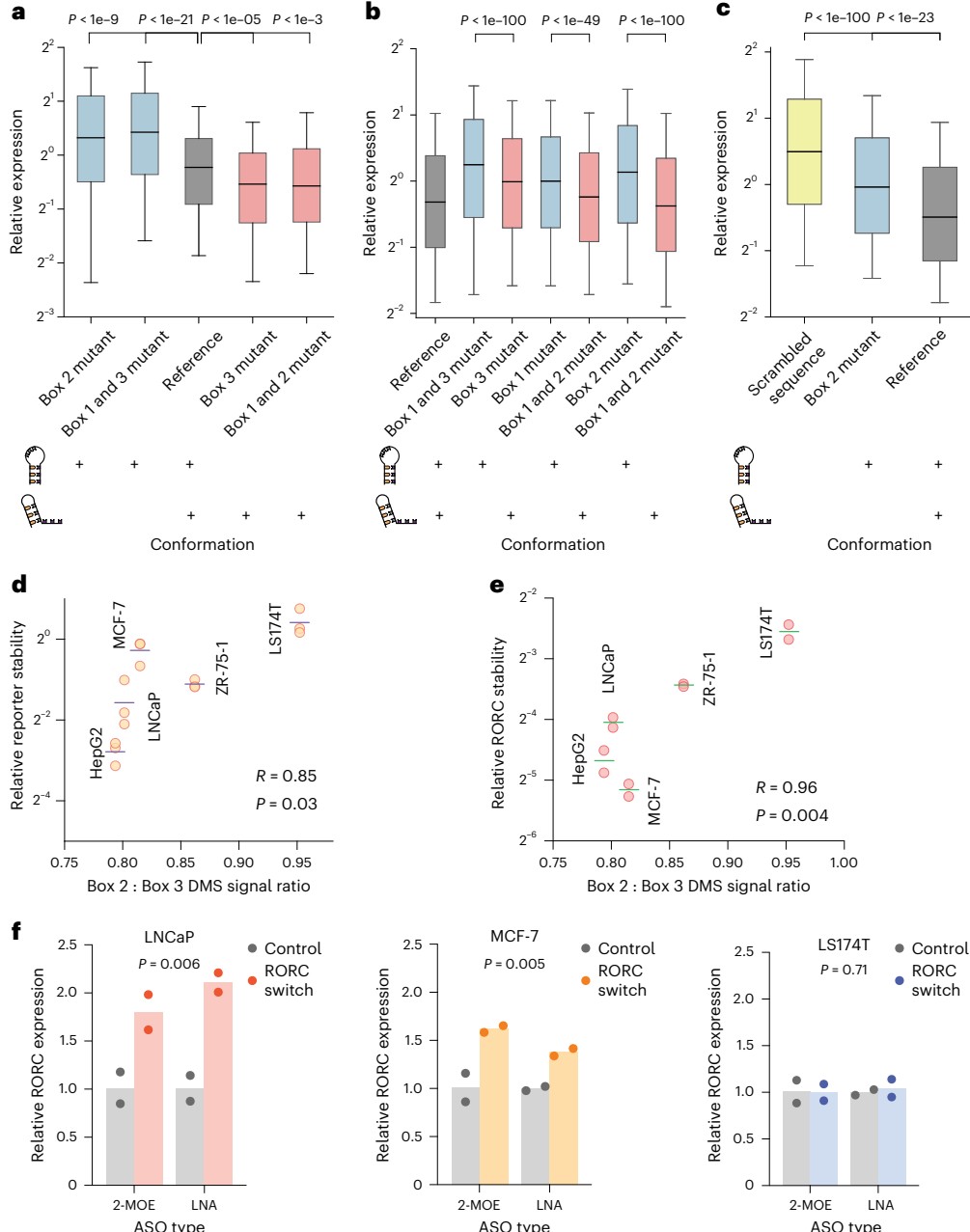

**Fig. 5 | The two alternative conformations of the RORC RNA switch have opposing effects on target gene expression. a–c,** Box plots of the relative expression of the reporter construct across different RNA conformations and sequences in HEK293 cells (**a**), reciprocal mutations (**b**) and primary Th17 cells (**c**). Relative expression is quantified as the ratio of eGFP to mCherry fluorescence for individual cells, as measured by flow cytometry ($n$ = 10,000 cells). The boxes shows the quartiles of the dataset, with the central line indicating the median value; the whiskers extend from the 10th to the 90th percentile. The colors denote specific RNA conformations or sequences: conformation 1 in blue, conformation 2 in red, reference sequence in gray, and a scrambled sequence in yellow. The diagrams below the box plots show the balance of the two conformations in the RNA populations, with existing conformations marked by a '+' sign. Statistical significance was determined with a two-sided independent $t$-test. **a,** The mutations left to right: 73-CCCTATGA; 61-TATATAA,116-TTATATA; reference; 116-CCCTAAG; 62-GCACAGT,73-ACTGTGC. $P$ values left to right: 1.1e−10, 2.6e−22, 1.6e−06, 0.00025. **b,** Effect of the shift in equilibrium between two conformations of the RORC switch on reporter gene expression for reciprocal mutations. The mutation–rescue experiments were performed as shown in Fig. 3b. The mutations left to right: reference; 65-GT,117-AC; 117-AC; 66-AC; 66-AC,74-GT; 77-GA; 63-TC,77-GA. $P$ values left to right: 7.1e−117, 3.6e−50,

5.9e−260. **c,** Effect of shift in the equilibrium between two conformations of the RORC switch on reporter gene expression in primary Th17 T cells. Human CD4+ T cells were infected with lentiviral constructs carrying one of the three sequences in the reporter gene's 3′UTR, and subsequently differentiated into Th17 cells. The mutations left to right: scrambled RORC RNA switch; 77-GA; reference. $P$ values left to right: 1.7e−124, 2.6e−24. **d,e,** Scatterplots of the relationship between the relative conformation ratio of the RORC element, as measured with DMS-MaPseq in reporter-expressing cell lines, and stability of the reporter mRNA ($n$ = 3 replicates) (**d**) and the endogenous RORC mRNA ($n$ = 2 replicates) (**e**), as measured by RT-qPCR following the α-amanitin treatment. The reporter contains the eGFP ORF, followed by the 3′UTR containing the RORC RNA switch sequence. Horizontal lines represent the mean of mRNA stability. Correlation of mean stability and the relative conformational ratio was measured using the Pearson correlation coefficient. **f,** Effect of ASOs on endogenous RORC mRNA expression, as measured by RT-qPCR. The targeting ASOs are complementary to Box 2 of the RNA switch; the control ASOs have the same nucleotide composition as the targeting ones but do not target the RORC RNA switch sequence. $P$ values were determined using the two-sided independent $t$-test, comparing the RORC-targeting and control ASOs, independent of the ASO chemistry. $n$ = 2 replicates. LNA, locked nucleic acids.

similar eGFP expression changes for each conformation: both mutants that stabilized conformation 1 increased reporter gene expression (Fig. 5a), while analogous strategies applied to stabilize conformation 2 decreased expression. We then investigated whether the modulation in gene expression was primarily influenced by the RNA's secondary structure rather than its sequence composition. Using cell lines stably expressing mutants from our earlier rescue–mutation experiments (Fig. 3b), we evaluated the impact on eGFP expression. Across three tested mutation–rescue pairs, the mutants favoring conformation 2 consistently showed reduced eGFP expression compared with those favoring conformation 1 (Fig. 5b). These findings from the reciprocal mutation–rescue experiments underscore the pivotal role of RNA secondary structure in the specific regulatory functions of the RORC RNA switch.

The RORC gene encodes the nuclear receptor *ROR-γ* that plays a crucial role in T-helper (Th)17 cell differentiation, a key process in the immune response, which is also implicated in autoimmune diseases[32,33]. To explore the functional impact of the RORC RNA switch in Th17 cells, we introduced into primary human CD4+ T cells a reporter construct carrying the RORC RNA switch sequence in the eGFP 3′ UTR. We then differentiated these cells into Th17 cells (Extended Data Fig. 6, ref. 34). Incorporating the native RORC RNA switch markedly reduced eGFP expression compared with a control with a scrambled sequence (Fig. 5c). Additionally, altering the switch's conformation with a 77-GA mutation (towards conformation 1) weakened this repression, confirming the activity of the RORC RNA switch in Th17 cells.

Having demonstrated the distinct regulatory effects of the RORC RNA switch's two conformations, we next asked whether their relative proportions in different cell types would result in differential regulation of the RORC transcript. To assess this, we compared the stability of the reporter mRNA containing the RORC switch between cell lines following inhibition of RNA polymerase II with α-amanitin. We discovered a strong correlation between the conformational ratio and reporter mRNA stability, indicating that higher proportions of conformation 1 resulted in higher stability, whereas higher proportions of conformation 2 resulted in lower stability ($R = 0.85$, $P = 0.03$, Fig. 5d). We extended this analysis to the endogenous RORC mRNA, where we observed a similar strong correlation ($R = 0.96$, $P = 0.004$, Fig. 5e).

Next, we investigated whether, instead of sequence mutations, trans-acting agents such as antisense oligonucleotides (ASOs) complementary to parts of the RNA switch sequence could shift the equilibrium between the two conformations and thereby influence gene expression[35]. We designed two ASOs to target the Box 2 region, aiming to shift the equilibrium towards conformation 1, which we would expect to increase the levels of RORC mRNA expression. We transfected three cell lines, representing different conformational ratios (LNCaP, MCF-7 and LS174T), with these ASOs carrying either 2′-O-(2-methoxyethyl) (2-MOE) oligoribonucleotides or locked nucleic acids. In both cases, ASO treatment led to a significant increase in RORC mRNA levels compared with nontargeting control ASO (Fig. 5f). Notably, this effect was more pronounced in cell lines with a higher proportion of conformation 2 (LNCaP, $P = 0.006$; MCF-7, $P = 0.005$) compared with those with a lower proportion (LS174T, $P = 0.71$). Together, these data further underscore the link between structural conformation and resultant gene expression, solidifying the role of the RORC element as a regulatory switch in its native gene context.

### Genome-scale genetic screens reveal molecular mechanisms underlying the RORC RNA switch

To investigate how the RORC RNA switch influences gene expression at the molecular level, we performed genome-wide CRISPRi screens in Jurkat T cells expressing one of two eGFP reporter constructs: one with the native RORC switch and another with the 77-GA mutation that favors conformation 1 (Extended Data Fig. 7a). These screens were intended to identify gene products, the depletion of which altered RORC RNA

switch-mediated control of reporter gene expression, indicating their functional connection to the RNA switch mechanism[36]. We focused on identifying two gene groups: those essential for repression induced by the RORC switch (as indicated by an increase in reporter gene expression), and those affecting the conformational dynamics of the switch (as indicated by a change in the ratio of reporter expression between the native switch and the 77-GA mutant).

To identify factors influencing the RORC RNA switch's repressive function, we analyzed the abundance of single-guide RNAs in cells with high versus low reporter gene expression in both screens. This analysis highlighted the NMD pathway, with top hits including core NMD factors such as SMG8, UPF1, UPF2 and UPF3B (Fig. 6a). Pathways associated with general gene expression, including ribosome biogenesis and endoplasmic reticulum stress, were also notable (Extended Data Fig. 7b). To pinpoint factors affecting the divergent activities of the switch's two conformations, we compared the distribution of sgRNAs across the high and low reporter expression bins between cells expressing the native switch and the 77-GA mutant. This comparison reinforced the central role of the NMD pathway (Fig. 6b), given that the knockdown of NMD components lessened the reporter expression difference between the native and mutant switch. Surprisingly, while knockdowns of SURF complex (that is, *SMG1–UPF1-eRF1–eRF3*; the complex that initiates NMD on stalled ribosomes[37]) components produced strong effects, the exon–junction complex (EJC) components did not produce significant changes in either screen, suggesting that the RORC RNA switch operates via a noncanonical EJC-independent NMD pathway[38,39]. Moreover, our findings suggest that the NMD pathway acts preferentially on conformation 2 of the RORC RNA switch, as evidenced by the stronger increase in expression of the 77-GA mutant compared with the native RORC sequence.

To confirm these results, we applied CRISPRi to individually knock down NMD factors in cells expressing the reference switch, the 77-GA mutant, or a scrambled sequence. Knockdowns of SURF complex members, but not EJC components, significantly affected the switch's repressive function, confirming our genome-wide screen results (Fig. 6c,d). Furthermore, reducing SURF complex expression also diminished the expression difference between the reference and 77-GA mutant, primarily by increasing reporter expression in the mutant (Extended Data Fig. 7d). This evidence indicates that NMD predominantly acts on conformation 2 of the RORC RNA switch.

Given its affinity for structured RNAs[40], we reasoned that *UPF1* might bind the two RORC RNA switch conformations with different affinities. To test this, we mixed together the reference and the Box 2 mutant (77-GA) reporter lines at a 1:1 ratio and measured *UPF1* binding using CLIP-qPCR (cross-linking and immunoprecipitation followed by qPCR). The reference RORC UTR sequence (containing a mixture of conformations 1 and 2) had significantly stronger binding to *UPF1* than its 77-GA mutant that could form only conformation 1 (Fig. 6e). Similarly, we observed a strong preference for *UPF1* to bind to a mutant 116-CCCTAAG that favors conformation 2 than to the 77-GA mutant, and this effect was even more pronounced than the difference between reference and 77-GA (logarithm of fold change of 1.12 versus 0.41). Together, these results underscore the preference of *UPF1* to bind to conformation 2 of the RORC switch (Extended Data Fig. 7e).

We reasoned that conformation-specific NMD would deplete mRNA molecules with conformation 1, thereby resulting in a relative increase in the proportion of conformation 2. To test this, we used NMDI14, a molecule that disrupts *SMG7–UPF1* interactions, to inhibit NMD[41]. Assessing the accessibility of Boxes 2 and 3 in endogenous RORC mRNA using DMS-MaPseq, we found a significant decrease in the accessibility of Box 2 upon NMD inhibition ($P = 0.03$, Fig. 6f), indicative of a shift towards conformation 2, possibly due to slower decay and accumulation of mRNAs in this conformation. Hence, inhibiting NMD led to a shift in the relative proportions of the two conformations.

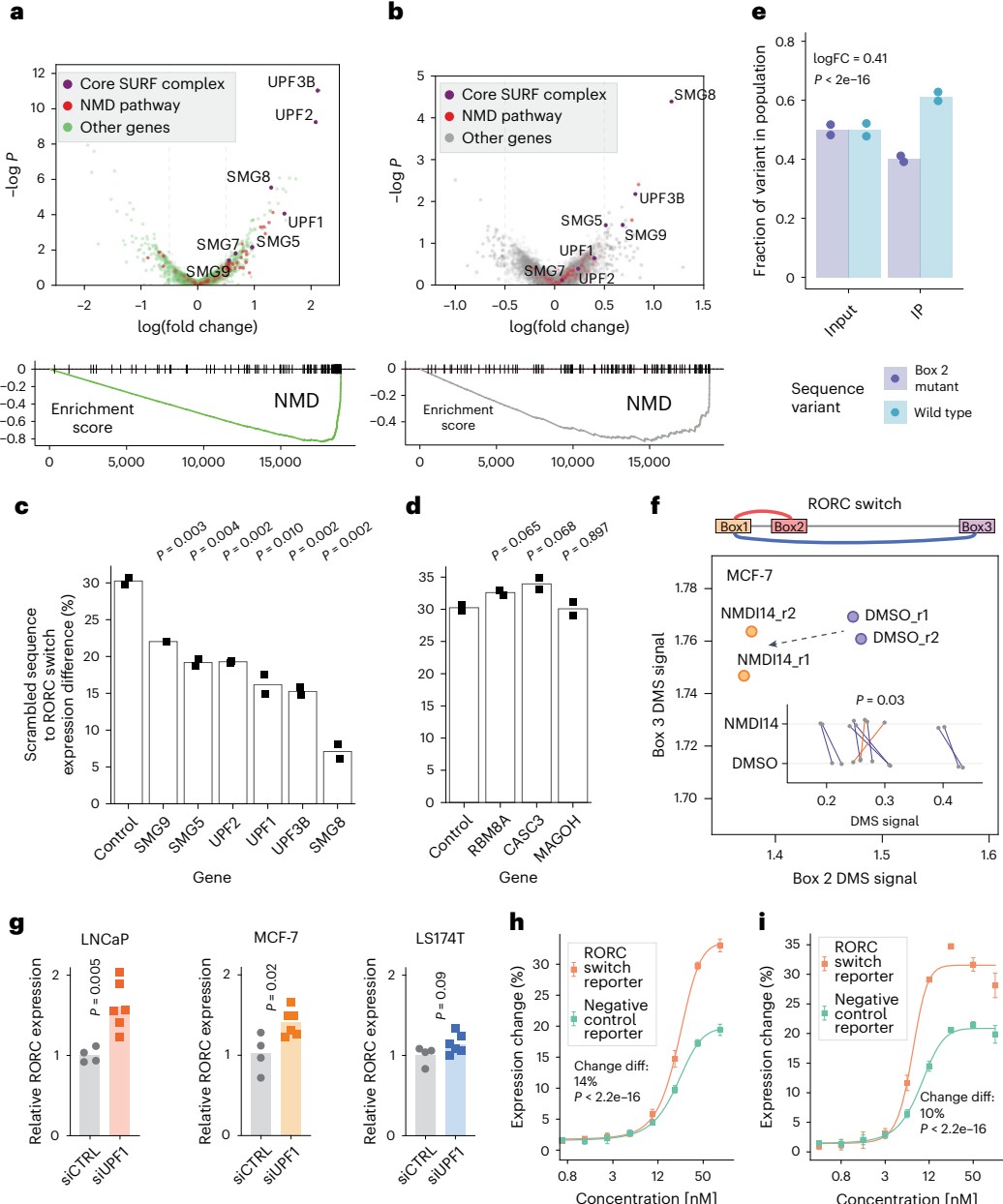

**Fig. 6 | Genome-wide CRISPRi screen identifies SURF complex as acting downstream of the RORC RNA switch. a**, Top: Expression change: high versus low: comparison of sgRNA representation between the bottom and the top quantiles of reporter gene expression (across both reference and 77-GA mutant cell lines), represented as a volcano plot. Genes, annotated as part of the NMD pathway by gene ontology (GO), are colored in red. The core components of the canonical NMD pathway are colored in purple and labeled. All other genes are colored in green. Bottom: Gene set enrichment analysis (GSEA) plot for the NMD pathway for the above comparison. −logP: negative logarithm of *P* value. **b**, Differences between conformations: wild type versus the 77-GA mutant. Comparison of ratios between top and bottom expression quantiles for the two cell lines. Higher values on the x axis indicate that sgRNAs targeting this gene have a stronger effect on reporter gene expression in the reference cell line compared with the 77-GA mutant cell line. Top: 'ratio of ratios' comparison[57] represented as a volcano plot. Genes are colored as in **a**. Bottom: GSEA plot for the NMD pathway for the above comparison. −logP: negative logarithm of *P* value. **c**,**d**, The effect of knockdown of SURF (**c**) and EJC (**d**) member proteins on the RORC RNA switch reporter gene expression, relative to a scrambled sequence. The individual genes were knocked down using the CRISPRi system in both the reference and the scrambled cell lines, then the change of reporter gene expression was measured using flow cytometry (*n* = 2 replicates). The bar plots show the ratio of the expression of the scrambled sequence to that

of the wild-type sequence of the RORC RNA switch. *P* values were calculated using the two-sided Student's *t*-test. **e**, Bar plots of the fractions of reads carrying the wild-type RORC switch sequence or B77-GA mutant variant in the UPF1 cross-linking and immunoprecipitation (CLIP) library. Left: input RNA libraries, extracted from the wild-type and 77-GA mutant-expressing Jurkat cells, mixed at a 1:1 ratio. Right: libraries after anti-UPF1 immunoprecipitation (IP). The fractions are normalized by the variant fractions in the input libraries. The *P* value was calculated using the translation efficiency ratio test[58]. FC, fold change. *n* = 2 replicates. **f**, The effect of NMDI14 on the accessibility of the Box 2 and the Box 3 regions of the RORC element, as measured by DMS-MaPseq. Changes in individual nucleotide accessibility are shown on the inner plot. Statistical significance was determined using a two-sided independent *t*-test. **g**, The effect of UPF1 knockdown on endogenous RORC mRNA expression, as measured by RT-qPCR (control, *n* = 4 replicates; UPF1 knockdown, *n* = 6 replicates). siCTRL, non-targeting dicer-substrate small interfering RNA; siUPF1, UPF1-targeting dicer-substrate small interfering RNA. *P* values were calculated using the two-sided Student's *t*-test. **h**,**i**, Effect of the proteasome inhibitors carfilzomib (**h**) and bortezomib (**i**) on the RNA switch-mediated expression change (*n* = 4 replicates). Data are given as the mean ± s.d. Statistical significance was determined using dose–response modeling followed by ANOVA, to compare the fitted models to assess differences in the effect of the inhibitors on the RNA switch-mediated expression.

**a**

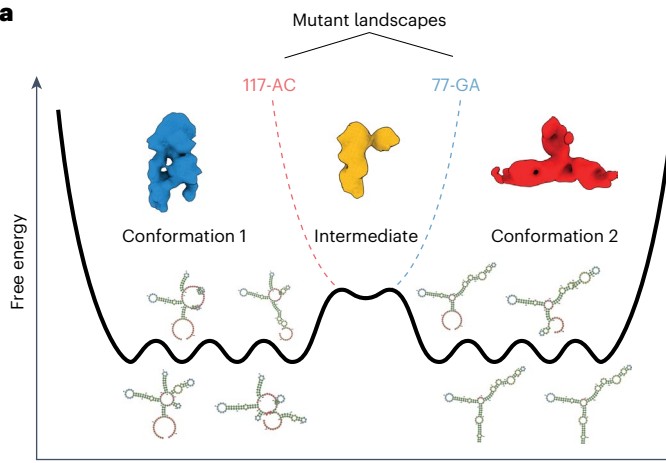

**b**

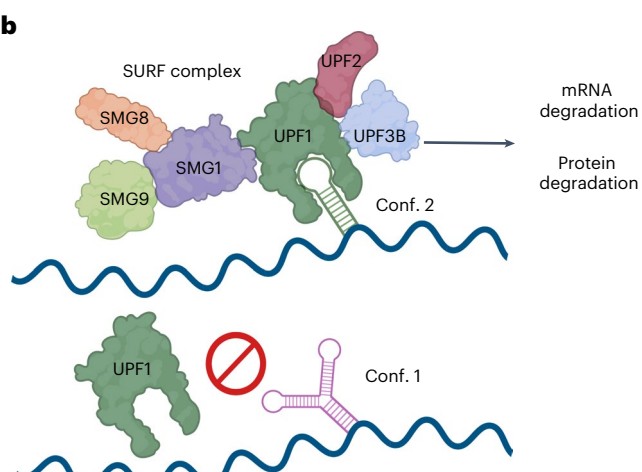

**Fig. 7 | The proposed mechanism of RORC RNA switch functioning.**
**a**, Schematic diagram of a shallow energy landscape for the RORC 3′ mRNA element. Shallow global minima characterizing the conformation 1 (cryo-EM Class A) and conformation 2 (cryo-EM Class B) structures themselves comprise multiple local minima in which various secondary structure elements fold or unfold while preserving overall tertiary structure and biological activity. These local minima are illustrated by secondary structure models for various DRACO cluster members. The two global minima are separated by a kinetic barrier that represents a partially folded intermediate (cryo-EM Class 3). The two dashed lines indicate alterations to the global landscape exhibited by the mutant sequences, blue for the 77-GA mutant and red for the 117-AC mutant. These altered landscapes eliminate one of the global minima without disrupting the intermediate. **b**, Proposed mechanism of the RORC RNA switch. The RNA switch exists in an ensemble of two states. One of them is recognized by the SURF complex; such recognition triggers mRNA degradation (likely to be mediated by SMG5) and protein degradation (mediated by the proteasome), thus affecting gene expression.

Having demonstrated the conformation-specific effect of NMD on the RORC switch in the reporter context, we sought to extend our analysis to the endogenous RORC mRNA. We knocked down UPF1 in various cell lines and assessed the levels of endogenous RORC mRNA using quantitative polymerase chain reaction with reverse transcription. UPF1 knockdown in various cell lines led to a substantial increase in RORC mRNA expression, notably more pronounced in cell lines with a higher prevalence of conformation 2 (LNCaP, $P = 0.005$; MCF-7, $P = 0.02$) compared with those with a lower prevalence (LS174T, $P = 0.09$) (Fig. 6g). This result emphasizes the role of *UPF1* in regulating endogenous RORC mRNA stability in a conformation-dependent manner.

Considering the NMD pathway's role in directing proteins translated from aberrant mRNA to proteasomal degradation[42], we reasoned that the RORC RNA switch might similarly target its gene product. To test this, we treated reporter cells with the proteasome inhibitors carfilzomib and bortezomib, each acting through different mechanisms. Proteasome inhibition resulted in a significantly greater increase in eGFP expression in cells expressing the RORC switch compared with the control (Fig. 6h,i), indicating that NMD-induced proteasomal degradation of the switch-containing gene product contributes to the observed effect on gene expression.

We propose that *UPF1* preferentially recognizes switch conformation 2 over conformation 1, and that the recruitment of the SURF complex by *UPF1* consequently leads to decreased gene expression through proteasome-mediated degradation of translation products and mRNA decay, preventing repeated rounds of translation (Fig. 7b). Moreover, sequence mutations that influence the conformational equilibrium not only alter the RNA's energy landscape but also modulate SURF recruitment and RNA stability, reflecting the nuanced control of gene repression by the switch. The mechanisms underlying the switching between conformations, however, remain an area for further investigation.

Collectively, we show that the RORC RNA switch influences gene expression through conformation-specific engagement of NMD factors that lead to control of mRNA and protein stability. Importantly, the RORC switch is only one example out of 245 functionally validated human RNA switches identified in this work, emphasizing the power of our SwitchSeeker approach to illuminate new areas of eukaryotic RNA biology.

## Discussion

Historically, RNA switches were identified primarily through biochemical experimentation, measuring direct ligand interactions[43,44], and comparative genomics to identify conserved noncoding regions that act as cis-regulatory elements in bacteria[45,46]. These methods, however, present challenges in eukaryotic contexts due to the dynamic nature of mRNA structures and the complexity of eukaryotic gene regulation[22,24]. Additionally, the vast genomic landscape and low sequence conservation in eukaryotes complicate the direct application of these approaches[47–49]. While numerous tools and algorithms exist for riboswitch prediction (reviewed in refs. 50,51), few of those focus on de novo discovery that is family-agnostic. The exceptions include SwiSpot[10], which focuses on identifying the putative switching sequence, and the conditional probability-based method[52]. None of these algorithms has been shown to predict functional RNA switches from novel families in eukaryotic genomes. Addressing these challenges, SwitchSeeker integrates biochemistry, systems biology and functional genomics to create a comprehensive platform for RNA switch discovery and characterization in eukaryotes. By covering the entire discovery process, from de novo predictions to the annotation of mechanisms, SwitchSeeker overcomes the limitations of existing methods. Looking forward, its capability to scale across complete transcriptomes sets the stage for a thorough characterization of RNA switches across diverse cell types and organisms, enhancing our understanding of their roles across the tree of life.

Advancements in genomic technologies such as RNA secondary structure probing (DMS-seq, SHAPE-seq) and single-particle cryo-EM have been instrumental in our systematic exploration of RNA switches, enabling us to delve into the diverse conformations of RNA molecules and their three-dimensional structures despite challenges such as size and flexibility[28,29,53]. This has opened up opportunities to study the functional differences between alternative RNA conformations and their role in gene expression control. Our DMS-MaPseq and cryo-EM data suggest that the RORC 3′ mRNA element inhabits a shallow energy landscape with two rugged minima linked to two major molecular conformations (Fig. 7a), thereby validating the SwitchSeeker approach to identifying RNA molecules with bi-stable energy landscapes. Genome-wide CRISPRi screens identified the EJC-independent

NMD pathway as a key mediator of the gene regulatory mechanism of the RORC switch. Together, our studies of the RORC switch not only uncover new regulatory biology but also provide a blueprint on how the SwitchSeeker pipeline can enable rapid functional and mechanistic characterization of new RNA switches.

RNA structure is known to influence gene expression in health and disease[35], as shown by our recent identification of specific RNA structures that influence splicing in metastatic cancers[54]. However, dynamic RNA structures such as RNA switches are a relatively unexplored aspect of gene expression control in eukaryotes. Our observations indicate a prevalence of RNA switches in the human transcriptome, suggesting that RNA conformation-dependent gene regulation is a widespread phenomenon. In our study we chose stringent criteria for selecting RNA switches, requiring them to be bi-stable in vivo, meaning that they populate two mutually exclusive structural conformations. However, it is important to note that not all RNA switches may conform to this binary model; some, such as the HIV-1 TAR RNA, have transient but functional conformations[55], and others might present multistability, adding layers to regulatory control. Modifications to the SwitchSeeker platform will be necessary to explore these distinct classes of RNA structural elements.

While SwitchSeeker offers a robust framework for identifying functional RNA structural switches, there are several caveats and limitations to consider. First, identifying RNA switches that operate under specific cellular conditions requires structure probing assays to be conducted in those exact conditions, which can be challenging and resource intensive. Additionally, SwitchSeeker does not identify ligands for RNA switches; this necessitates complementary approaches to uncover the specific molecules interacting with these RNA elements. Future technological advancements could significantly enhance the tool's efficacy. Currently, the absence of high-quality RNA structure datasets across full transcriptomes limits the comprehensive application of SwitchSeeker. The development of such datasets would enable more efficient and accurate RNA switch identification. Moreover, integrating additional functional assays, such as those targeting RNA switches that influence splicing, could broaden the scope and impact of SwitchSeeker.

The known examples of human RNA switch mechanisms include mutually exclusive binding of RNA-binding proteins by two different RNA conformations[8] and m6A modification-based switching[7]. In this study, we introduce a novel switch mechanism that operates via the NMD pathway, suggesting a vast potential for diverse metabolic pathways in RNA switch functionality. SwitchSeeker's utility lies in its ability to identify and elucidate these mechanisms in high throughput, irrespective of their specific pathways. The modulation of gene expression through shifts in RNA conformation, as achieved with ASOs in this study, opens new possibilities for targeting RNA switches in future therapeutics. SwitchSeeker is available for use and adaptation, and we hope that it will pave the way for many new discoveries in RNA-based regulation in eukaryotes.

## Online content

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

## Methods

### SwitchFinder: detailed description of the algorithm

**Conflicting base pairs identification.** Conflicting base pairs were detected using a modification of the MIBP algorithm developed by L. Lin and W. McKerrow[59]. First, a large number of folds (default N = 1,000) is sampled from the Boltzmann distribution. If structure probing data (such as DMS-seq or SHAPE-seq) is provided, the Boltzmann distribution modeling software (part of the RNAstructure package[56]) incorporates the data as a pseudofree energy change term. Then, the base pairs are filtered: the base pairs that are present in almost all of the folds or are absent from almost all of the folds are removed from the further analysis. Then, mutual information for each pair of base pairs is estimated. To do so, each base pair is represented as a binary vector of length N, where N is the number of folds considered; in this binary vector, a given fold is represented as 1 if this base pair is present there, or as 0 if it is not. Mutual information between each two base pairs is calculated as in ref. [60]. This results in an M × M table of mutual information values, where M is the number of base pairs considered. Then, the sum of each row of the square table is calculated. In the resulting vector K of length M, each base pair is represented by a sum of mutual information values across all of the other base pairs. Then, only the base pairs for which the sum of mutual information values passes the threshold of U × MAX(K) are considered, where U is a parameter (default value 0.5). We call the base pairs that pass this threshold the 'conflicting base pairs'.

**Conflicting stems identifications.** Once the conflicting base pairs are identified, they are assembled into conflicting stems, or series of conflicting base pairs that directly follow each other and therefore could potentially form a stem-like RNA structure. More specifically, the base pairs (a, b) and (c, d) form a stem if either (a == c − 1) and (b == d + 1), or (a == c + 1) and (b == d − 1). The stem is defined as a pair of intervals ((u, v), (x, y)), where v − u == y − x. Then, the conflicting stems are filtered by length: only the stems that are longer than a certain threshold value (default value: 3) are considered. Among these stems, the stems that directly conflict with each other are identified. Two stems ((u$_1$, v$_1$), (x$_1$, y$_1$)) and ((u$_2$, v$_2$), (x$_2$, y$_2$)) conflict with each other if there is an overlap longer than a threshold value between either (u$_1$, v$_1$) and (u$_2$, v$_2$), or (u$_1$, v$_1$) and (x$_2$, y$_2$), or (x$_1$, y$_1$) and (u$_2$, v$_2$), or (x$_1$, y$_1$) and (x$_2$, y$_2$). The default threshold value is 3. The pairs of conflicting stems are sorted by the average value of their K values (sums of mutual information). The highest scoring pair of conflicting stems is considered the winning prediction, representing the major switch between two of the local minima present in the energy folding landscape of the given sequence. If no pairs of conflicting stems pass the threshold, SwitchFinder reports that no potential switch is identified for the given sequence.

**Identifying the two conflicting structures.** Given the prediction of the two conflicting stems, the folds that represent the two local minima of the energy folding landscape are predicted. Importantly, SwitchFinder focuses on optimizing the prediction accuracy, as opposed to the commonly used approach of energy minimization[61]. The MaxExpect program from the RNAstructure package[56] is used; the base pairings of each of the conflicting stems are provided as folding constraints (in Connectivity Table format). Furthermore, the two predicted structures are referred to as conformations 1 and 2.

**Activation barrier estimation.** The RNApathfinder software[62] is used to estimate the activation energy needed for a transition between the conformations 1 and 2.

**Classifier for prediction of RNA switches.** The curated representative alignments for each of the 50 known riboswitch families were downloaded from the Rfam database[9]. Each sequence is complemented by its shuffled counterpart (while preserving dinucleotide frequencies[63]).

For all of the sequences, the two conflicting conformations, their folding energies and their activation energies are predicted as above. To estimate the performance of SwitchFinder for a given riboswitch family, all of the sequences from this family are placed into the test set, while all of the sequences from the other families are placed into the training set. Then, a linear regression model is trained on the training set, in which the response variable is binary and indicates whether the sequence is a real riboswitch or is a shuffled counterpart, and the predictor variables are the average folding energy of the two conformations and the activation energy of the transition between them. The trained linear regression model is then run on the test set, and its performance is estimated using the receiver operating characteristic curve.

**Prediction of RNA switches in human transcriptome.** The coordinates of 3′UTRs of the human transcriptome were downloaded from UCSC Table Browser[64], table tb_wgEncodeGencodeBasicV28lift37. The sequences of 3′UTRs were cut into overlapping fragments of 186 nucleotides in length (with overlaps of 93 nucleotides). For all of the sequences, the two conflicting conformations, their folding energies and their activation energies were predicted as above. A linear regression model was trained as described above on all 50 known riboswitch families. The model was applied to the 3′UTR fragments from the human genome, and the fragments were sorted according to the model prediction scores. The top 3,750 predictions were selected for further investigation.

**Incorporation of in vivo probing data.** In vivo probing data, such as DMS-MaPseq, is used to apply pseudoenergy restraints when sampling folds from the Boltzmann distribution (that is, using the −SHAPE parameter in RNAstructure package commands[56]). To test the hypothesis of whether the in vivo probing data support the presence of two conflicting conformations in a given sequence, the following workflow was used. First, the two conflicting folds were predicted with SwitchFinder using in silico folding only. Then, SwitchFinder was run on the same sequence with the inclusion of in vivo probing data. If the same two conflicting folds were predicted among the top conflicting folds, the probing data were considered supportive of the presence of the two predicted conformations.

**Mutation generation.** To shift the RNA conformation ensemble towards one or another state, mutations of two types were introduced.
(1) 'Strengthen a stem' mutations: given two conflicting stems ((u$_1$, v$_1$), (x$_1$, y$_1$)) and ((u$_2$, v$_2$), (x$_2$, y$_2$)), one of the stems (for example, the first one) was changed in a way that would preserve its base pairing but deny the possibility of forming the second stem. To do so, the nucleotides in the interval (u$_1$, v$_1$) were replaced with all possible sequences of equal length, and the nucleotides (x$_1$, y$_1$) were replaced with the reverse complement sequence. Then, the newly generated sequences were filtered by two predetermined criteria: (i) the second stem cannot form more than a fraction of its original base pairs (default value 0.6), and (ii) the modified first stem cannot form long paired stems with any region of the existing sequence (default threshold length 4). The sequences that passed both criteria were ranked by the introduced change in the sequence nucleotide composition; the mutations that changed the nucleotide composition the least were chosen for further analysis. Each mutated sequence was additionally analyzed by SwitchFinder to ensure that the Boltzmann distribution is heavily shifted towards the desired conformation.
(2) 'Weaken a stem' mutations: given two conflicting stems ((u$_1$, v$_1$), (x$_1$, y$_1$)) and ((u$_2$, v$_2$), (x$_2$, y$_2$)), one of the stems (for example, the second one) was changed in such a way that this stem would not be able to form base pairing, while the base pairing of the other stem (in this example, the first stem) would be preserved.

To do so, the nucleotides in either of the intervals ($u_2$, $v_2$) or ($x_2$, $y_2$) were replaced with all possible sequences of equal length. The newly generated sequences were filtered by three predetermined criteria: (i) the first stem stays unchanged, (ii) the second stem cannot form more than a fraction of its original base pairs (default value 0.6), and (iii) the modified part of the sequence cannot form long paired stems with any region of the existing sequence (default threshold length 4). The sequences that passed all of the criteria were ranked by the introduced change in the sequence nucleotide composition: the mutations that changed the nucleotide composition the least were chosen for further analysis. Each mutated sequence was additionally analyzed using SwitchFinder to ensure that the Boltzmann distribution is heavily shifted towards the desired conformation.

## Cell culture

All cells were cultured in a 37 °C 5% $CO_2$ humidified incubator. The HEK293 cells (purchased from ATCC, cat. no. CRL-3216) were cultured in DMEM high-glucose medium supplemented with 10% FBS, ʟ-glutamine (4 mM), sodium pyruvate (1 mM), penicillin (100 units ml$^{-1}$), streptomycin (100 µg ml$^{-1}$) and amphotericin B (1 µg ml$^{-1}$) (Gibco). The Jurkat cell line (purchased from ATCC, cat. no. TIB-152) was cultured in RPMI-1640 medium supplemented with 10% FBS, glucose (2 g l$^{-1}$), ʟ-glutamine (2 mM), 25 mM HEPES, penicillin (100 units ml$^{-1}$), streptomycin (100 µg ml$^{-1}$) and amphotericin B (1 µg ml$^{-1}$) (Gibco). All cell lines were routinely screened for mycoplasma with a PCR-based assay.

## Cryo-electron microscopy

**Sample preparation and data collection.** A total of 3.5 µl target mRNA at an approximate concentration of 1.5 mg ml$^{-1}$ was applied to gold, 300 mesh transmission electron microscopy grids with a holey carbon substrate of 1.2 µm and 1.3 µm spacing (Quantifoil). The grids were blotted with no. 4 filter papers (Whatman) and plunge frozen in liquid ethane using a Mark IV Vitrobot (Thermo Fisher), with blot times of 4–6 s, blot force of −2, at a temperature of 8 °C and 100% humidity. All grids were glow discharged in an easiGlo (Pelco) with rarefied air for 30 s at 15 mA, no more than 1 h prior to preparation. Duplicate wild-type and mutant RNA specimens were imaged under different conditions on several microscopes as per Data File S8; all were equipped with K3 direct electron detector (DED) cameras (Gatan), and all data collection was performed using SerialEM[65]. Detailed data collection parameters are listed in Data File S8.

**Image processing.** Dose-weighted and motion-corrected sums were generated from raw DED movies during data collection using University of California, San Francisco (UCSF) MotionCor2[66]. Images from super-resolution datasets were downsampled to the physical pixel size before further processing. Estimation of the contrast transfer function (CTF) was performed in CTFFIND4[67], followed by neural net-based particle picking in EMAN2[68]. Two-dimensional (2D) classification, *ab initio* three-dimensional (3D) classification, and gold-standard refinement were done in cryoSPARC[69]. CTFs were then re-estimated in cryoSPARC and particles repicked using low-resolution (20 Å) templates generated from chosen 3D classes. Extended datasets were pooled when appropriate, and particle processing was repeated through gold-standard refinement as before. All structure figures were created using UCSF ChimeraX (ref. 70). Further details are given in Data File S7 and Extended Data Fig. 5.

## Reporter vector design and library cloning

First, mCherry-P2A-Puro fusion was cloned into the BTV arbovirus backbone (Addgene, cat. no. 84771). Then, the vector was digested with MluI-HF and PacI restriction enzymes (NEB), with the addition of Shrimp Alkaline Phosphatase (NEB). The digested vector was purified with the Zymo DNA Clean and Concentrator-5 kit.

DNA oligonucleotide libraries (one for functional screen and one for massively parallel mutagenesis analysis) consisting of 7,500 sequences in total were synthesized by Agilent. The second strand was synthesized using Klenow Fragment (3′ → 5′ exo-) (NEB). The double-stranded DNA library was digested with MluI-HF and PacI restriction enzymes (NEB) and run on a 6% TBE (Tris base, boric acid, EDTA) polyacrylamide gel. The band of the corresponding size was cut out and the gel was dissolved in the DNA extraction buffer (10 mM Tris, pH 8, 300 mM NaCl, 1 mM EDTA). The DNA was precipitated with isopropanol. The digested DNA library and the digested vector were ligated with T4 DNA ligase (NEB). The ligation reaction was precipitated with isopropanol and transformed into MegaX DH10B T1R electrocompetent cells (Thermo Fisher). The library was purified with Zymo-PURE II Plasmid Maxiprep Kit (Zymo). The representation of individual sequences in the library was verified by sequencing the resulting library on an MiSeq instrument (Illumina).

## Massively parallel reporter assay

The DNA library was co-transfected with pCMV-dR8.91 and pMD2.G plasmids using TransIT-Lenti (Mirus) into HEK293 cells, following the manufacturer's protocol. Virus was collected 48 h after transfection and passed through a 0.45 µm filter. HEK293 cells were then transduced overnight with the filtered virus in the presence of 8 µg ml$^{-1}$ polybrene (Millipore); the amount of virus used was optimized to ensure an infection rate of ~20%, as determined by flow cytometry. The infected cells were selected with 2 µg ml$^{-1}$ puromycin (Gibco). Cells were collected at 90–95% confluency for sorting and analysis on a BD FACSaria II sorter. The distribution of mCherry : GFP ratios was calculated. For sorting a library into subpopulations, we gated the population into eight bins each containing 12.5% of the total number of cells. A total of 1.2 million cells were collected for each bin to ensure sufficient representation of sequence in the population in two replicates each. For each subpopulation, we extracted genomic DNA and total RNA with the Quick-DNA/RNA Miniprep kit. gDNA was amplified by PCR with Phusion polymerase (NEB) using the primers CAAGCAGAAGACGGCATACGAGAT–i7– GTGACTGGAGTTCA-GACGTGTGCTCTTCCGATCACTGCTAGCTAGATGACTAAACGCG and AATGATACGGCGACCACCGAGATCTACAC–i5– ACACTCTTTC-CCTACACGACGCTCTTCCGATCTGTGGTCTGGATCCACCGGTCC. Different i7 indexes were used for eight different bins, and different i5 indexes were used for the two replicates. RNA was reverse transcribed with Maxima H Minus Reverse Transcriptase (Thermo Fisher) using primer CTCTTTCCCTACACGACGCTCTTCCGATCT-NNNNNNNNNNNTGGTCTGGATCCACCGGTCCGG. The complementary DNA was amplified with Q5 polymerase (NEB) using primers CAAGCAGAAGACGGCATACGAGAT–i7–GTGACTGGAGTTCA-GACGTGTGCTCTTCCGATCCTGCTAGCTAGATGACTAAACGC and CAAGCAGAAGACGGCATACGAGAT–i5–GTGACTGGAGTTCAGACGT-GTGCTCTTCCGATCTTACCCGTCATTGGCTGTCCA. Different i7 indexes were used for eight different bins, and different i5 indexes were used for the two replicates. The amplified DNA libraries were size purified with the Select-a-Size DNA Clean and Concentrator MagBead Kit (Zymo). Deep sequencing was performed using the HiSeq4000 platform (Illumina) at the UCSF Center for Advanced Technologies.

The adapter sequences were removed using cutadapt[71]. For RNA libraries, the unique molecular identifier (UMI) was then removed from the reads and appended to read names using UMI tools[72]. The reads were matched to the fragments using the bwa mem command. The reads were counted using featureCounts[73]. The read counts were normalized using median of ratios normalization[74]. The one-way chi-squared test was used to estimate how different its distribution across the sorting bins is from the null hypothesis (that is uniform distribution). mRNA stability was estimated by comparing the RNA and DNA read counts with MPRAnalyze[75].

## Massively parallel mutagenesis analysis

**Library design and measurement.** For each candidate switch, two alternative conformations were identified using SwitchFinder. Each conformation is defined by a stem structure: ((u1, v1) and (x1, y1)) and ((u2, v2), (x2, y2)), representing two conflicting stems. The Switch-Finder mutation generation algorithm was used to design four mutations in the candidate switch sequence: A, 'strengthen a stem' mutation favoring conformation 1: the regions (u1, v1) and (x1, y1) are altered while preserving complementarity; B, 'weaken a stem' mutation favoring conformation 1: either the region (u2, v2) or (x2, y2) is modified, preserving the regions (u1, v1), (x1, y1); C, 'strengthen a stem' mutation favoring conformation 2: the regions (u2, v2), (x2, y2) are changed while maintaining complementarity; and D, 'weaken a stem' mutation favoring conformation 2: either the region (u1, v1) or (x1, y1) is altered, ensuring that the regions (u2, v2), (x2, y2) remain intact.

Subsequently, the mutated sequences for selecting candidate RNA switches, along with the reference sequence, were pooled into a single DNA oligonucleotide library. The impact of each sequence on reporter gene expression was evaluated in cells, as outlined in the Massively Parallel Reporter Assay section. Consequently, each candidate RNA switch in the library is represented by its reference sequence, two mutated sequences favoring conformation 1 (A and B), and two mutated sequences favoring conformation 2 (C and D).

**Candidate RNA switch ranking.** For each candidate RNA switch, its effect on reporter gene expression was assessed in cells, following the protocol described in the Massively Parallel Reporter Assay section. This resulted in 16 measurements, corresponding to normalized read counts in sorting bins 1 (lowest expression) to bin 8 (highest expression), across two replicates; these arrays of counts are referred to as 'bin_counts'. Measurements were obtained for mutants A, B, C, D, and the reference sequence. Correlations between the effects of mutations designed to favor the same or opposite conformations were computed as follows: correlation_same_1 = Pearsonr(bin_counts(mutant A), bin_counts(mutant B)); correlation_same_2 = Pearsonr(bin_counts(mutant C), bin_counts(mutant D)); correlation_opposite_1 = Pearsonr(bin_counts(mutant A), bin_counts(mutant C)); and correlation_opposite_2 = Pearsonr(bin_counts(mutant A), bin_counts(mutant D)). The score of each candidate switch was then calculated as: score = mean(correlation_same_1, correlation_same_2) − mean(correlation_opposite_1, correlation_opposite_2). Candidate switches were ranked based on this score. Those with a score exceeding the mean + 1 s.d. were considered significant.

## DMS-MaPseq

DMS-MaPseq was performed as described in ref. [54]. In brief, cells were incubated in culture with 1.5% DMS (Sigma) at room temperature for 7 min, the media was removed, and DMS was quenched with 30% BME (β-mercaptoethanol). Total RNA from DMS-treated cells and untreated cells was then isolated using Trizol (Invitrogen). RNA was reverse transcribed using TGIRT-III reverse transcriptase (InGex) and target-specific primers. PCR was then performed to amplify the desired sequences and to add Illumina-compatible adapters. The libraries were then sequenced on a HiSeq4000 instrument (Illumina).

Pear (v0.9.6) was used to merge the paired reads into a single combined read. The UMI was then removed from the reads and appended to read names using UMI tools (v1.0). The reads were then reverse complemented (fastx toolkit) and mapped to the amplicon sequences using bwa mem (v0.7). The resulting bam files were then sorted and deduplicated (umi_tools, with method flag set to unique). The alignments were then parsed for mutations using the CTK (CLIP Tool Kit) software. The mutation frequency at every position was then reported. The signal normalization was performed using boxplot normalization[76]. The top 10% of positions with the highest mutation rates were considered outliers[77]. The clustering of DMS-MaPseq signal was performed with DRACO[28].

## SHAPE chemical probing of RNAs

Chemical probing and mutate-and-map experiments were carried out as described previously[78]. In brief, 1.2 pmol RNA was denatured at 95 °C in 50 mM Na-HEPES, pH 8.0, for 3 min, and folded by cooling to room temperature over 20 min, and then adding MgCl$_2$ to a 10 mM concentration. RNA was aliquoted in 15 μl volumes into a 96-well plate and mixed with nuclease-free H$_2$O (control), or chemically modified in the presence of 5 mM 1-methyl-7-nitroisatoic anhydride (1M7)[79], for 10 min at room temperature. Chemical modification was stopped by adding 9.75 μl quench and purification mix (1.53 M NaCl, 1.5 μl washed oligo-dT beads, Ambion), 6.4 nM FAM-labeled, reverse-transcriptase primer (/56-FAM/AAAAAAAAAAAAAAAAAAAAGTTGTTCTTGTTGTTTCTTT), and 2.55 M Na-MES. RNA in each well was purified by bead immobilization on a magnetic rack and two washes with 100 μl 70% ethanol. RNA was then resuspended in 2.5 μl nuclease-free water prior to reverse transcription.

RNA was reverse transcribed from annealed fluorescent primer in a reaction containing 1× First Strand Buffer (Thermo Fisher), 5 mM dithiothreitol, 0.8 mM dNTP mix and 20 U SuperScript III Reverse Transcriptase (Thermo Fisher) at 48 °C for 30 min. RNA was hydrolyzed in the presence of 200 mM NaOH at 95 °C for 3 min, then placed on ice for 3 min and quenched with 1 volume 5 M NaCl, 1 volume 2 M HCl, and 1 volume 3 M sodium acetate. cDNA was purified on magnetic beads, then eluted by incubation for 20 min in 11 μl Formamide-ROX350 mix (1,000 μl Hi-Di Formamide (Thermo Fisher) and 8 μl ROX350 ladder (Thermo Fisher)). Samples were then transferred to a 96-well plate in 'concentrated' form (4 μl sample + 11 μl ROX mix) and 'dilute' form (1 μl sample + 14 μl ROX mix) for saturation correction in downstream analysis. Sample plates were sent to Elim Biopharmaceuticals for analysis by capillary electrophoresis.

## Antisense oligonucleotide infection

ASOs were purchased from Integrated DNA Technologies; the Morpholino ASOs were purchased from Gene Tools LLC (see sequences in Data File S9). A total of 95,000 HEK cells were seeded into the wells of a 24-well cell culture-treated plate in a total volume of 500 μl. At 24 h later, either 1 nmol Morpholino ASO together with 3 μl EndoPorter reagent (Gene Tools LLC), or 6 pmol other ASO were added to each well. LNCaP, MCF-7 and LS174T cells were infected with ASOs using Lonza SE Cell Line 4D-Nucleofector X Kit S (cat. no. V4XC-1032) according to the manufacturer's protocol. At 48 h later, the mCherry and eGFP fluorescence was measured on a BD FACSCelesta Cell Analyzer, or RNA was isolated for RT-qPCR measurement with the Zymo Quick-RNA Microprep isolation kit with in-column DNase treatment per the manufacturer's protocol.

## CRISPRi screen

Reporter screens were conducted using established flow cytometry screen protocols[80] (Horlbeck et al., 2016; Sidrauski et al., 2015). Jurkat cells with previously verified CRISPRi activity were used (Horlbeck et al., 2018). The CRISPRi-v2 (5 sgRNA/TSS, Addgene cat. no. 83969) sgRNA library was transduced into Jurkat cells at a multiplicity of infection of <0.3 (the percentage of blue fluorescent protein (BFP)-positive cells was ~30%). For the flow-based CRISPRi screen with the Jurkat cells, the sgRNA library virus was transfected at an average of 500-fold coverage after transduction (day 0). Puromycin (1 μg ml$^{-1}$) selection for positively transduced cells was performed at 48 h (day 2) and 72 h (day 3) after transduction (day 3). On day 11, cells were collected in PBS and sorted with the BD FACSAria Fusion cell sorter. Cells were gated into the 25% of cells with the highest GFP : mCherry fluorescence intensity ratio, and the 25% of cells with the lowest ratio. The screens were performed with two conditions: cells with a reference RORC element−GFP reporter and a mutated 77-23 RORC element−GFP reporter. Screens were additionally performed in duplicate. After sorting, genomic DNA was collected (Macherey-Nagel Midi Prep kit) and amplified using NEB Next Ultra II Q5 master mix and primers containing TruSeq Indexes for next-generation

sequencing. Sample libraries were prepared and sequenced on a HiSeq 4000. Guides were then quantified with the published ScreenProcessing (https://github.com/mhorlbeck/ScreenProcessing) method and phenotypes generated with an in-house processing pipeline, iAnalyzer (https://github.com/goodarzilab/iAnalyzer). In brief, iAnalyzer relies on fitting a generalized linear model to each gene. Coefficients from this generalized linear model were z-score normalized to the negative control guides and finally the largest coefficients were analyzed as potential hits. For the comparison of gene phenotypes between the two cell lines, the DESeq2 ratio of ratios test was used[57].

## CRISPRi-mediated and small interfering RNA-mediated gene knockdown

Jurkat cells expressing the dCas9–KRAB fusion protein were constructed by lentiviral delivery of pMH0006 (Addgene, cat. no. 135448) and FACS isolation of BFP-positive cells.

Guide RNA sequences for CRISPRi-mediated gene knockdown were cloned into pCRISPRia-v2 (Addgene, cat. no. 84832) via BstXI-BlpI sites. After transduction with sgRNA lentivirus, Jurkat cells were selected with 2 µg ml$^{-1}$ puromycin (Gibco). The fluorescence of eGFP and of mCherry was measured on a BD FACSCelesta Cell Analyzer.

For UPF1 siRNA-mediated knockdown, the TriFECTa DsiRNA Kit from Integrated DNA Technologies (cat. no. hs.Ri.UPF1.13) was used. LNCaP, MCF-7 and LS174T cells were infected with siRNAs using the Lonza SE Cell Line 4D-Nucleofector X Kit S (cat. no. V4XC-1032) according to the manufacturer's protocol. At 48 h later, RNA was collected using the Zymo QuickRNA Microprep isolation kit with in-column DNase treatment as per the manufacturer's protocol.

## Reporter cell line generation

Mutated or reference sequences of RORC 3′UTR were cloned into the dual GFP–mCherry reporter using the MluI-HF and PacI restriction enzymes (NEB) as described above. The reporters were lentivirally delivered to HEK293 and Jurkat cells and analyzed with flow cytometry as described above.

## Drug treatment

Jurkat cells were seeded at a density of $0.25 \times 10^7$ cells per ml. Either the proteasome inhibitors (Carfilzonib or Bortezomib, Cayman Chemical) or negative control (dimethyl sulfoxide, DMSO) were added at the given concentration. After 24 h of incubation, the fluorescence of eGFP and of mCherry was measured on a BD FACSCelesta Cell Analyzer.

MCF-7 cells were treated either with 50 µM NMDI14 (TargetMol), or with DMSO, for 24 h. Afterwards, cells were treated with DMS as describe above and the RNA was collected as described above.

## mRNA stability measurements

Jurkat cells were treated with 10 µg ml$^{-1}$ α-amanitin (Sigma-Aldrich, cat. no. A2263) for 8–9 h prior to total RNA extractions. Total RNA was isolated using the Zymo QuickRNA Microprep isolation kit with in-column DNase treatment as per the manufacturer's protocol. mRNA levels were measured with RT-PCR, using 18S ribosomal RNA (transcribed by RNA Pol I) as the control.

## T-cell isolation, transduction and Th17 cell differentiation

Th17 cells were derived as described previously[34]. Plates were coated with 2 µg ml$^{-1}$ anti-human CD3 (UCSF monoclonal antibody core, clone: OKT-3) and 4 µg ml$^{-1}$ anti-human CD28 (UCSF monoclonal antibody core, clone: 9.3) in PBS with calcium and magnesium for at least 2 h at 37 °C or overnight at 4 °C with the plate wrapped in parafilm. Human CD4+ T cells were isolated from human peripheral blood using the EasySep human CD4+ T cell isolation kit (17952; STEMCELL) and stimulated in ImmunoCult-XF T-cell expansion medium (10981; STEMCELL) supplemented with 10 mM HEPES, 2 mM L-glutamine, 100 µM 2-MOE, 1 mM sodium pyruvate and 10 ng ml$^{-1}$ transforming growth factor-β.

At 24 h after T-cell isolation and initial stimulation on a 96-well plate, 7 µl lentivirus was added to each sample. After 24 h, the media was removed from each sample without disturbing the cells and replaced with 200 µl fresh media. After 48 h, cells were stimulated with 1.2 µM ionomycin, 25 nM propidium monoazide and 6 µg ml$^{-1}$ brefeldin-A, resuspended by pipetting, incubated for 4 h at 37 °C, and collected for analysis. Half of each sample was stained for CD4, FoxP3, interleukin (IL)-13, IL-17A, interferon (IFN)-γ and analyzed on a BD LSRFortessa cell analyzer (see below). The other half of the sample was not stained and was analyzed for the expression of eGFP and mCherry on a BD LSRFortessa cell analyzer.

Cultured human T cells were collected, washed and stained with antibodies against cell surface proteins and transcription factors. Cells were fixed and permeabilized with the eBioscience Foxp3/Transcription Factor Staining Buffer Set or the Transcription Factor Buffer Set (BD Biosciences). Extracellular nonspecific binding was blocked with the anti-CD16/CD32 antibody (clone 2.4G2; UCSF Monoclonal Antibody Core). Intracellular nonspecific binding was blocked with anti-CD16/CD32 antibodies) and 2% normal rat serum. Dead cells were stained with Fixable Viability Dye eFluor 780 (eBioscience) or Zombie Violet Fixable Viability Kit (BioLegend). Cells were stained with the following fluorochrome-conjugated anti-human antibodies: anti-CD4 (Invitrogen, cat. no. 17-0049-42), anti-FOXP3 (eBioscience, cat. no. 25-4777-61), anti-IL-13 (eBioscience, cat. no. 11-7136-41), anti-IL-17A (eBioscience, cat. no. 12-7179-42) and anti-IFNγ (BioLegend, cat. no. 502520). All of the antibodies were used at 1:200 dilution. Samples were analyzed on a BD LSRFortessa cell analyzer. Data were analyzed using FlowJo 10.7.1 and BD FACSDiva v9 software.

## Analysis of capillary electrophoresis data with HiTRACE

Capillary electrophoresis runs from chemical probing and mutate-and-map experiments were analyzed with the HiTRACE MAT-LAB package[81]. Lanes were aligned, bands fitted to Gaussian peaks, background subtracted using the no-modification lane, corrected for signal attenuation, and normalized to the internal hairpin control. The end result of these steps is a numerical array of 'reactivity' values for each RNA nucleotide that can be used as weights in structure prediction.

## UPF1 targeted CLIP-seq

Jurkat cells expressing RORC reporters (reference, 77-GA mutant variant or 116-CCCTAAG mutant variant) were collected and crosslinked by ultraviolet radiation (400 mJ cm$^{-2}$). Cells were then lysed with low salt wash buffer (1x PBS, 0.1% SDS, 0.5% sodium deoxycholate, 0.5% IGEPAL). To probe preferential *UPF1* binding towards different reporters, lysates from 77-GA mutant cells were mixed with lysates from either wild-type or 116-CCCTAAG mutant cells at a 1:1 ratio prior to immunoprecipitation. Samples were then treated with a high dose (1:3,000 RNase A and 1:100 RNase I) and a low dose (1:15,000 RNase A and 1:500 RNase I) of RNase A and RNase I separately and combined after treatment. To immunoprecipitate *UPF1*–RNA complex, a UPF1 antibody (Thermo, cat. no. A301-902A) was incubated with Protein A/G beads (Pierce) first and then incubated with the mixed cell lysates for 2 h at 4 °C. Immunoprecipitated RNA fragments were then dephosphorylated (T4 PNK, NEB), polyadenylated and end-labeled with 3′-azido-3′-dUTP and IRDye 800CW DBCO Infrared Dye (LI-COR) on beads. SDS–PAGE was then performed to separate protein–RNA complexes, and RNA fragments were collected from nitrocellulose membrane by proteinase K digestion. cDNA was then synthesized using Takara smarter small RNA sequencing kit reagents with a custom UMI-oligoDT primer (CAAGCA-GAAGACGGCATACGAGATNNNNNNNNNGTGACTGGAGTTCAGACGT-GTGCTCTTCCGATCTTTTTTTTTTTTTTTTTT). The RORC reporter locus was then amplified with a custom primer (ACACTCTTTCCCTACAC-GACGCTCTTCCGATCTTGGGGTGATCCAAATACCACC) and sequencing libraries were then prepared with SeqAmp DNA Polymerase (Takara). Libraries were then sequenced on an illumina Hiseq 4000 sequencer.

## Reporting summary

Further information on research design is available in the Nature Portfolio Reporting Summary linked to this article.

## Data availability

Sequencing data have been deposited in the Gene Expression Omnibus (GEO accession GSE266070). Cryo-EM density maps have been deposited in EMDB, accession numbers EMD- 42275 (WT Class A), EMD- 42276 (WT Class B), EMD- 42277 (WT Class C), EMD- 42400 (77-GA Class C), EMD- 42401 (77-GA Class A), EMD- 42403 (117-AC Class C) and EMD-42404 (117-AC Class B). Rfam database 14.10 (https://rfam.org/) was used in the study.

## Code availability

SwitchFinder source code is available at https://github.com/goodarzilab/SwitchFinder.

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

## Acknowledgements

The authors thank C. Mathy, A. Natale, M. Imakaev, Y. Gomez, M. Zimanyi, A. Smith and A. Pawluk for helpful discussions. H.G. is an Era of Hope Scholar (W81XWH-2210121) and supported by R01CA240984 and R01CA244634. This work was partly supported by National Institutes of Health (NIH) grants 1R35GM140847 (Y.C.). L.A.G. is funded by an NIH New Innovator Award (DP2 CA239597), a Pew-Stewart Scholars for Cancer Research award and the Goldberg-Benioff Endowed Professorship in Prostate Cancer Translational Biology. Cryo-EM equipment at UCSF is partially supported by NIH grants S10OD020054, S10OD021741 and S10OD026881. Y.C. is an Investigator at Howard Hughes Medical Institute. Sequencing was performed at the UCSF CAT, supported by UCSF PBBR, RRP IMIA and NIH 1S10OD028511-01 grants. A.N. was supported by the DoD PRCRP Horizon Award W81XWH-19-1-0594. L.F. was supported by an NIH training grant T32CA108462-15.

## Author contributions

M.K. and H.G. designed the study. M.K. developed SwitchFinder. A.B. and C.C. designed a docker environment for SwitchFinder. M.K. and A.N. performed the massively parallel reporter assays. M.K., S.Z. and L.F. performed the DMS-MaPseq experiments. M.K. and C.P. performed the SHAPE experiments. D.A. and Y.C. performed the cryo-EM experiments. M.K. performed the mutagenesis experiments. M.K., K.Y. and J.G. performed the antisense oligonucleotide transfection experiments. M.K., S.K.Z. and K.M.A. performed the Th17 differentiation experiments. M.K, A.W. and L.A.G. performed the CRISPRi screens. M.K. performed the CRISPRi knockdown experiments. M.K. and J.Y. performed the proteasome inhibition experiments. S.Z. performed the CLIP-seq experiments. M.K. and H.G. wrote the manuscript with input from all of the authors.

## Competing interests

M.K. and H.G. are inventors on a provisional patent related to this study. L.A.G. has filed patents on CRISPR functional genomics. The other authors have no competing interests.

## Additional information

**Extended data** are available for this paper at https://doi.org/10.1038/s41592-024-02335-1.

**Correspondence and requests for materials** should be addressed to Hani Goodarzi.

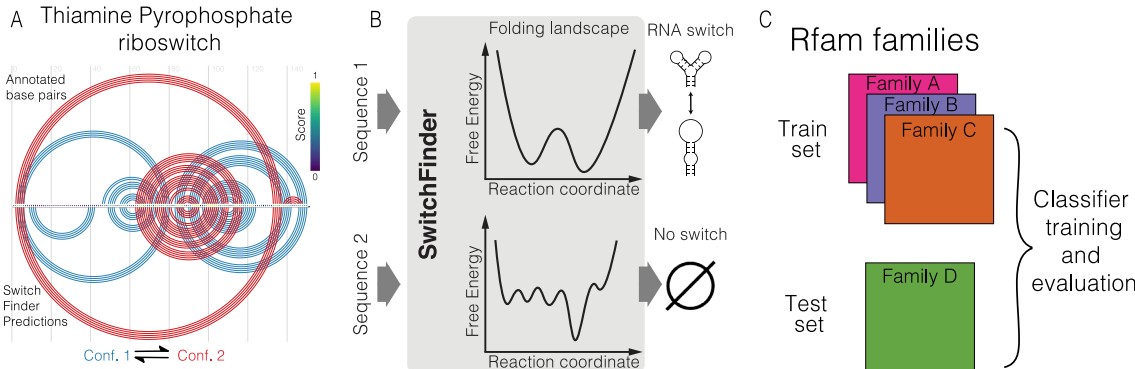

**Extended Data Fig. 1 | SwitchFinder identifies saddles in RNA folding energy landscape. a** Example of SwitchFinder locating the thiamine pyrophosphate RNA switches within the mRNA sequence. Top: arc representation of the RNA base pairs that change between the two conformations of the *E.coli* TPP RNA switch, as in (Barsacchi et al. [10]). The two conformations are shown in red and blue, respectively. Bottom: the two conformations of the RNA switch as predicted by SwitchFinder. Middle: SwitchFinder score reflecting the likelihood of a given nucleotide to be involved in two mutually exclusive base pairings. **b** Scheme of SwitchFinder model. SwitchFinder analyzes RNA folding energy landscape of a given RNA sequence and assigns higher score to the landscapes that demonstrate riboswitch-like features. **c** The set-up for evaluating the ability of a model to find RNA switches from novel families. At the classifier training step, riboswitches from one of the Rfam families get separated into the 'test set', while the model gets trained on the riboswitches from other Rfam families. The test set then is used to evaluate the model performance.

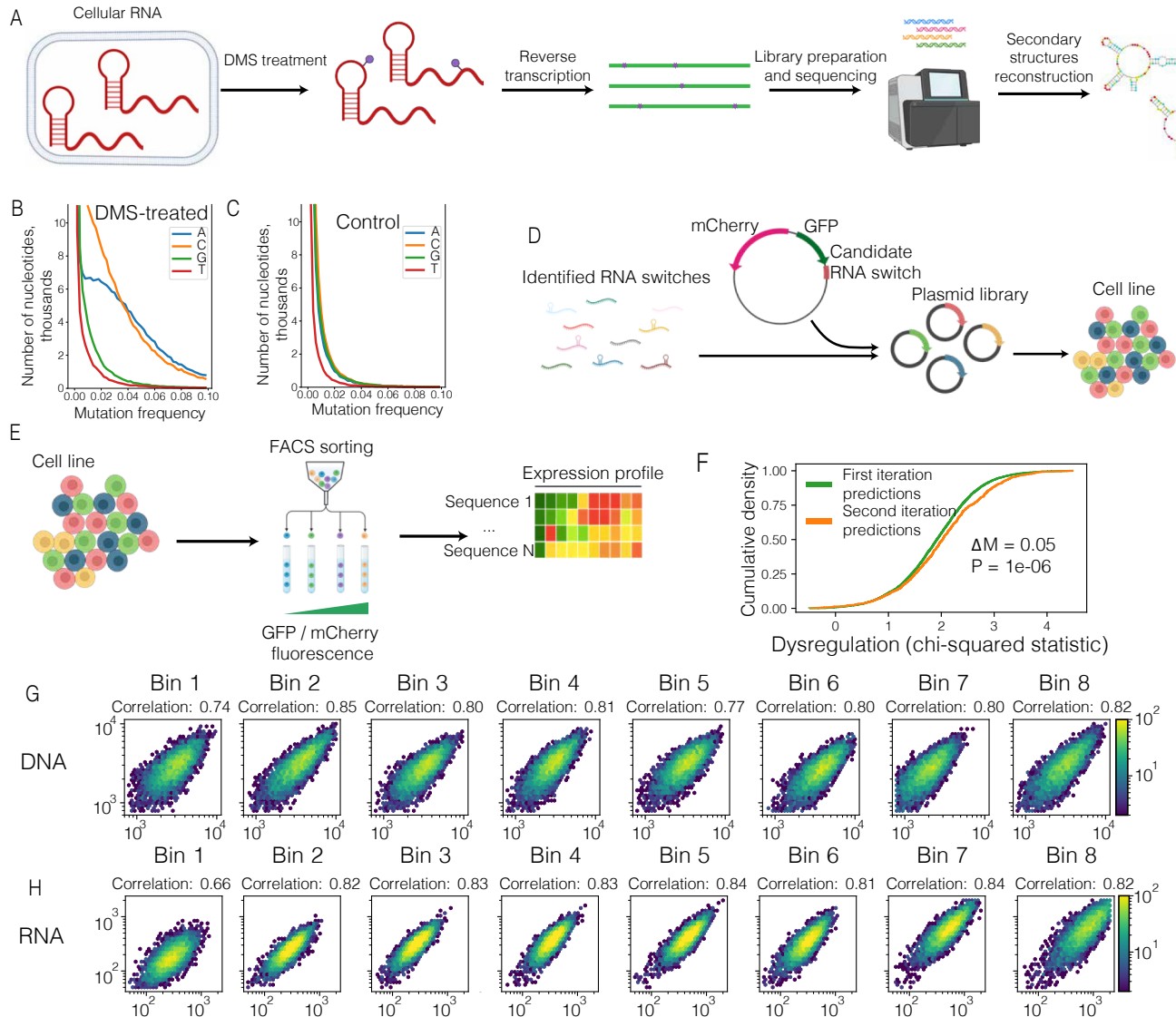

**Extended Data Fig. 2 | Overview of high-throughput screening approaches for improved RNA switch predictions. a** Overview of DMS-MaPseq workflow. Mammalian cells are treated with DMS. DMS-modified nucleotides cause mutations when cDNA is synthesized from RNA templates. The cDNA libraries are sequenced, the DMS-caused mutations are counted, providing the Watson-Crick face accessibility estimates for each A- or C- nucleotide. **b** Cumulative mutation frequency in DMS-treated candidate RNA switches, separated by nucleotide. **c** Cumulative mutation frequency in nontreated candidate RNA switches, separated by nucleotide. **d** Overview of the library generation workflow for Massively Parallel Reporter Assay (MPRA). Sequences of candidate RNA switches are synthesized as DNA oligonucleotides and cloned into a reporter vector into 3′UTR region of a eGFP cDNA. The plasmid library is packaged into lentiviral particles, and used for infecting mammalian cells. The infection is performed

at low MOI (infection rate) to ensure that most cells get only a single plasmid copy. **e** Overview of the MPRA workflow. A population of mammalian cells is separated into bins based on GFP/mCherry fluorescence ratio. In the schematic, cells are colored according to the sequence they carry in the 3′UTR of the GFP reporter. **f** Cumulative density plot of dysregulation values, comparing the candidate RNA switches predicted in first and second (DMS-MaPseq informed) iterations of SwitchFinder. Dysregulation values are estimated using chi-square test for every individual candidate RNA switch across 8 expression bins. Median difference (ΔM) and *P* value (calculated using Mann-Whitney U-test) are shown. **g** Correlations of read counts of gDNA libraries between the biological replicates of massively parallel mutagenesis analysis. **h** Correlations of read counts of RNA libraries between the biological replicates of massively parallel mutagenesis analysis.

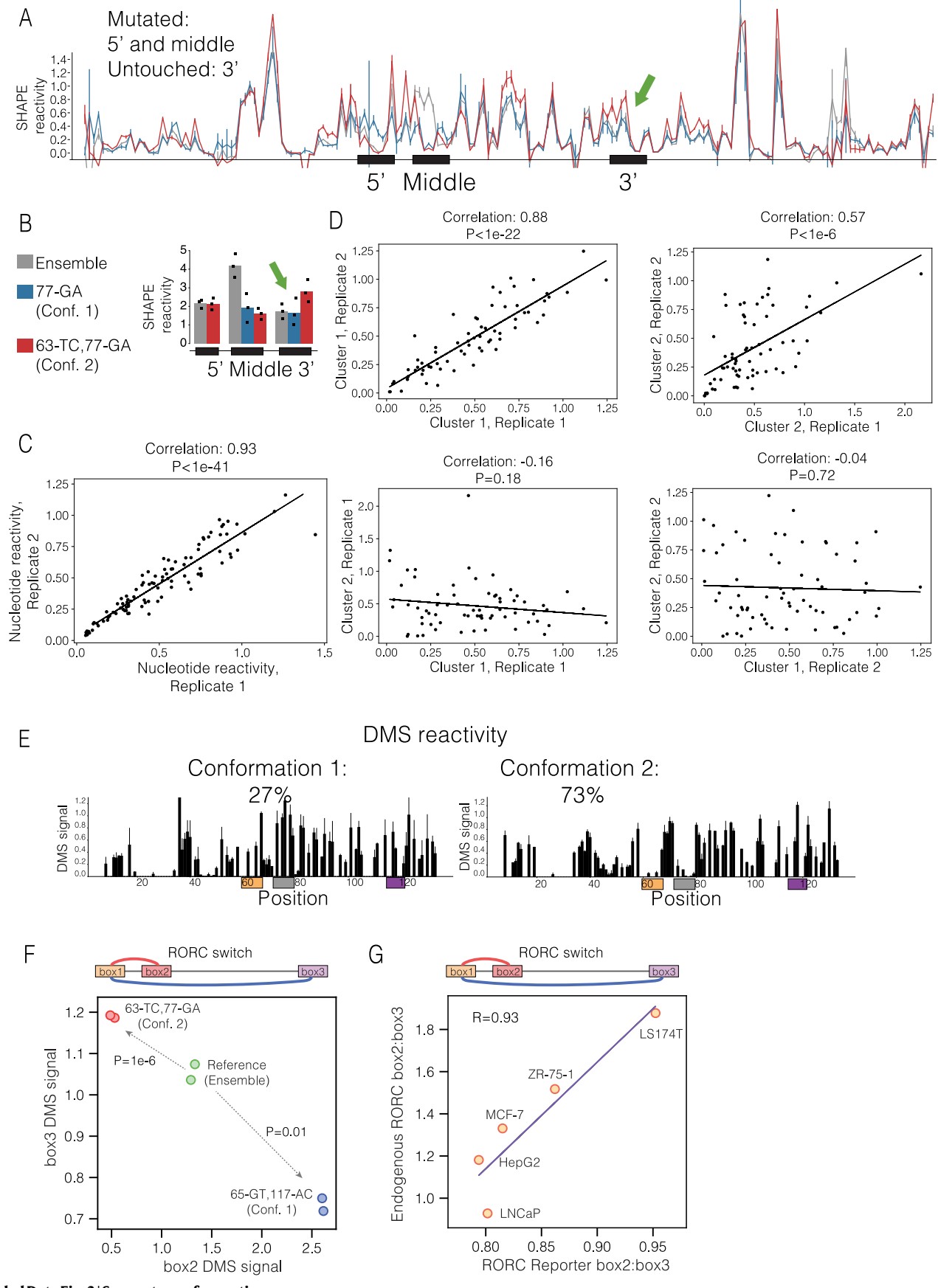

**Extended Data Fig. 3 | See next page for caption.**

**Extended Data Fig. 3 | In vitro SHAPE reactivity of the RORC RNA switch sequence in vitro. a** SHAPE reactivity profiles for the reference sequence and for the mutation–rescue pair of sequences (blue - '77-GA', red - '63-TC,77-GA'). Shown is the average for 3 replicates with the respective error bars (SD). The SHAPE reactivity changes in the nonmutated regions are highlighted in bold arrows. **b** Barplots of cumulative SHAPE reactivity within the switching regions for the reference sequence (in gray) and for the mutation–rescue pair of sequences (blue - '77-GA', red - '63-TC,77-GA'). N replicates = 3. **c** Scatter plot showing the reproducibility of the DMS signal between two replicates. Each dot represents a single nucleotide. Normalized DMS signal is shown on both axes. Correlation and *P* value is determined with Pearson correlation coefficient (P = 1.59-42). **d** Scatter plots showing the reproducibility of the DRACO clusters between replicates (N = 2). Each replicate's reads were clustered with DRACO, the DMS reactivity was calculated for each cluster; the clusters were subsequently matched between replicates. Shown are DMS reactivities

for a given cluster in a given replicate; each dot represents a single nucleotide. Correlation and *P* value is determined with Pearson correlation coefficient. *P* values left to right: 2.60e-23, 3.62e-07, 0.18, 0.73. **e** DMS reactivities of the two clusters identified by the DRACO unsupervised deconvolution algorithm (Morandi et al. [28]). The algorithm was run on two replicates independently, and identified the same clusters in both of them. The ratios of the clusters reported by DRACO are 22% to 78% in replicate 1 and 32% to 68% in replicate 2. The ratio shown is an average between the two replicates. The switching regions are shown in color. **f** The effect of sequence mutations in the 'Box 2' and 'Box 3' regions of RORC element on their reactivity, as measured by DMS-MaPseq in a reporter cell line. *P* values were determined using the two-sided independent T-test. **g** Correlation of relative proportions of the two conformations between the reporter context and the endogenous RORC mRNA. Linear regression is shown with a line. The relative conformations' proportion is defined as the ratio of reactivities of Box 2:Box 3.

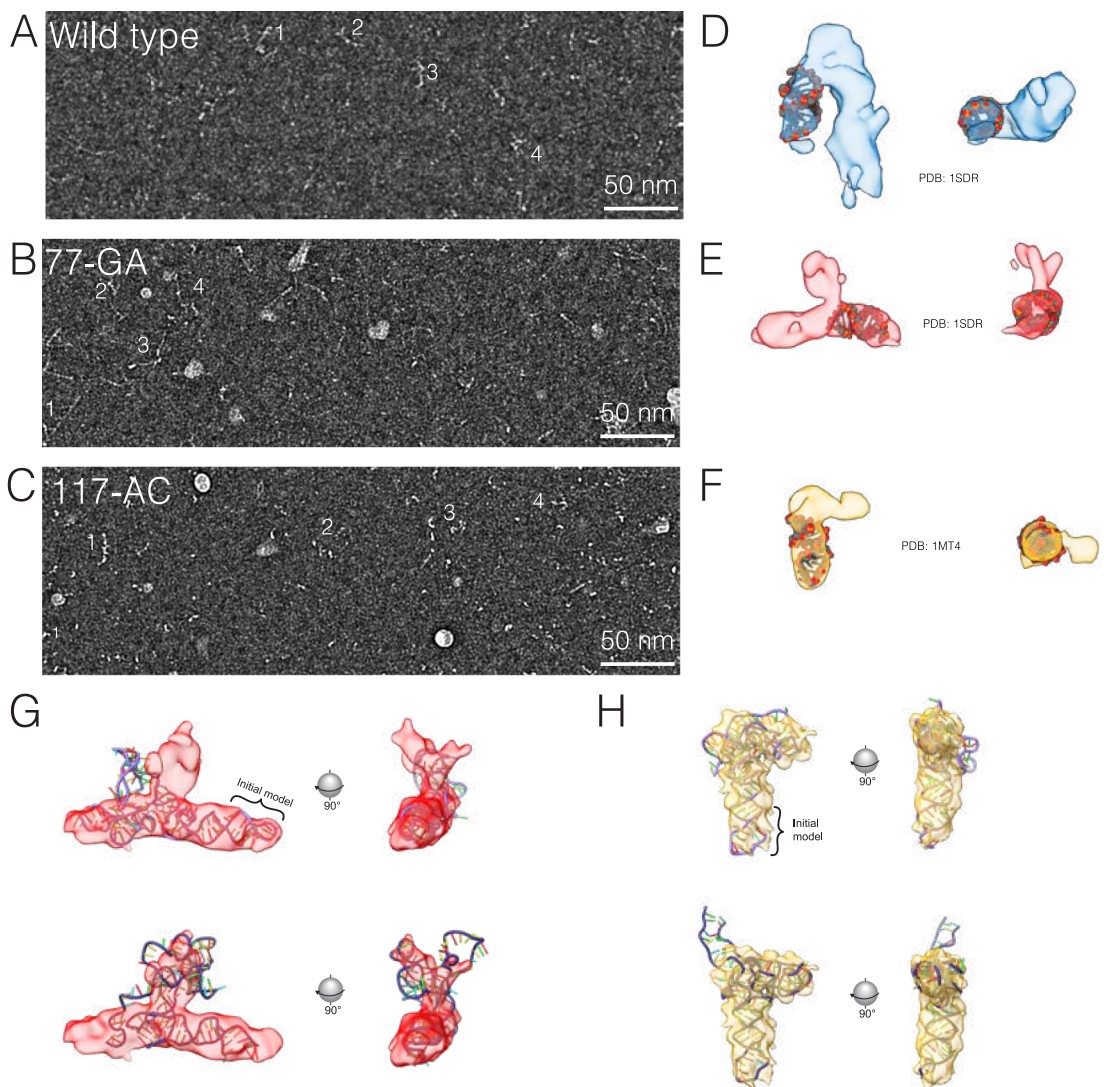

**Extended Data Fig. 4 | Qualitative modeling of cryo-EM data. (a–c)** Source cryo-EM images for the example particles shown in Fig. 4a, with phase-flipping to correct contrast and CTF delocalization. The WT image (A) evinces a greater diversity of particles, while 77-GA (B) appears to contain primarily elongated particles and those of 117-AC (C) seem more compact. The data collection statistic is available in Data file S7. **(d-f)** Cryo-EM 3D classes A, B, and C of the WT RORC RNA overlaid with stereotypical RNA tertiary structures from the PDB including dsRNA B-helix and RNA hairpin. Features representing the major groove and a hairpin are visible in regions of the maps. **(g, h)** Pairs of high-scoring models created by DRRAFTER for WT 3D classes B and C with density overlaid. The pre-positioned, idealized RNA structures used as initial models are indicated by a bracket. Although the individual models are of low-confidence, they demonstrate that the class densities likely represent all or the majority of the RNA molecule.

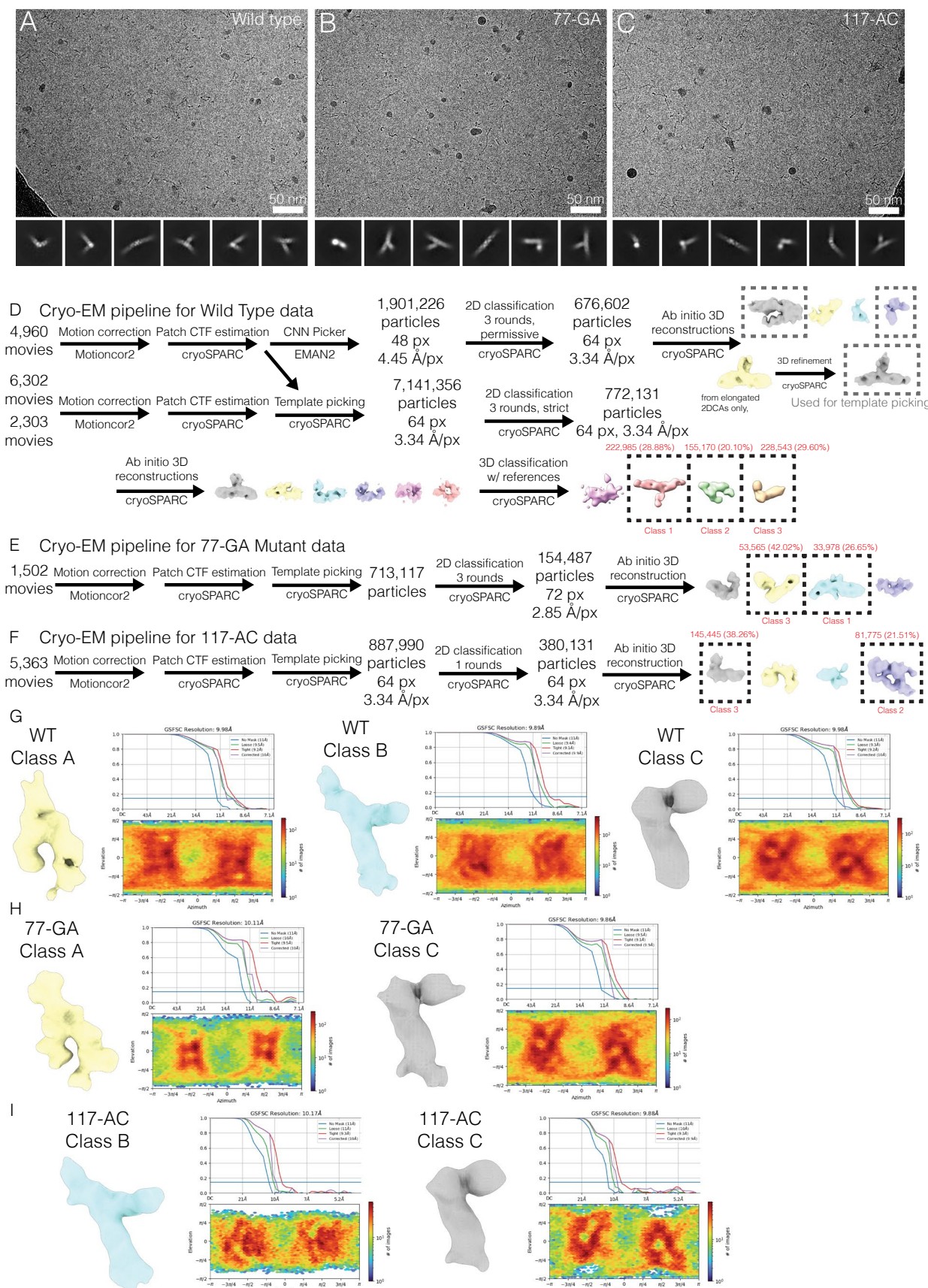

**Extended Data Fig. 5 | See next page for caption.**

**Extended Data Fig. 5 | Cryo-EM image processing and validation.**
(**a-c**) Representative micrographs and 2D class averages for RORC RNA switch WT sequence (A), 77-GA (B) and 117-AC (C). The data collection statistic is available in Data file S7. (**d**) Schematic cryo-EM image processing pipelines for WT RORC RNA. During template picking, templates and micrographs were low-pass filtered to 20 Å. (**e, f**) Schematic cryo-EM image processing pipelines for 77-GA (E), and 117-AC (F) mutants. During template picking, templates and micrographs were low-pass filtered to 20 Å. (**g**) Gold-standard half-map refinement volume, FSC curves, and orientation distribution plot for 3D classes from WT RNA sample. (**h**) Gold-standard half-map refinement volume, FSC curves, and orientation distribution plot for 3D classes from 77-GA sample. (**i**) Gold-standard half-map refinement volume, FSC curves, and orientation distribution plot for 3D classes from 117-AC sample.

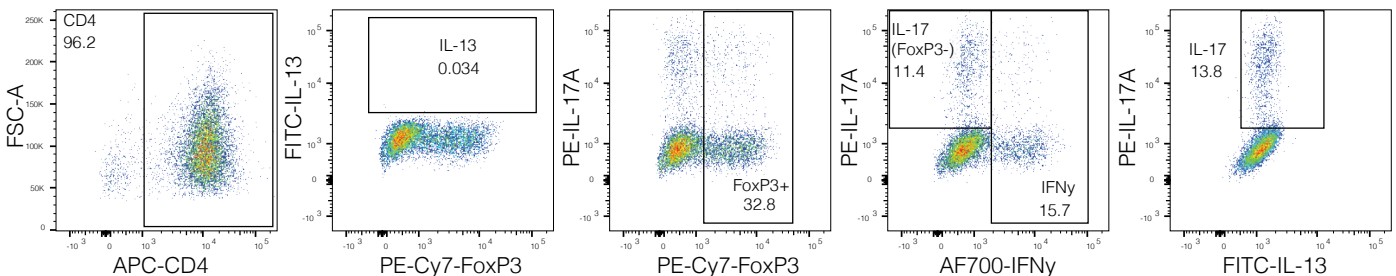

**Extended Data Fig. 6 | Differentiation of Th17 cells from primary human CD4+ cells.** Representative fluorescence-activated cell sorting plots of human primary Th17 cells, infected with RORC RNA switch 3′UTR reporter. On the day 5 of differentiation, each sample was split in half; one half was analyzed for mCherry and GFP expression (shown in Fig. 5c), the other half was stained for the expression of CD4, FoxP3, IL-13, IL-17A, IFN-gamma. The cells expressing a given marker are highlighted with a frame and a fraction of the parental cellular population is given. Each sample was analyzed in 4 replicates; a single representative replicate is displayed. CD4 is a marker for T-helper cells, including Th17. FoxP3 is typically associated with regulatory T cells, contrasting the pro-inflammatory role of Th17 cells. IL-13 and IL-17A are cytokines indicative of Th2 and Th17 cell activity, respectively, with IL-17A being a key marker for Th17 cell identity. IFN-gamma is a signature cytokine of Th1 cells.

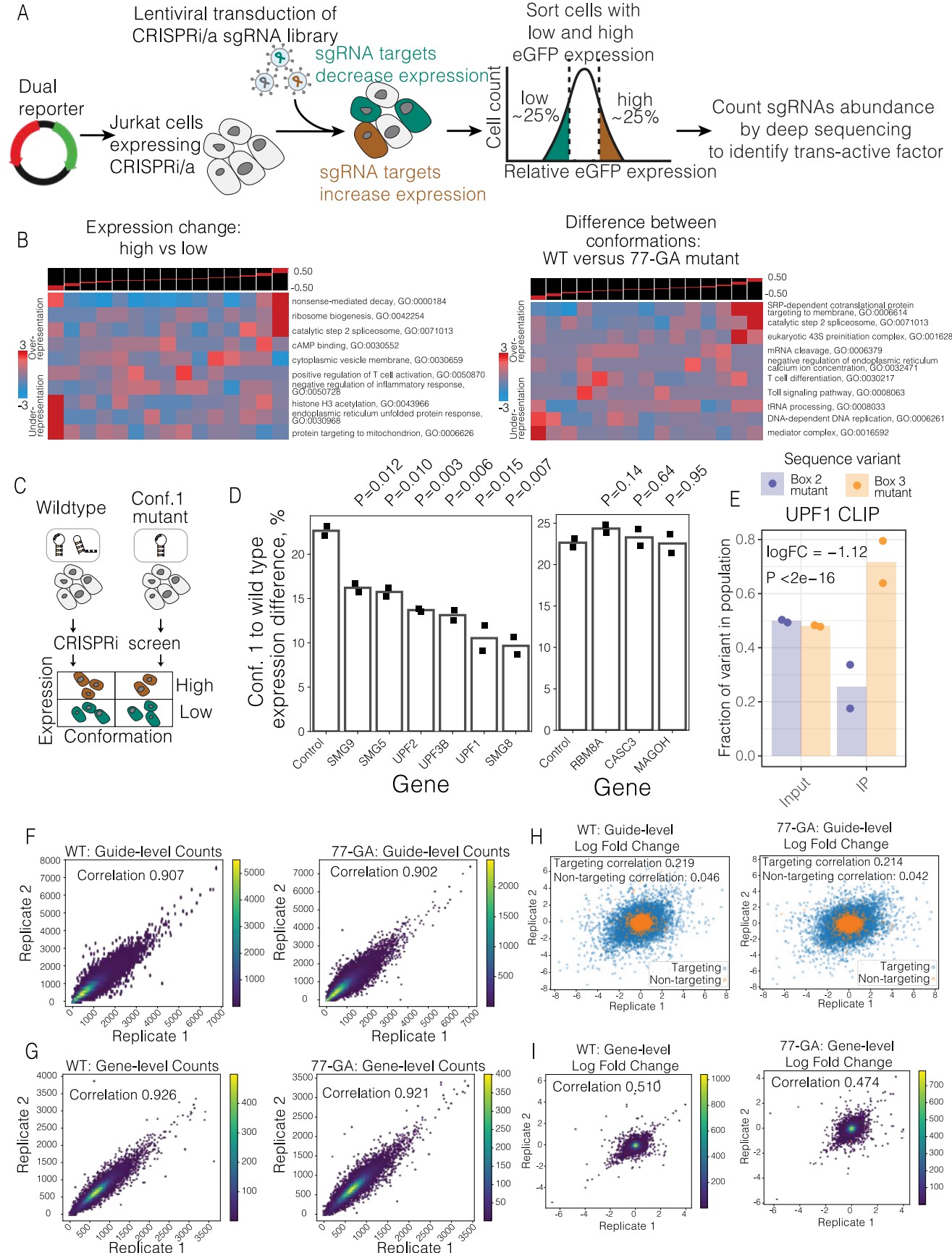

**Extended Data Fig. 7 | See next page for caption.**

**Extended Data Fig. 7 | CRISPRi screen highlights the pathways acting downstream of the RORC RNA switch. a** Overview of the flow cytometry-based CRISPRi screen workflow. **b** Gene set enrichment analysis of the data depicted in Fig. 6a (left) and Fig. 6b (right). The genes were distributed into equally populated bins based on their comparative abundance between high expression and low expression quartiles (left), or based on their comparative phenotype in the CRISPRi screens performed in WT or 77-GA mutant backgrounds (right). Then the enrichment of a given gene set was calculated in each bin using iPAGE, a mutual information-based algorithm (Goodarzi et al. 2009). **c** Experiment design table. **d** The effect of knockdown of SURF and EJC complex member proteins on the expression change upon the conformation equilibrium shift. The individual genes were knocked down using the CRISPRi system in both WT and 77-GA mutant cell lines, then the change of reporter gene expression was measured by flow cytometry (N replicates = 2). The bar plots demonstrate the expression ratios of WT to 77-GA mutation cell lines. **e** The bar plots demonstrate the fractions of reads carrying the Box 2 (77-GA) mutant sequence or Box 3 (116-CCCTAAG) mutant sequence in UPF1 cross-linking and immunoprecipitation (CLIP) library. Box 2 mutant favors conformation 1, Box 3 mutant favors conformation 2. Left: input RNA libraries, extracted from the

Box 3 and Box 2 mutant-expressing Jurkat cells, mixed at 1:1 ratio. Right: libraries after anti-*UPF1* immunoprecipitation. *P* value was calculated using Translation Efficiency Ratio test as in (Navickas et al. [58]). N replicates = 2. **f** Density plots showing the correlation of sgRNA counts between the replicates of the CRISPRi screens performed in the WT (left) and 77-GA mutant (right) backgrounds. **g** Density plots showing the correlation of gene counts between the replicates of the CRISPRi screens performed in the WT (left) and 77-GA mutant (right) backgrounds. The counts of all the sgRNAs targeting a given gene are pooled and reported as a single number (N = 5 sgRNAs per gene). **h** Scatter plots showing the correlation of sgRNA phenotypes between the replicates of the CRISPRi screens performed in the WT (left) and 77-GA mutant (right) backgrounds. Logarithmic fold changes between the sgRNA abundance 'high' and 'low' expression bins are shown on both axes. Nontargeting sgRNAs are shown in orange; all the other sgRNAs are shown in blue. The correlation values are reported separately for nontargeting and targeting sgRNAs. **i** Density plots showing the correlation of gene phenotypes between the replicates of the CRISPRi screens performed in the WT (left) and 77-GA mutant (right) backgrounds. Logarithmic fold changes between the abundance of sgRNAs targeting a given gene in 'high' and 'low' expression bins are shown on both axes.

# Reporting Summary

## Statistics

For all statistical analyses, confirm that the following items are present in the figure legend, table legend, main text, or Methods section.

| n/a | Confirmed | |
|---|---|---|
| ☐ | ☒ | The exact sample size (*n*) for each experimental group/condition, given as a discrete number and unit of measurement |
| ☐ | ☒ | A statement on whether measurements were taken from distinct samples or whether the same sample was measured repeatedly |
| ☐ | ☒ | The statistical test(s) used AND whether they are one- or two-sided<br>*Only common tests should be described solely by name; describe more complex techniques in the Methods section.* |
| ☐ | ☒ | A description of all covariates tested |
| ☐ | ☒ | A description of any assumptions or corrections, such as tests of normality and adjustment for multiple comparisons |
| ☐ | ☒ | A full description of the statistical parameters including central tendency (e.g. means) or other basic estimates (e.g. regression coefficient) AND variation (e.g. standard deviation) or associated estimates of uncertainty (e.g. confidence intervals) |
| ☐ | ☒ | For null hypothesis testing, the test statistic (e.g. *F*, *t*, *r*) with confidence intervals, effect sizes, degrees of freedom and *P* value noted<br>*Give P values as exact values whenever suitable.* |
| ☒ | ☐ | For Bayesian analysis, information on the choice of priors and Markov chain Monte Carlo settings |
| ☒ | ☐ | For hierarchical and complex designs, identification of the appropriate level for tests and full reporting of outcomes |
| ☐ | ☒ | Estimates of effect sizes (e.g. Cohen's *d*, Pearson's *r*), indicating how they were calculated |

*Our web collection on statistics for biologists contains articles on many of the points above.*

## Software and code

Policy information about availability of computer code

| Data collection | The code is at https://github.com/goodarzilab/SwitchFinder |
|---|---|
| Data analysis | The code is at https://github.com/goodarzilab/SwitchFinder. The packages used are: UCSF MotionCor2, CTFFIND4, EMAN2 v2.99.47, cryoSPARC v4.4.1, UCSF ChimeraX 1.7.1, cutadapt 4.9, UMI tools 1.1.5, bwa 0.7.18, featureCounts v1.6.2, MPRAnalyze 1.20.0, Pear v0.9.6, CTK 2023.07, DRACO v1.1, HiTRACE, FlowJo v10, FACSDiva v9.0 |

For manuscripts utilizing custom algorithms or software that are central to the research but not yet described in published literature, software must be made available to editors and reviewers. We strongly encourage code deposition in a community repository (e.g. GitHub). See the Nature Portfolio guidelines for submitting code & software for further information.

## Data

Policy information about availability of data

All manuscripts must include a data availability statement. This statement should provide the following information, where applicable:
- Accession codes, unique identifiers, or web links for publicly available datasets
- A description of any restrictions on data availability
- For clinical datasets or third party data, please ensure that the statement adheres to our policy

Sequencing data has been deposited in the Gene Expression Omnibus (GEO accession GSE266070). Cryo-EM density maps have been deposited in EMDB, accession

numbers EMD-42275 (WT Class A), EMD-42276 (WT Class B), EMD-42277 (WT Class C), EMD-42400 (77-GA Class C), EMD-42401 (77-GA Class A), EMD-42403 (117-AC Class C), and EMD-42404 (117-AC Class B). RFAM database 14.10 (https://rfam.org/) was used in the study.

## Human research participants

Policy information about studies involving human research participants and Sex and Gender in Research.

| | |
|---|---|
| Reporting on sex and gender | PBMCs purchased from StemCell are produced from both male and female donors, although specific information on donor gender was not collected for the purpose of this research. |
| Population characteristics | Peripheral blood mononuclear cells (PBMCs) from anonymous healthy human donors were purchased fresh from StemCell technologies. |
| Recruitment | PBMCs were purchased from a commercial source: StemCell. |
| Ethics oversight | PBMCs from anonymous donors were purchased from StemCell Technologies, which collected PBMCs from healthy donors under protocols approved by the StemCell Technologies IRB. |

Note that full information on the approval of the study protocol must also be provided in the manuscript.

# Field-specific reporting

Please select the one below that is the best fit for your research. If you are not sure, read the appropriate sections before making your selection.

☒ Life sciences          ☐ Behavioural & social sciences          ☐ Ecological, evolutionary & environmental sciences

For a reference copy of the document with all sections, see nature.com/documents/nr-reporting-summary-flat.pdf

# Life sciences study design

All studies must disclose on these points even when the disclosure is negative.

| | |
|---|---|
| Sample size | For all the high-throughput screens, namely the (i) functional screen, the (ii) structure screen, the (iii) massively parallel mutagenesis screen, and the (iv) CRISPRi screen, the cellular population was maintained at all times with at least 500X coverage of the library size. For CryoEM experiments, the number of particles collected is reported in the Supplementary Tables. For all flow cytometry experiments, data were collected from >10.000 cells to ensure the sufficient coverage. |
| Data exclusions | No data was excluded from our analyses |
| Replication | The high-throughput screens, namely the (i) functional screen, the (ii) structure screen, the (iii) massively parallel mutagenesis screen, and the (iv) CRISPRi screen were performed in 2 replicates.<br>DMS-MaP-seq experiments were performed in 2 replicates.<br>SHAPE probing experiments were performed in 3 replicates.<br>CLIP-seq experiments were performed in 2 replicates.<br>Th17 differentiation experiments were performed in 4 replicates.<br>CRISPRi knockdown experiments were performed in 2 replicates.<br>Proteasome inhibition experiments were performed in 3 replicates.<br>qPCR measurements were performed in 2-3 replicates, depending on the experiment.<br>The replicates showed consistent results for all the experiments. |
| Randomization | The RNA samples were randomly allocated for NMDI14 or DMS treatment, and/or for Bortezomib or Carfilzomib treatment. |
| Blinding | Where possible, the RT-qPCR step was performed by a different researcher. |

# Reporting for specific materials, systems and methods

We require information from authors about some types of materials, experimental systems and methods used in many studies. Here, indicate whether each material, system or method listed is relevant to your study. If you are not sure if a list item applies to your research, read the appropriate section before selecting a response.

## Materials & experimental systems

| n/a | Involved in the study |
|---|---|
| ☐ | ☒ Antibodies |
| ☐ | ☒ Eukaryotic cell lines |
| ☒ | ☐ Palaeontology and archaeology |
| ☒ | ☐ Animals and other organisms |
| ☒ | ☐ Clinical data |
| ☒ | ☐ Dual use research of concern |

## Methods

| n/a | Involved in the study |
|---|---|
| ☒ | ☐ ChIP-seq |
| ☐ | ☒ Flow cytometry |
| ☒ | ☐ MRI-based neuroimaging |

# Antibodies

| Antibodies used | The antibody information is described in the Material and Methods section of the manuscript.<br>primary<br>anti-CD4 (Invitrogen 17-0049-42)<br>anti-FOXP3 (eBioscience 25-4777-61)<br>anti-IL-13 (eBioscience 11-7136-41)<br>anti-IL-17A (eBioscience 12-7179-42)<br>anti-IFNγ (BioLegend 502520)<br>anti-CD16/CD32 antibody (clone 2.4G2; UCSF Monoclonal Antibody Core AM004)<br>anti-human CD3 (UCSF monoclonal antibody core, clone: OKT-3, AH003)<br>anti-human CD28 (UCSF monoclonal antibody core, clone: 9.3, AH002) |
|---|---|
| Validation | The antibody information is described in the Material and Methods section of the manuscript.<br>primary<br>anti-CD4 (Invitrogen 17-0049-42)<br>Applications Tested: This RPA-T4 antibody has been pre-titrated and tested by flow cytometric analysis of normal human peripheral blood cells. his can be used at 5 μL (0.5 μg) per test. A test is defined as the amount (μg) of antibody that will stain a cell sample in a final volume of 100 μL. Cell number should be determined empirically but can range from 10^5 to 10^8 cells/test.<br><br>anti-FOXP3 (eBioscience 25-4777-61)<br>Applications Tested: This 236A/E7 antibody has been pre-titrated and tested by intracellular staining and flow cytometric analysis of normal human peripheral blood cells using the Foxp3/Transcription Factor Staining Buffer Set (cat. 00-5523) and protocol. Please refer to Best Protocols: Protocol B: One step protocol for (nuclear) intracellular proteins. This can be used at 5 μL (0.125 μg) per test. A test is defined as the amount (μg) of antibody that will stain a cell sample in a final volume of 100 μL. Cell number should be determined empirically but can range from 10^5 to 10^8 cells/test.<br><br>anti-IL-13 (eBioscience 11-7136-41)<br>Applications Tested: This 85BRD antibody has been pre-titrated and tested by intracellular staining followed by flow cytometric analysis of stimulated normal human peripheral blood cells using the Intracellular Fixation & Permeabilization Buffer Set (cat. 88-8824) and protocol. Please refer to Best Protocols: Protocol A: Two step protocol for (cytoplasmic) intracellular proteins located under the Resources Tab online. This can be used at 5 μL (0.25 μg) per test. A test is defined as the amount (μg) of antibody that will stain a cell sample in a final volume of 100 μL. Cell number should be determined empirically but can range from 10^5 to 10^8 cells/test.<br><br>anti-IL-17A (eBioscience 12-7179-42)<br>Applications Tested: This eBio64DEC17 antibody has been pre-titrated and tested by intracellular staining and flow cytometric analysis of stimulated normal human peripheral blood cells. This can be used at 5 μL (0.25 μg) per test. A test is defined as the amount (μg) of antibody that will stain a cell sample in a final volume of 100 μL. Cell number should be determined empirically but can range from 10^5 to 10^8 cells/test.<br><br>anti-IFNγ (BioLegend 502520) was tested at https://www.biolegend.com/en-us/products/purified-anti-human-ifn-gamma-antibody-1537?GroupID=BLG2229<br><br>All the antibodies purchased from UCSF Monoclonal Antibody Core were validated by UCSF Monoclonal Antibody Core . |

# Eukaryotic cell lines

Policy information about cell lines and Sex and Gender in Research

| Cell line source(s) | HEK293 and Jurkat cells were purchased from ATCC |
|---|---|
| Authentication | None of the cell lines were authenticated with STR. |
| Mycoplasma contamination | Cell lines were tested regularly for mycoplasma contamination. No mycoplasma contamination was detected. |
| Commonly misidentified lines<br>(See ICLAC register) | No commonly misidentified samples were used in this study |

# Flow Cytometry

## Plots

Confirm that:

☒ The axis labels state the marker and fluorochrome used (e.g. CD4-FITC).

☒ The axis scales are clearly visible. Include numbers along axes only for bottom left plot of group (a 'group' is an analysis of identical markers).

☒ All plots are contour plots with outliers or pseudocolor plots.

☒ A numerical value for number of cells or percentage (with statistics) is provided.

## Methodology

| | |
|---|---|
| Sample preparation | For intracellular stains, T cells were fixed and permeabilized with the eBioscience™ Foxp3 / Transcription Factor Staining Buffer Set or the Transcription Factor Buffer Set (BD Biosciences). Extracellular nonspecific binding was blocked with the anti-CD16/CD32 antibody. Intracellular nonspecific binding was blocked with anti-CD16/CD32 Abs and 2% normal rat serum. Finally, up to 0.5 million T cells from culture were washed with PBS + 1% FBS. For HEK293 cells, cells were washed with PBS once, incubated with Trypsin for 10 minutes, then detached from the plate, resuspended in PBS + 1% FBS and strained through industrial mesh with a pore size of 90 uM (ELKO filtering). For Jurkat cells, cells were resuspended in PBS + 1% FBS. For all experiments, known negatives served as gating controls. |
| Instrument | BD FACSCelesta, BD FACSaria II, BD LSRFortessa |
| Software | FlowJo 10.7.1 and BD FACSDiva v9 |
| Cell population abundance | All sorts were end-point sorts and not for subsequent culture. |
| Gating strategy | For all flow cytometry data, viable cells were gated by FSC-A/SSC-A (as well as live/dead cell markers for some experiments), and singlets by FSC-A/FSC-H. Positive populations were determined by unstained (in case of T cells) or non-transduced (in case of Jurkat or HEK293) samples. Gating strategy is shown in the supplementary data. |

☒ Tick this box to confirm that a figure exemplifying the gating strategy is provided in the Supplementary Information.

