## [Peer Review File · Nature Methods]

Peer Review Information

Manuscript Title: A systematic search for RNA structural switches across the human transcriptome

Corresponding author name(s): Hani Goodarzi

Editorial Notes: None

Reviewer Comments & Decisions:

Decision Letter, initial version:

Dear Hani,

Your Article, "A systematic search for RNA structural switches across the human transcriptome", has now been seen by three reviewers. As you will see from their comments below, although the reviewers find your work of considerable potential interest, they have raised a number of concerns. We are interested in the possibility of publishing your paper in Nature Methods, but would like to consider your response to these concerns before we reach a final decision on publication.

We therefore invite you to revise your manuscript to address these concerns. Before you do, we wanted to offer some guidance on the revision. When revising, please make sure the method is the star of the paper and is fully described, validated, and benchmarked as requested by the reviewers. We also need the software to be provided in an easily usable form.

In terms of the RORC study, the referees did not find your cryoEM data convincing. This should either be greatly strengthened or removed. In addition, they had concerns about the proposed NMD mechanism. We think this could be addressed by toning down some claims and suggesting a speculative mechanism instead of further experimental pursuit, but we leave this to you if you want to pin it down for this paper. We do think that referee 1's concern that the switch should be explored for function in a more native context is a fair one.

* include a point-by-point response to the reviewers and to any editorial suggestions

- * please underline/highlight any additions to the text or areas with other significant changes to facilitate review of the revised manuscript
- * address the points listed described below to conform to our open science requirements
- * ensure it complies with our general format requirements as set out in our guide to authors at www.nature.com/naturemethods
- * resubmit all the necessary files electronically by using the link below to access your home page

[Redacted]

We hope to receive your revised paper within three months. If you cannot send it within this time, please let us know. In this event, we will still be happy to reconsider your paper at a later date so long as nothing similar has been accepted for publication at Nature Methods or published elsewhere.

OPEN SCIENCE REQUIREMENTS

REPORTING SUMMARY AND EDITORIAL POLICY CHECKLISTS

IMAGE INTEGRITY

When submitting the revised version of your manuscript, please pay close attention to our Digital Image Integrity Guidelines and to the following points below:

DATA AVAILABILITY

All novel DNA and RNA sequencing data, protein sequences, genetic polymorphisms, linked genotype and phenotype data, gene expression data, macromolecular structures, and proteomics data must be deposited in a publicly accessible database, and accession codes and associated hyperlinks must be provided in the "Data Availability" section.

Please include a "Data availability" subsection in the Online Methods. This section should inform readers about the availability of the data used to support the conclusions of your study, including accession codes to public repositories, references to source data that may be published alongside the paper, unique identifiers such as URLs to data repository entries, or data set DOIs, and any other statement about data availability. At a minimum, you should include the following statement: "The data that support the findings of this study are available from the corresponding author upon request", describing which data is available upon request and mentioning any restrictions on availability. If DOIs are provided, please include these in the Reference list (authors, title, publisher (repository name), identifier, year). For more guidance on how to write this section please see:

<http://www.nature.com/authors/policies/data/data-availability-statements-data-citations.pdf>

CODE AVAILABILITY

Please include a "Code Availability" subsection in the Online Methods which details how your custom code is made available. Only in rare cases (where code is not central to the main conclusions of the paper) is the statement "available upon request" allowed (and reasons should be specified).

MATERIALS AVAILABILITY

ORCID

Nature Methods is committed to improving transparency in authorship. As part of our efforts in this direction, we are now requesting that all authors identified as 'corresponding author' on published papers create and link their Open Researcher and Contributor Identifier (ORCID) with their account on the Manuscript Tracking System (MTS), prior to acceptance. This applies to primary research papers only. ORCID helps the scientific community achieve unambiguous attribution of all scholarly contributions. You can create and link your ORCID from the home page of the MTS by clicking on 'Modify my Springer Nature account'. For more information please visit please visit www.springernature.com/orcid.

Sincerely,
Rita

Rita Strack, Ph.D.
Senior Editor
Nature Methods

Reviewers' Comments:

Reviewer #1:

Remarks to the Author:

The manuscript entitled „A systematic search for RNA structural switches across the human transcriptome” by Khoroshkin et al. describes a pipeline combining bioinformatics and experimental methods to identify mRNA regions that can result in alternative structural conformations and alter gene expression. Specifically, RNA structures predicted in silico are tested for experimental support by probing methods in a high-throughput manner. The most promising candidates are then tested for their potential to alter expression of a reporter gene in a structure-dependent manner. From the approximately 250 high-confidence candidates identified in this work, the top potential switch derived from the 3' UTR of the RORC transcript was further characterized. These experiments confirmed that this RNA sequence can form two alternative structures that are linked to different expression levels in a reporter gene context. Furthermore, the authors provided evidence that the lower expression observed for one of the conformations is caused by its targeting for degradation via the RNA surveillance system nonsense-mediated decay (NMD).

I think this is an interesting study using high-throughput approaches to identify novel structured mRNA motifs with potential functions in gene regulation. The strategy is overall clearly described and evidence is provided that the candidates can form alternative structures which are linked to altered expression, at least in a reporter context. However, I don't think that the study provides compelling evidence that these motifs are indeed switches, i.e., that the proportions of alternative conformations can change in cells and thereby alter gene expression. What the authors have identified are RNA regions that can fold into different conformations, which can affect gene expression differently. Both findings are not surprising from my point of view, as RNA is well known to adopt alternative folds and many ways how these folds can interfere with gene expression, in particular with respect to the rather weak changes as seen for the RORC motif, can be imagined. So, I think the authors have identified switch candidates, but more experiments would be needed to confirm that switching actually takes place in vivo and has an impact on gene expression. Furthermore, their functional experiments on the RORC candidate are restricted to the reporter context. To show that this motif is biologically relevant, the authors would need to analyse the motif's impact on expression of the endogenous gene.

Major comments:

1) This manuscript aims at identifying novel RNA switches similar to metabolite-sensing riboswitches or protein-interacting RNA folds such as the VEGFA switch. Demonstrating that an RNA has the potential to form different RNA structures and can regulate expression of a reporter construct in a conformation-specific manner is not sufficient to establish it as a switch. From my point of view, further experimental data would be needed to show that the relative proportions of alternative conformations change between conditions and that this is linked to altered gene expression.

2) It doesn't become clear why the authors focused in their search on 3' UTRs, given what we know about other RNA switches. Riboswitches are widespread in bacteria, whereas in eukaryotes so far only

TPP-sensing riboswitches have been characterized. These eukaryotic riboswitches are present in filamentous fungi, green algae, and plants, where they are located in introns and control splicing decisions. As introns can house regulatory elements including structured RNAs without affecting the open reading frame, I think these regions would be a logical choice to look for novel structured RNA motifs in (other) eukaryotes. Any reference to these known eukaryotic riboswitches is missing in this manuscript. Why didn't they analyse intronic sequences? When testing their pipeline (SFig. 1B), why did they use only bacterial riboswitches? Testing TPP riboswitches in more complex eukaryotic genomes would be more revealing with respect to the aims of this study. How did they decide to analyse 3' UTRs in fragments of 186 nt length? The current pipeline can only detect RNA elements that affect gene expression independent from other elements outside of this rather small window, and therefore would fail to detect, at least at the step of functional characterisation, elements such as the eukaryotic TPP riboswitches.

3) The analysis of the RORC motif is limited to the reporter context. The authors provide evidence for the existence of alternative conformations and their effect on transcript levels in a reporter context. However, what would be observed in the natural gene context? Did the authors also perform reporter assays with the complete 3' UTR? Furthermore, the effect of mutations and conformations for endogenous RORC would need to be analysed to refer to this element as a regulatory switch. Based on the presented data, it remains open if the weak differences seen in transcript levels for the two conformations in the reporter context would also cause a change in RORC transcripts, when the element is in its natural exon/intron context and can also get involved in base pairing with more distant regions. As also mentioned before, the authors would need to show that the conformational distribution of RORC 3' UTRs differs between certain conditions *in vivo* and that this is linked to altered RORC levels. The assays involving ASOs and proteasomal inhibition could also be used to test the relevance of the structure *in vivo*. Again, the corresponding assays in this manuscript are restricted to the reporter context. Finally, it does not become clear how distinct this motif is from the other ~250 candidates, as any information on these is missing.

4) To test for the presence of the alternative conformations *in vivo*, the authors performed DMS-MaPseq of the RORC switch followed by a computational analysis (DRACO). To validate this method, the same type of analysis should be performed for the mutant set shown in Fig. 3C. It would be expected that 117-AC has an increased proportion of conformation 2, and that 65-GT, 117-AC gives a similar output as the wildtype.

5) The link between RORC turnover and NMD remains unclear. The authors provide evidence that NMD affects the turnover of reporter transcripts containing the RORC element in a conformation-dependent manner. According to these data, NMD inhibition would then also be expected to affect endogenous RORC levels. Why didn't the authors test this? Their model in Fig. 7 suggests that interaction with UPF1 is directly dependent on the RNA conformation and they refer to a previous study by Fischer et al. reporting structure-mediated RNA decay (SRD) by UPF1. However, SRD was previously reported to be independent of other NMD factors, while here in this study besides UPF1 several other NMD factors were found to regulate the expression of the reporter containing the RORC motif. Moreover, it seems more likely that UPF1 may be recruited by other factors to this specific RNA conformation, rather than it directly recognizes a specific structure (as indicated in Fig. 7).

6) Data presentation and analysis lack clarity at several places. Cryo-EM is performed to provide further evidence for the existence of the alternative conformations. As it is currently presented, I think it's very difficult for the reader to make any conclusion on this. Fig. 4A shows segments of EM

micrographs, for which even after zooming in the numeric labels and structures are hardly visible. SFig. 4 shows more details of the corresponding analyses. What would be needed from my point of view is a compilation of the alternative structures that were combined into the structural classes and conformations shown in Fig. 4, including individual counts. Fig. 5A-D are examples, where the data analysis is not sufficiently described. What are the boxes, whiskers and lines (maybe mean values)? And was the statistical test done on the mean value? I didn't find any supplemental material with these kind of details (which should be provided in the legend as well). I also didn't have access to a supplemental table and other supplemental data files mentioned in the supplement section.

Minor points:

- 1) The authors make the following statement: "The two oldest groups of RNA-based regulatory mechanisms are ribozymes (catalytically active RNA molecules) and RNA structural switches (or riboswitches)." I think this is an overstatement and should be better put into context. Also, as riboswitches control formation of proteins, it's rather their function as metabolite sensor and not their entire architecture that can be linked to an RNA world.
- 2) Sometimes the authors use the terms "RNA switch" and "riboswitch" interchangeably, although differently defined in the field and by their introduction. For example, in "For this, we extended our MPRA to include targeted mutations designed to shift the equilibrium between the two conformations of each riboswitch." the statement should have "candidate RNA switch", not "riboswitch". Another example is in legend to SFig. 2B.
- 3) Sentence "In total, we tested 3 mutation-rescue pairs. In all three cases, we observed lower eGFP expression of the conformation 2 mutant (117-AC), as compared to (77-GA) which favors conformation 1 (Fig. 5B)." – It's confusing to refer here to these specific mutations as additional mutants are shown in the display.
- 4) Why was the activity of the RORC switch tested in Th17 cells, where only the longer protein isoform is expressed? Is this switch in any way isoform-specific?
- 5) The text mentions the terms "structure screen" and "functional screen" in the section headed "Discovery of RNA switches with regulatory function in the human transcriptome". However, it remains unclear if "structure screen" refers to the next section, which deals with DMS-MaPseq, or to the section headed "Massively parallel mutagenesis identifies conformation-specific RNA switch activities", or both. If the terms are already coined, it would be helpful if they were consistently used. The workflow in Fig. 2A is only helpful if one understands what sections the terms are referring to.
- 6) Fig. 1A: At first, it can be confusing what is meant by "top", "middle", and "bottom", and the images of a human and a computer are not helping much. Brackets and direct labelling in the figure might be better.
- 7) Fig. 2B/functional screen: it should be mentioned if bin 1 contains cells with highest or lowest eGFP/mCherry ratio. It is possible to deduce that from the figure, but it may easily confuse the reader.
- 8) Legend to Fig. 4B: Text says "Class A is presented in red, Class B in blue", but it's the other way around in the display. And in the last sentence, class 3 is mentioned, which probably refers to class C. Furthermore, according to Fig. 4, Class A from the Cryo-EM images represents conformation 1, whereas Class B represents conformation 2. However, in Fig. 7, the schematic previously illustrating

Class A appears for conformation 2, and Class B for conformation 1. The colours match, but the shapes of the schematics don't.

9) Display of Fig. 5: Axis labels uses different font sizes and some are hard to read; furthermore, the scaling of the y-axis varies and should be defined. Display scheme in A and B could be more similar for easier interpretation (e.g., major conformation consistently indicated by pictogram or text)

10) Legend to Fig. 6C: The legend says ratio WT to scrambled sequence, the axis label says scrambled sequence to RORC switch expression difference, %. What is shown?

11) SFig. 1A: doesn't mention which species this TPP riboswitch was taken from. I assume it is *E. coli*, since that is the most prominent switch in the paper cited (Barsacchi et al. 2016), but TPP riboswitches also exist in other species

12) Further information would be needed for SFig. 5 to clarify its contribution here. Are the fractions supposed to be different or identical, and if 4 replicates were analysed, why are then not the data from all replicates compared?

13) Legend to SFig. 5: "IL-117A" should be "IL-17A"

Reviewer #2:

Remarks to the Author:

The authors present an original, innovative approach to discover RNA structural switches in eukaryotic cells. They combine a wide array of informatics, RNA structure probing, reporter gene assays, and even cryo-EM, which has not yet been applied to RNA switches to date. The methods are all novel. This kind of work can be a major methodological step forward in the field of RNA biology.

My main concern is that none of the methods presented have been benchmarked against known riboswitches in bacteria or even artificial switches and aptamers. As an example, the theophylline binding aptamer is one case which is widely applied to regulate gene expression in current eukaryotic model systems. I think that proper benchmarking of the methods presented is what is really lacking before publication as a methodological paper. There are a few other issues that the authors should consider:

Major:

For a work with a sizable computational component it is imperative to share code in such a way that fellow researchers can easily reproduce the analysis, or at the very least test it to ensure the code runs and produces the expected results. Unfortunately, although the authors do share a github repository with the code they used in the study it is practically unusable in the present form. The software is not pip-installable because (contrary to what's stated in README file) it lacks appropriate functions typically provided in setup.py and setup.cfg files. The requirements specified in README differ from those provided in requirements.txt. RNAPathfinder which is one of the two non-python dependencies for the software is not available at the address provided by the authors. Its web server version appears to have been down to some extended period of time as shown here:

<https://openebench.bsc.es/tool/rnapathfinder>

I was unable to find alternative channels to obtain RNAPathfinder including via github or bioconda.

Perhaps (if the license allows) the authors could include it as a submodule in their repository. These issues should be addressed before this work could be recommended for publication.

Line 110: The authors must present data demonstrating that SwitchFinder accurately selects against highly structured RNA elements such as transfer RNAs and ribosomal RNAs. Having multiple low energy minima in the landscape is not unique to riboswitches. I fear the algorithm has overfit the bacterial riboswitch class.

Line 167: 14% of the predicted switches demonstrated differences from the scrambled control. This is a low accuracy for the algorithm presented. As a comparison, the authors should present an experiment cloning a SwitchFinder predicted "non-RNA-structural switch" downstream of eGFP and compare it to its scrambled control, would there be a difference in expression between the two? I assume there would be a difference in expression at the thresholds chosen.

Line 233: Which features are suggestive of RNA secondary structure? The authors need to translate the 3D classes into structures using the Leontis-Westhoff classification. As far as this reviewer is aware, no study has ever demonstrated that different 3D Cryo-EM classes translate into different RNA secondary structures.

Line 269: The experiment with antisense oligos needs to be presented in comparison to untransfected cells. Antisense oligos activate the RNase H degradation pathway thus I highly doubt that addition of an antisense oligo that targets this region will activate gene expression by altering the conformation of the mRNA.

Line 265: The authors claim in the preceding paragraph that the alternative RNA conformations play divergent functional roles but they then present mutagenesis data that "all three mutants lower eGFP expression." Favoring one of the alternative conformations should increase eGFP expression and it did not. Can the authors explain this?

Minor

Line 44: How can the authors claim that ribozymes and riboswitches are the oldest known groups of RNA-based regulatory mechanisms? There needs to be a citation or reasoning for this. Are these mechanisms older than transcription per se, or older than the alternative sigma factors or ribonuclease decay pathways? Older than transcription termination? This claim is unfounded.

Line 45: Please change the words "RNA switches" to "Riboswitches." They are not the same.

Line 48: How can the authors claim that bacterial riboswitches are one of the most widely observed mechanisms for gene expression control? The authors either need to present a citation or some experimental data supporting this.

Line 67: Please change the word "showed" to "hypothesize."

Line 150: I am not aware of any data to support the notion that if multiple conformations co-exist, they all contribute the reactivity profile. The DMS modification rate depends on solvent accessibility, protein binding, R-loop formation, reverse transcription read-through, sequence context, and other factors, not on structure only. Many poorly resolved nucleotides in DMS or SHAPE-seq experiments

occur genome-wide, they are low probability modification sites which are unable to be accurately called for any number of reasons, not RNA switches.

Line 197: The authors need to explain how the mutations were designed. Covariance mutations should be used in this case.

Line 240: Change the word "evinced" to "evidenced"

Line 242: The authors claim that the low-resolution models presented are sufficient for making predictions of RNA fold and handedness. How is this possible? The authors need to show their structure probing data in a coherent structural model that also corresponds to the 3D states of the same RNA molecule. How can you state that Class B represents one confirmation or another?

Line 257: Two-fold difference in relative expression is very low. Most bacterial riboswitches regulate gene expression on the order of 5-10-fold activation or repression. One cannot rule out here that there is a sequence effect on the UTR.

Line 260: Since no control data presented from a non-switch 3' UTR, one cannot conclude that the two conformations play divergent functional roles.

Line 376: "Both approaches were used to discover the first known RNA switches in bacteria..." Cite here Mironov et al (2002) PMID: 12464185 too, as this paper indeed reports the first two known riboswitches before the term "riboswitch" has been adopted.

Reviewer #3:

Remarks to the Author:

Summary of the key results

Although ligand-dependent conformational switching is prominent in the transcripts of numerous bacterial RNAs – with >55 validated classes of riboswitches — the extent to which structural switches control gene expression in eukaryotes remains largely unexplored. Here, Khoroshkin and co-workers used a massively parallel approach to identify a large number of human regulatory elements that adopt two conformations, and behave as gene-regulatory RNA switches. The findings suggest widespread conformation-dependent control of gene regulation in the human transcriptome.

Originality and significance

A major innovation is the development of the methodology called Switchseeker, which is likely applicable to many other switches beyond the human transcriptome and test cases presented. The authors also give compelling evidence that specific regulatory elements adopt two (or more) conformations, along with a plausible mechanism of action by which the upstream gene is regulated.

Clarity and context

Introduction. In the Introduction, the authors describe RNA switches as elements that control gene expression by direct binding of a small-molecule ligand or other trans-acting factor. Importantly, this is the definition of a riboswitch. A better working definition for this study is a 'switch RNA' that adopts two mutually exclusive conformations that lead to different gene-regulatory outcomes. This early differentiation of RNA switches is essential because the average reader will be confused that the RNA

switches of this study are not actually riboswitches. It is better to define riboswitches as a specialized case of an RNA switch.

Introduction. The authors also state, "The search for such ligand-binding riboswitches in eukaryotes has had limited success to date. Just two human examples are known: the RNA switch in VEGFA and the m6A modification-based switches (Liu et al. 2015; Ray et al. 2009)." This sounds like the VEGFA and m6A switches are ligand-sensing RNA switches, which is not correct based on the accepted definition of a riboswitch. As the authors know, eukaryotes have been shown to possess riboswitches that sense TPP (in plants and fungi), and these must be included if the authors wish to point out ligand-sensing RNA switches in eukaryotes.

Results. At the start of the Results, the authors should define what they mean by "RNA structural switches", which will be the premise of the paper. As worded, this is vague. In particular the authors should define that there are two mutually exclusive conformations that are probably nearly isoenergetic in folding but may be perturbed to adopt a conformation that stabilizes or destabilizes the upstream transcript leading to changes in gene expression. In other words, a cogent working definition is needed here.

Results. Although a minor point, the authors describe DMS-seq as "single-nucleotide resolution." However, some qualification is needed here because the method does not give a full picture of the modification landscape since only A and C are modified.

Results. The authors state, "The accessibility of a single nucleotide is a population average of multiple RNA molecules that represent different minima in the RNA folding conformation ensemble." This statement should mention different "Gibbs free energy minima."

Results. In the second iteration of SwitchSeeker, how did the application of chemical modification data improve the results that led to a high confidence set of RNA switches? Did the chemical probing data help to better define the accuracy of regions that adopt two mutually exclusive conformations? Or did it help to eliminate problem sequences that do not adopt two clear switch conformations based on RNA structure prediction alone? Or both? Some clarification would be appreciated. A major point for the user is that either approach would succeed in the workflow, but what are the pros and cons of each?

For the TCF7 RNA switch in Fig 2D, what was the difference in Gibbs free energy for the two predicted conformations? This information will be interesting to the reader.

Suggested improvements/Data & methodology

Cryo-EM Results. The reviewer disagrees that the 2D class averages provide sufficient evidence to recognize the RNA fold and handedness. No cogent evidence is provided to support this statement. Specifically, the various particles do not appear to contain the amount of RNA attributed to the folds produced from chemical modification. Only partial models were docked into the potential maps providing an incomplete analysis of the putative RNA conformation. The addition of other evidence is needed to support the composition of the particles, such as SEC-MALS. This is necessary because the class A particle appears much larger than class C and appears to accommodate twice as much RNA. Even at the stated resolution of ~ 10 Å, it should be feasible to model a secondary structure based on chemical modification. This aspect of the study was one of the weakest.

For the cryo-EM analysis, a major concern is whether the samples used form a self-complementary dimer. This could be analyzed to examine the free energy difference between the dimer and monomer to support their understanding of the cryo-EM particle compositions. SEC-MALS analysis could be used to support this assignment.

Notably, information from the cryo-EM section was absent in the reviewer's version of the manuscript: (i) Table S8 described in the Methods was not included in this manuscript submission; there does not appear to be any previous Tables either in the Supplemental Information. (ii) Table S7 described in the Methods was not included in this manuscript submission. (iii) Extended Data does not appear to have been included in this submission. Please correct these call outs and provide the information as needed.

DMS-MapSeq and cryo-EM data suggest that the RORC 3' mRNA element inhabits a shallow energy landscape with two rugged minima linked to two major molecular conformations

In the Discussion the authors state that they have provided experimental structures of switch states, which validates the SwitchSeeker approach to identify RNA molecules with of bistable energy landscapes. The reviewer believes this is an overinterpretation of the cryo-EM data. The structure is not really determined until an atomistic model is fit into the potential maps. The authors present small pieces of RNA structure docked into their molecular envelopes in Supp Fig. 4, but the complete models are not shown. With DMS-seq data and the molecular envelopes, it seems reasonable that models could be made even at ~ 10 Å. This would be much more supportive of the claim that experimental structures were determined, and that the RNA adopts the conformations claimed. A better approach to interrogate the thermodynamic ensemble is SAXS. In this instance, all three conformations could be identified in solution and compared to the particles restored by cryo-EM. AFM has also seen recent advances in the ability to monitor particles in the thermodynamic ensemble [Ding et al. Wang 2023 Nat Comm 14, 714]. A main concern with cryo-EM is that not all particles observed give rise to the class averages used for potential map calculations.

Conclusions: robustness, validity, reliability

One of the hallmarks of riboswitches is the presence a consensus model for a particular switch in multiple species. Are there similar elements like RORC RNA non-human primates or metazoans that suggest comparable switching of the RORC transcript?

As noted by the authors, UPF1 has been reported to be part of a structure-mediated decay pathway (Fischer et al. 2020). This mechanism of action is plausible here but is differentiated by the requirement for the dsRNA binding protein G3BP1, which is not associated with NMD factors. Is there any evidence that G3BP1 knockdown restores transcript levels as shown for knockdowns of the core SURF complex? This has implications for the model proposed in Fig. 7. Some discussion of this structure-mediated decay mechanism is needed.

As noted by the authors, a lingering question for the switch is what factors influence the ability to partition into each gene-regulatory state? As the authors know, riboswitches provide feedback by adopting one of two mutually exclusive conformations that depend upon the concentration of a cognate cellular effector. Do the authors envision differences in the protein expression landscape that promote one conformation over another? Some discussion of known RNA switches, such as VEGF,

would be appropriate.

Are the switches described here under thermodynamic control or kinetic control? The use of DMS-seq and other equilibrium methods suggests the former, but this point is not discussed. The model in figure 7 suggests there is some kinetic barrier in the free-energy landscape that joins conformation 1 and 2. If this is true, the kinetics must be extraordinarily slow to allow identification by cryo-EM. Wouldn't this be better described as a local, stable free energy intermediate? The energy diagram suggests it is high-energy but wouldn't this be separated by a high-energy barrier as shown but with a deeper local minimum? Is there any experimental evidence for the interconversion of any conformation to one of the other folds? For example, if the sample is run on gel filtration or a native gel, does a single isolated species repartition into all three states? At present, there is very little support for the proposed model (except that it explains a potential artifact of cryo-EM).

A major concern is that this paper is not written like a Nature Methods paper. The Methods are difficult to follow and there is no significant consideration that the end user would want to replicate the result or develop derivative approaches.

Minor

As a comment, it would be interesting to see if any of the identified sequences in the 3'-UTR show evidence for aptamer homology to known bacterial riboswitches. Riboswitch aptamers are the most conserved elements of these regulatory molecules, and one could envision how ligand-dependent folding could promote decay pathways in metazoans.

Is it known whether any of the RNA structures have been modified by post-transcriptional modifications that could modulate the structures? (Perhaps beyond the scope of this work)

Several places in the Methods and Supplemental Information refer to the 'RNA switches' as "riboswitches". This is incorrect because the switches identified here do not conform to the definition of riboswitches, which respond to small molecules or ions as effectors of RNA conformation (e.g., Supple Fig. 2 and Methods section on CRISPRi).

There are several minor typos in the document that can be easily identified using the MSWord spellcheck tool. These are flagged by red underscoring.

Supple Fig 2. The authors state that DMS allows assignment of base pairing flexibility, but this modification is better correlated with WC-face accessibility rather than flexibility (which best describes SHAPE).

References: appropriate

Author Rebuttal to Initial comments

Reviewers' Comments:

Reviewer #1:

Remarks to the Author:

The manuscript entitled „A systematic search for RNA structural switches across the human transcriptome” by Khoroshkin et al. describes a pipeline combining bioinformatics and experimental methods to identify mRNA regions that can result in alternative structural conformations and alter gene expression. Specifically, RNA structures predicted in silico are tested for experimental support by probing methods in a high-throughput manner. The most promising candidates are then tested for their potential to alter expression of a reporter gene in a structure-dependent manner. From the approximately 250 high-confidence candidates identified in this work, the top potential switch derived from the 3' UTR of the RORC transcript was further characterized. These experiments confirmed that this RNA sequence can form two alternative structures that are linked to different expression levels in a reporter gene context. Furthermore, the authors provided evidence that the lower expression observed for one of the conformations is caused by its targeting for degradation via the RNA surveillance system nonsense-mediated decay (NMD).

I think this is an interesting study using high-throughput approaches to identify novel structured mRNA motifs with potential functions in gene regulation. The strategy is overall clearly described and evidence is provided that the candidates can form alternative structures which are linked to altered expression, at least in a reporter context. However, I don't think that the study provides compelling evidence that these motifs are indeed switches, i.e., that the proportions of alternative conformations can change in cells and thereby alter gene expression. What the authors have identified are RNA regions that can fold into different conformations, which can affect gene expression differently. Both findings are not surprising from my point of view, as RNA is well known to adopt alternative folds and many ways how these folds can interfere with gene expression, in particular with respect to the rather weak changes as seen for the RORC motif, can be imagined. So, I think the authors have identified switch candidates, but more experiments would be needed to confirm that switching actually takes place in vivo and has an impact on gene expression. Furthermore, their functional experiments on the RORC candidate are restricted to the reporter context. To show that this motif is biologically relevant, the authors would need to analyse the motif's impact on expression of the endogenous gene.

We thank the reviewer for their thoughtful review and positive comments. We have taken steps to address the comments raised below,

Major comments:

Comment 1.1: This manuscript aims at identifying novel RNA switches similar to metabolite-sensing riboswitches or protein-interacting RNA folds such as the VEGFA switch. Demonstrating that an RNA has the potential to form different RNA structures and can regulate expression of a reporter construct in a conformation-specific manner is not sufficient to establish it as a switch. From my point of view, further experimental data would be needed to show that the relative proportions of alternative conformations change between conditions and that this is linked to altered gene expression.

Response to 1.1: We thank the reviewer for bringing this point to our attention. In our initial telling of this work, our intention with using the term 'switch' was to emphasize a switch in function as a consequence of RNA conformation rather than a toggling switch in RNA molecules. In other words, even if an element takes on one of the two conformations at the time of transcription and stays in that conformation for the life-span of RNA, that element is still a functional switch as the two conformations diverge in their regulatory impact. Reviewer #3 had suggested we define this early on in the text: "...A better working definition for this study is a 'switch RNA' that adopts two mutually exclusive conformations that lead to different gene-regulatory outcomes." We have now taken steps to ensure that this point is made clear in the introduction (lines 49-51).

However, we now also have data that demonstrates that our RORC switch does behave as the reviewer describes, i.e. "the relative proportions of alternative conformations change between conditions and that this is linked to altered gene expression". We first examined if the relative proportions of alternative conformations would change based on the genetic background of various cell lines: LNCaP (prostate), MCF-7 (breast), HepG2 (liver), ZR-75-1 (breast), 293T (kidney), LS174T (colon). Using DMS-seq, we assessed the changes in the accessibility of regions 2 and 3 within these RNA conformations. Our data, shown in **Figure R.1A**, confirms that the relative proportions of these alternative conformations do vary among the cell lines, and that the relative accessibility of these regions is strongly anticorrelated as measured by DMS signal ($R=-0.75$). This observation reveals that the relative proportions of the two conformations can change in different cellular contexts.

We then investigated if these conformational changes are tied to gene expression changes. For this, we performed RNA stability measurements by inhibiting RNA Polymerase II activity with α -amanitin. We measured the eGFP-RORC 3'UTR reporter mRNA stability by comparing the mRNA levels in amanitin-treated and non-treated cells across the same set of cell lines. We found that the ratio between the conformations is strongly correlated with the reporter mRNA stability ($R=0.85$, $P=0.03$, **Figure R.1B**), further highlighting the link between RNA conformation and gene expression.

Figure R.1. (A) Accessibility of "box 2" (X axis) and "box 3" (Y axis) regions of RORC element across cell lines, as measured by DMS-seq. **(B)** Relationship between the relative conformation ratio of RORC element (X axis) and stability of the reporter mRNA, as measured by RT-qPCR.

Comment 1.2: It doesn't become clear why the authors focused in their search on 3' UTRs, given what we know about other RNA switches. Riboswitches are widespread in bacteria, whereas in eukaryotes so far only TPP-sensing riboswitches have been characterized. These eukaryotic riboswitches are present in filamentous fungi, green algae, and plants, where they are located in introns and control splicing decisions. As introns can house regulatory elements including structured RNAs without affecting the open reading frame, I think these regions would be a logical choice to look for novel structured RNA motifs in (other) eukaryotes. Any reference to these known eukaryotic riboswitches is missing in this manuscript. Why didn't they analyse intronic sequences? When testing their pipeline (SFig. 1B), why did they use only bacterial riboswitches? Testing TPP riboswitches in more complex eukaryotic genomes would be more revealing with respect to the aims of this study. How did they decide to analyse 3' UTRs in fragments of 186 nt length? The current pipeline can only detect RNA elements that affect gene expression independent from other elements outside of this rather small window, and therefore would fail to detect, at least at the step of functional characterisation, elements such as the eukaryotic TPP riboswitches.

Response to 1.2: We appreciate the reviewer's thoughtful comments on the limitations of our methodology. Indeed, this work is intended to serve as an initial study to establish basic methodologies for RNA switch discovery in eukaryotes. We fully acknowledge that our approach is not comprehensive and likely misses many relevant RNA switches, particularly those that may be located in the coding sequence, introns, or 5'UTRs. And in fact, we intend

to expand to these regions in future studies. However, we selected 3'UTRs for two main reasons: (i) we were intrigued by the possibility of finding RNA switches in 3'UTRs, as 5'UTRs has been the main focus of the search for such elements, and (ii) elements in 3'UTR are not confounded by changes in the dynamics of ribosome entry and translation initiation. Since it is known that structured RNA can impede effective translation, it is not surprising that a more stable conformation would hinder initiation. In case of 3'UTR elements, however, the molecular mechanisms would be more intriguing and complex.

Regarding the limitation of examining 186 nt fragments, this was a technical constraint imposed by the current capabilities of DNA oligo pool synthesis. We anticipate that technological advances will enable broader analyses in future studies.

We also appreciate the reviewer's suggestion to test our software on known eukaryotic TPP riboswitches. SwitchFinder demonstrated increased effectiveness when applied to these switches, as shown in **Figure R.2**. This result further supports the utility of our approach.

Figure R.2. auROC values of SwitchFinder predictions of RNA switches within the select groups of RNAs. We applied SwitchFinder to a mix of “real” sequences and their shuffled counterparts, and measured the ability to correctly select the “real” sequences only. SwitchFinder performs significantly worse on non-switching ribosomal RNAs compared to bacterial, eukaryotic, and artificial riboswitches.

Comment 1.3: The analysis of the RORC motif is limited to the reporter context. The authors provide evidence for the existence of alternative conformations and their effect on transcript levels in a reporter context. However, what would be observed in the natural gene context? Did the authors also perform reporter assays with the complete 3' UTR? Furthermore, the effect of mutations and conformations for endogenous RORC would need to be analysed to

refer to this element as a regulatory switch. Based on the presented data, it remains open if the weak differences seen in transcript levels for the two conformations in the reporter context would also cause a change in RORC transcripts, when the element is in its natural exon/intron context and can also get involved in base pairing with more distant regions. As also mentioned before, the authors would need to show that the conformational distribution of RORC 3' UTRs differs between certain conditions in vivo and that this is linked to altered RORC levels. The assays involving ASOs and proteasomal inhibition could also be used to test the relevance of the structure in vivo. Again, the corresponding assays in this manuscript are restricted to the reporter context. Finally, it does not become clear how distinct this motif is from the other ~250 candidates, as any information on these is missing.

Response to 1.3: We thank the reviewer for this comment. To directly address the limitations pointed out regarding the reporter context, we expanded our analysis to include the endogenous RORC locus. Importantly, this was conducted across the five cell lines previously mentioned in response to Comment 1. Using DMS-seq, we assessed the accessibility of regions 2 and 3 within the endogenous RORC mRNA. We found a similarly strong anti-correlation in accessibility of the two competing boxes ($R=-0.81$) relative to what was observed in the reporter context ($R=-0.75$) (**Figure R.3A**). This figure displays results for both the reporter and endogenous contexts side by side. Furthermore, the accessibility ratios between the reporter and endogenous contexts were highly correlated ($R=0.93$, **Figure R.3B**), strengthening the argument that the RORC switch behaves consistently across different genetic settings.

For stability measurements, we followed the same approach as outlined in response to Comment 1. The stability of endogenous RORC mRNA showed a strong correlation with RNA conformation ratios ($R=0.96$, $P=0.004$, **Figure R.3C**), underscoring that the RORC element functions as a regulatory switch in its native context as well.

Lastly, we assessed the impact of ASOs targeting the RORC switch on the stability of the endogenous RORC mRNA. The introduction of these ASOs led to significant shifts in mRNA stability (**Figure R.3D**). Notably, these effects were more significant in cell lines with a higher proportion of conformation 2 (LNCaP $P=0.006$, MCF-7 $P=0.005$) as opposed to those with a lower proportion (LS174T $P=0.71$). These findings align with our hypothesis that ASOs, which specifically target box 2, function by tipping the balance away from the inhibitory conformation 2 towards conformation 1. Thus, the data supports the notion that the relative proportions of these conformations are intrinsically linked to mRNA stability in the endogenous setting.

We have also taken the reviewer's advice to further delineate RORC from other candidates by elaborating on our methods section and clarifying our scoring process for candidate switches (see lines 218-220 and the "*Massively Parallel Mutagenesis Analysis*" subsection in Methods).

In summary, our additional data provide strong support for the role of the RORC switch in its natural, endogenous context.

Figure R.3. (A) Accessibility of "box 2" (X axis) and "box 3" (Y axis) regions of RORC element across cell lines, as measured by DMS-seq. Left: the accessibility was measured in the context of a GFP reporter containing the RORC element sequence. Right: the accessibility was measured in the context of endogenous RORC mRNA. **(B)** Correlation of relative proportions of the two conformations between the reporter context and the

endogenous RORC mRNA. **(C)** Relationship between the relative conformation ratio of RORC element (X axis) and stability of the endogenous RORC mRNA, as measured by RT-qPCR. **(D)** The effect of ASOs targeting one conformation of the RORC element on expression of the endogenous RORC gene across 3 cell lines. Only the cell lines with the high proportion of conformation 2 (LNCaP, MCF-7) demonstrate a significant change.

Comment 1.4: To test for the presence of the alternative conformations *in vivo*, the authors performed DMS-MaPseq of the RORC switch followed by a computational analysis (DRACO). To validate this method, the same type of analysis should be performed for the mutant set shown in Fig. 3C. It would be expected that 117-AC has an increased proportion of conformation 2, and that 65-GT, 117-AC gives a similar output as the wildtype.

Response to 1.4: We appreciate the reviewer's suggestion. It should be noted that we had already confirmed the switch for these mutants using RNA SHAPE analysis (Fig. 3C). To further address this, we conducted *in vivo* DMS-MaPseq experiments for the mentioned mutants, consistent with our previous *in vitro* analyses. As indicated in **Figure R.4**, the 63-TC, 77-GA mutation favors conformation 2, while the 65-GT, 117-AC mutation leans toward conformation 1. These results corroborate our hypothesis that the candidate switch exhibits the same conformational behavior *in vivo* as it does *in vitro*.

Figure R.4. The effect of sequence mutations in the "box 2" and "box 3" regions of RORC element on their accessibility, as measured by DMS-seq.

Comment 1.5: The link between RORC turnover and NMD remains unclear. The authors provide evidence that NMD affects the turnover of reporter transcripts containing the RORC element in a conformation-dependent manner. According to these data, NMD inhibition would then also be expected to affect endogenous RORC levels. Why didn't the authors test this?

Their model in Fig. 7 suggests that interaction with UPF1 is directly dependent on the RNA conformation and they refer to a previous study by Fischer et al. reporting structure-mediated RNA decay (SRD) by UPF1. However, SRD was previously reported to be independent of other NMD factors, while here in this study besides UPF1 several other NMD factors were found to regulate the expression of the reporter containing the RORC motif. Moreover, it seems more likely that UPF1 may be recruited by other factors to this specific RNA conformation, rather than it directly recognizes a specific structure (as indicated in Fig. 7).

Response to 1.5: We thank the reviewer for this comment. To investigate the relationship between NMD and endogenous RORC levels, we knocked down UPF1 using siRNA and assessed the expression of endogenous RORC mRNA via RT-qPCR. We observed a substantial increase in RORC expression upon UPF1 knockdown (**Figure R.5A**). This effect was more pronounced in cell lines with a higher occurrence of conformation 2 (LNCaP $P=0.005$, MCF-7 $P=0.02$) compared to those with a lower occurrence (LS174T $P=0.09$), substantiating our hypothesis that UPF1 influences RORC mRNA stability in a conformation-dependent manner.

Building on this, we further explored how NMD inhibition impacts the distribution of RORC mRNA conformations. We utilized NMDI14, a small molecule that inhibits NMD by disrupting SMG7-UPF1 interactions, and assessed the accessibility of regions 2 and 3 of endogenous RORC mRNA using DMS-seq. We observed a significant decrease in the accessibility of region 2 ($P=0.03$, **Figure R.5B**), aligning with our expectation of slower decay and accumulation of mRNAs in conformation 2. Hence, inhibiting NMD led to a shift in the relative proportions of the two conformations.

To further clarify the role of UPF1 in regulating RNA, we point the reviewer to data presented in the original manuscript, specifically in Fig. 6E. In this part of the study, we utilized crosslinking immunoprecipitation (CLIP) followed by targeted sequencing to examine UPF1's interaction with RNA. Importantly, CLIP is a method specifically designed to identify direct protein-RNA binding events, as established in previous literature (Lee and Ule 2018). Therefore, our CLIP data, which showed that the wildtype RORC UTR sequence was significantly more represented among UPF1-bound RNAs compared to its 77-GA mutant, strongly supports the direct binding of UPF1 to the RNA. This evidence underscores our model of conformation-specific regulation by UPF1.

While our data suggests a direct interaction between UPF1 and the RNA switch, it is important to consider that the underlying mechanism could involve additional factors influencing this interaction. However, the evidence we present here primarily supports UPF1's direct binding to the mRNA. We propose that this interaction is part of a mechanism falling under the category of EJC-independent NMD, as discussed in (Kurosaki, Popp, and Maquat 2019). This implies that UPF1, along with other NMD factors, is instrumental in regulating RORC expression. Further investigation into the intricate roles of these NMD factors,

particularly in relation to RNA structure and function, remains an important and promising focus for future research.

Figure R.5. (A) The effect of siRNAs targeting UPF1 on expression of the endogenous RORC gene across 3 cell lines. Only the cell lines with the high proportion of conformation 2 (LNCaP, MCF-7) demonstrate a significant change. **(B)** The effect of NMDI14 on the accessibility of "box 2" and "box 3" regions of RORC element. The inner plot demonstrates the normalized accessibility at individual nucleotide positions in box 2 (as measured by DMS-seq), across two conditions.

Comment 1.6: Data presentation and analysis lack clarity at several places. Cryo-EM is performed to provide further evidence for the existence of the alternative conformations. As it is currently presented, I think it's very difficult for the reader to make any conclusion on this. Fig. 4A shows segments of EM micrographs, for which even after zooming in the numeric labels and structures are hardly visible. SFig. 4 shows more details of the corresponding analyses. What would be needed from my point of view is a compilation of the alternative structures that were combined into the structural classes and conformations show in Fig. 4, including individual counts.

Response to 1.6: We thank the reviewer for pointing out that Fig. 4A was unclear and hard to read. We have changed Fig. 4A to show closeups of individual particles, instead of the

micrograph segments on which the particles were difficult to see. The segments themselves, with the example particles labeled, have been moved to Extended Data Fig. 5A-C. These examples are intended to provide a qualitative sense for the heterogeneity of the particles and their potential changes in conformation only. Even with phase-flipping as used here, the images contain residual corruption by the CTF and of course are projections in one direction only. The 3D classes, which are originally derived from a reference-free *ab initio* method, and are CTF corrected and fully 3D are the best way to understand the stable or metastable structures adopted by the sample. We do provide the particle counts for these 3D classes in Extended Data Fig. 4, however we caution that due to the nature of cryo-EM processing the particles are unlikely to represent the full thermodynamic ensemble and should not be used to estimate free energy differences. Therefore, we limit ourselves to qualitative interpretation in the main text, for example the complete absence of class B in the 77-GA mutant (or class C in the 117-AC mutant).

Fig. 5A-D are examples, where the data analysis is not sufficiently described. What are the boxes, whiskers and lines (maybe mean values)? And was the statistical test done on the mean value? I didn't find any supplemental material with these kind of details (which should be provided in the legend as well). I also didn't have access to a supplemental table and other supplemental data files mentioned in the supplement section.

We thank the reviewer for pointing this out. We have corrected the description of the data analysis for the Fig. 5 in the revised manuscript. We apologize that the uploaded SI was not made available, we have asked the editor to ensure they will have access to the material.

Minor points:

Comment 1.7: The authors make the following statement: "The two oldest groups of RNA-based regulatory mechanisms are ribozymes (catalytically active RNA molecules) and RNA structural switches (or riboswitches)." I think this is an overstatement and should be better put into context. Also, as riboswitches control formation of proteins, it's rather their function as metabolite sensor and not their entire architecture that can be linked to an RNA world.

Response to 1.7: We thank the reviewer for this comment. Our statement regarding ribozymes and riboswitches as some of the oldest known RNA-based regulatory mechanisms is grounded in the RNA world hypothesis. This hypothesis, as detailed by Gilbert in 1986, posits that RNA molecules, particularly ribozymes, predate both DNA and proteins in evolutionary history (Gilbert 1986). The review by Vitreschak et al. (2004), a seminal work in the field, is titled "Riboswitches: the oldest mechanism for the regulation of gene expression?" (Vitreschak et al. 2004). We have updated the statement in the revised manuscript (lines 45-48).

Comment 1.8: Sometimes the authors use the terms “RNA switch” and “riboswitch” interchangeable, although differently defined in the field and by their introduction. For example, in “For this, we extended our MPRA to include targeted mutations designed to shift the equilibrium between the two conformations of each riboswitch.” the statement should have “candidate RNA switch”, not “riboswitch”. Another example is in legend to SFig. 2B.

Response to 1.8: We thank the reviewer for this observation. We have updated the statements to use the proper terminology.

Comment 1.9: Sentence “In total, we tested 3 mutation-rescue pairs. In all three cases, we observed lower eGFP expression of the conformation 2 mutant (117-AC), as compared to (77-GA) which favors conformation 1 (Fig. 5B).” – It’s confusing to refer here to these specific mutations as additional mutants are shown in the display.

Response to 1.9: We thank the reviewer for pointing this out. We have clarified this statement in the revised manuscript (lines 305-307).

Comment 1.10: Why was the activity of the RORC switch tested in Th17 cells, where only the longer protein isoform is expressed? Is this switch in any way isoform-specific?

Response to 1.10: We appreciate the reviewer’s inquiry about our choice to test the activity of the RORC switch in Th17 cells. We acknowledge that including information about the two isoforms of RORC might have introduced some confusion. To clarify, the RORC switch is not specific to any particular isoform. Our decision to focus on the longer isoform of RORC in Th17 cells was driven by its well-established role as a key regulator of Th17 cell differentiation. Given the significant biological relevance of RORC in Th17 cells, as opposed to other tissues, our analysis of the RORC switch in the Th17 cells was tailored to this specific context. This rationale has been more clearly articulated in the revised version of the manuscript to ensure a better understanding of our experimental approach (lines 310-318).

Comment 1.11: The text mentions the terms “structure screen” and “functional screen” in the section headed “Discovery of RNA switches with regulatory function in the human transcriptome”. However, it remains unclear if “structure screen” refers to the next section, which deals with DMS-MaPseq, or to the section headed “Massively parallel mutagenesis identifies conformation-specific RNA switch activities”, or both. If the terms are already coined, it would be helpful if they were consistently used. The workflow in Fig. 2A is only helpful if one understands what sections the terms are referring to.

Response to 1.11: We thank the reviewer for this suggestion. In the revised manuscript, we have updated and clarified the Fig. 2A and ensured the consistent usage of the terms across the Results section.

Comment 1.12: Fig. 1A: At first, it can be confusing what is meant by “top”, “middle”, and “bottom”, and the images of a human and a computer are not helping much. Brackets and direct labelling in the figure might be better.

Response to 1.12: We thank the reviewer for this comment. We have clarified the Fig. 1A in the updated manuscript.

Comment 1.13: Fig. 2B/functional screen: it should be mentioned if bin 1 contains cells with highest or lowest eGFP/mCherry ratio. It is possible to deduce that from the figure, but it may easily confuse the reader.

Response to 1.13: We thank the reviewer for this suggestion; we have implemented it in the revised manuscript (see Fig. 2B caption).

Comment 1.14: Legend to Fig. 4B: Text says “Class A is presented in red, Class B in blue”, but it’s the other way around in the display. And in the last sentence, class 3 is mentioned, which probably refers to class C. Furthermore, according to Fig. 4, Class A from the Cryo-EM images represents conformation 1, whereas Class B represents conformation 2. However, in Fig. 7, the schematic previously illustrating Class A appears for conformation 2, and Class B for conformation 1. The colours match, but the shapes of the schematics don’t.

Response to 1.14: We thank the reviewer for this comment. These errors have been corrected in the revised manuscript.

Comment 1.15: Display of Fig. 5: Axis labels uses different font sizes and some are hard to read; furthermore, the scaling of the y-axis varies and should be defined. Display scheme in A and B could be more similar for easier interpretation (e.g., major conformation consistently indicated by pictogram or text)

Response to 1.15: We thank the reviewer for this comment. We have updated the Fig. 5 in the revised manuscript.

Comment 1.16: Legend to Fig. 6C: The legend says ratio WT to scrambled sequence, the axis label says scrambled sequence to RORC switch expression difference, %. What is shown?

Response to 1.16: We thank the reviewer for pointing this out. This error has been corrected in the revised manuscript.

Comment 1.17: SFig. 1A: doesn’t mention which species this TPP riboswitch was taken from. I assume it is *E. coli*, since that is the most prominent switch in the paper cited (Barsacchi et al. 2016), but TPP riboswitches also exist in other species

Response to 1.17: We thank the reviewer for this comment. In the revised manuscript, we clarified which species the TPP riboswitch was taken from.

Comment 1.18: Further information would be needed for SFig. 5 to clarify its contribution here. Are the fractions supposed to be different or identical, and if 4 replicates were analysed, why are then not the data from all replicates compared?

Response to 1.18: We appreciate the reviewer's request for further clarification regarding Supplementary Figure 5, now referred to as Extended Data Figure 6 in the revised manuscript. This figure is intended to serve as a quality control illustration for the Th17 differentiation assay. It was not our intention to compare IL17A+ cell fractions between different samples in this figure.

Upon reflection, we understand that including quality control data for several samples may have led to some confusion. To address this and streamline the presentation, we have revised the figure in the manuscript. We now showcase the data from just one of the three assays, ensuring that the focus remains on the quality control aspect of the Th17 differentiation assay without implying unintended comparisons.

Comment 1.19: Legend to SFig. 5: "IL-117A" should be "IL-17A"

Response to 1.19: We thank the reviewer for this comment. This error has been corrected in the revised manuscript.

Reviewer #2:

Comment 2.1: The authors present an original, innovative approach to discover RNA structural switches in eukaryotic cells. They combine a wide array of informatics, RNA structure probing, reporter gene assays, and even cryo-EM, which has not yet been applied to RNA switches to date. The methods are all novel. This kind of work can be a major methodological step forward in the field of RNA biology.

My main concern is that none of the methods presented have been benchmarked against known riboswitches in bacteria or even artificial switches and aptamers. As an example, the theophylline binding aptamer is one case which is widely applied to regulate gene expression in current eukaryotic model systems. I think that proper benchmarking of the methods presented is what is really lacking before publication as a methodological paper. There are a few other issues that the authors should consider:

Response to 2.1: We appreciate the reviewer's point regarding the need for thorough benchmarking of our method. However, unfortunately, we are limited by the lack of a proper

dataset for benchmarking. In our review of numerous studies on natural and artificial riboswitches, we found that detailed characterizations of secondary structures for the two alternative states are often missing. While many studies demonstrate the ability of riboswitch RNAs to modulate expression in response to a ligand, they typically do not provide structural details for both states. Additionally, the variety of experimental methods used across these studies makes compiling a single, high-quality dataset for benchmarking a significant challenge.

To address this limitation, we benchmarked SwitchFinder by analyzing its performance on known riboswitch sequences versus their dinucleotide-shuffled counterparts. The premise is that while a shuffled sequence lacks a functional riboswitch, its degree of secondary structure formation remains broadly similar due to preserved dinucleotide content. Our results indicate that SwitchFinder effectively distinguishes real riboswitch sequences from their shuffled versions (**Figure R.6A**). In contrast, when tested with highly structured non-switching RNAs such as ribosomal RNAs, the algorithm showed limited differentiation capability between real and shuffled sequences (auROC = 0.54). This suggests that SwitchFinder is adept at identifying regions in RNA that likely contain conformational switches.

Acknowledging the reviewer's suggestion, we have now extended our benchmarking to include a comprehensive set of theophylline riboswitches (Wang et al. 2023) and a collection of artificial protein-sensing riboswitches (Vezeau, Gadila, and Salis 2023). In these tests, SwitchFinder demonstrated an ability to differentiate artificial riboswitches from their shuffled counterparts, achieving auROC values of 0.74 and 0.69, respectively. We note that this computational step is intended to enrich for real switch elements that are then investigated through the MPRA measurement.

We believe these additional benchmarks reinforce the efficacy of SwitchFinder in identifying RNA switches and add value to our methodological approach.

Figure R.6. (A) auROC values of SwitchFinder predictions of RNA switches within the select groups of RNAs. We applied SwitchFinder to a mix of “real” sequences and their shuffled counterparts, and measured the ability to correctly select the “real” sequences only. SwitchFinder performs significantly worse on non-switching ribosomal RNAs compared to bacterial, eukaryotic, and artificial riboswitches. **(B)** Example of SwitchFinder locating the RNA switch within the theophylline riboswitch RNA sequence. Arc representation of the RNA base pairs that change between the two conformations of theophylline riboswitch, as in (Lynch et al. 2007) (top), and as predicted by SwitchFinder (bottom). The two conformations are shown in red and blue, respectively.

Major:

Comment 2.2: For a work with a sizable computational component it is imperative to share code in such a way that fellow researchers can easily reproduce the analysis, or at the very least test it to ensure the code runs and produces the expected results. Unfortunately, although the authors do share a github repository with the code they used in the study it is practically unusable in the present form. The software is not pip-installable because (contrary to what’s stated in README file) it lacks appropriate functions typically provided in setup.py and setup.cfg files. The requirements specified in README differ from those provided in requirements.txt. RNAPathfinder which is one of the two non-python dependencies for the software is not available at the address provided by the authors. Its web server version appears to have been down to some extended period of time as shown here: <https://openebench.bsc.es/tool/mapathfinder>

I was unable to find alternative channels to obtain RNAPathfinder including via github or bioconda. Perhaps (if the license allows) the authors could include it as a submodule in their

repository. These issues should be addressed before this work could be recommended for publication.

Response to 2.2: We thank the reviewer for these suggestions. We recognize the importance of ensuring that our computational tools are easily reproducible and testable by the research community. In response to these concerns, we have made several improvements.

We have included the source code of RNAPathfinder in our GitHub repository. However, since RNAPathfinder requires compilation, this could pose a challenge for some users regarding installation. To streamline the process and ensure a more user-friendly experience, we have packaged our software, SwitchFinder, along with RNAPathfinder, in a Docker container. This approach simplifies the installation process and ensures consistent functionality across different computing environments.

The Docker image for SwitchFinder, including RNAPathfinder, is now available and can be easily accessed from the Docker hub. We have updated the README file on our GitHub repository with the instructions for using the Docker container, ensuring that users can smoothly pull the image and run the software without the complexities of manual installation. Additionally, we have provided a Supplementary Protocol that covers all the steps (both computational and experimental) of the method presented.

These enhancements address the critical points you raised and significantly improve the usability and accessibility of our software. We hope that these changes will facilitate a more efficient and effective use of SwitchFinder in the research community.

Comment 2.3: Line 110: The authors must present data demonstrating that SwitchFinder accurately selects against highly structured RNA elements such as transfer RNAs and ribosomal RNAs. Having multiple low energy minima in the landscape is not unique to riboswitches. I fear the algorithm has overfit the bacterial riboswitch class.

Response to 2.3: In response to the concern about multiple low energy minima not being unique to riboswitches, we agree with this observation. SwitchFinder is designed to identify RNAs with such minima, which encompasses riboswitches and other RNA classes like tRNAs known for conformational switching (Chan et al. 2020). Recognizing the potential for identifying false positives due to this characteristic, we implemented downstream *in vivo* functional screens in our methodology. These screens are essential for filtering out RNAs that, although possessing multiple conformations, do not show divergent regulatory functions between them. In other words, SwitchFinder is only the first step in our framework.

In response to the concerns about potential overfitting to the bacterial riboswitch class and the capability of SwitchFinder to differentiate against highly structured RNAs, we extended

our testing as suggested by the reviewer. We evaluated SwitchFinder's performance on different RNA classes including ribosomal RNAs, bacterial riboswitches, eukaryotic riboswitches, and synthetic riboswitches. As outlined in our response to comment 2.1, we approached this by providing the software with sequences containing the elements of interest and their shuffled counterparts, assessing its ability to distinguish between these two groups. To quantify this performance, we employed the auROC metric.

Our results revealed a markedly higher performance for bacterial riboswitches compared to ribosomal RNAs (auROC values of 0.63 and 0.54, respectively, **Figure R.6A**). Notably, SwitchFinder demonstrated even better performance on eukaryotic and synthetic riboswitches than on bacterial riboswitches, which is significant considering that the initial training focused solely on bacterial riboswitches (**Figure R.6A**). An example of this capability is the identification of a theophylline riboswitch by SwitchFinder, highlighted in **Figure R.6B**.

These findings support our assertion that SwitchFinder is not narrowly tailored to bacterial riboswitches but is proficient at identifying riboswitches across different categories. Furthermore, it effectively distinguishes non-switching, highly structured RNAs from actual riboswitches, addressing the reviewer's concern about overfitting to a specific riboswitch class.

Comment 2.4: Line 167: 14% of the predicted switches demonstrated differences from the scrambled control. This is a low accuracy for the algorithm presented. As a comparison, the authors should present an experiment cloning a SwitchFinder predicted "non-RNA-structural switch" downstream of eGFP and compare it to its scrambled control, would there be a difference in expression between the two? I assume there would be a difference in expression at the thresholds chosen.

Response to 2.4: We thank the reviewer for this comment and the opportunity to clarify a potential misunderstanding regarding the performance of SwitchFinder. It is important to note that the algorithm is designed to identify RNA elements that have the potential to switch between two structural conformations. However, it does not explicitly predict whether these elements will impact gene expression. This distinction is crucial, as the capability to switch conformations does not inherently imply a switch in functional effect on gene expression. Therefore, the downstream functional screening is a vital part of our methodology, necessary to differentiate between mere structural switches and those that are functionally significant in regulating gene expression. The 14% figure reflects this layered approach, highlighting the need for further experimental validation to confirm the functional impact of predicted switches. This is why we emphasize that the algorithm alone is not sufficient to definitively annotate a given fragment as a functional RNA switch.

Comment 2.5: Line 233: Which features are suggestive of RNA secondary structure? The authors need to translate the 3D classes into structures using the Leontis-Westhoff

classification. As far as this reviewer is aware, no study has ever demonstrated that different 3D Cryo-EM classes translate into different RNA secondary structures.

Response to 2.5: Specific base pairing chemistries unfortunately cannot be resolved in the structures presented in this manuscript, which are limited to identification of certain secondary structure features such as helical grooves or hairpins, and discrimination of overall 3D conformation. Groove and hairpin secondary structure features are highlighted in the braided appearance of the 2D classes in Extended Data Figure 5. A-C, and the idealized 3D structures from PDB shown with the 3D class densities in the new Extended Data Figure 4 D-F. The different 3D structures we present do appear to differ in secondary structure content, for example class A appears to represent a helical segment with a central turn leading to a “paperclip” like shape, while class B is consistent with two short helical segments surrounding the central compacted loop elements.

Comment 2.6: Line 269: The experiment with antisense oligos needs to be presented in comparison to untransfected cells. Antisense oligos activate the RNase H degradation pathway thus I highly doubt that addition of an antisense oligo that targets this region will activate gene expression by altering the conformation of the mRNA.

Response to 2.6: We appreciate the reviewer’s comment regarding the use of antisense oligos (ASOs) in our experiments. To address this concern, we would like to clarify that we utilized 2'-O-methylated (MOE) oligonucleotides and locked nucleic acids (LNAs), rather than gapmers. MOEs and LNAs do not activate the RNase H degradation pathway. They function by base pairing with complementary sequences, thereby specifically blocking the function of targeted regulatory elements (Kauppinen, Vester, and Wengel 2005; Kurreck et al. 2002).

In response to the reviewer’s suggestion, we assessed the impact of ASOs targeting the RORC switch on the stability of endogenous RORC mRNA. Following the introduction of these ASOs, we observed significant shifts in mRNA stability, as detailed in **Figure R.7**. These effects were more pronounced in cell lines with a higher occurrence of conformation 2 (LNCaP $P=0.006$, MCF-7 $P=0.005$) than in those with a lower occurrence (LS174T $P=0.71$). This finding aligns with our hypothesis that ASOs can shift the equilibrium between RNA conformations, moving it away from the inhibitory conformation 2 towards conformation 1. Thus, our results support the hypothesis that the relative proportions of these RNA conformations are intrinsically linked to mRNA stability in the endogenous setting.

Figure R.7. The effect of ASOs targeting one conformation of the RORC element on expression of the endogenous RORC gene across 3 cell lines. Only the cell lines with the high proportion of conformation 2 (LNCaP, MCF-7) demonstrate a significant change.

Comment 2.7: Line 265: The authors claim in the preceding paragraph that the alternative RNA conformations play divergent functional roles but they then present mutagenesis data that "all three mutants lower eGFP expression." Favoring one of the alternative conformations should increase eGFP expression and it did not. Can the authors explain this?

Response to 2.7: We thank the reviewer for this comment, which allows us to clarify a misunderstanding regarding our findings presented on line 265. In that section of the manuscript, we state, "In all three cases, we observed lower eGFP expression of the conformation 2 mutant (117-AC), as compared to (77-GA) which favors conformation 1 (Fig. 5B)." This statement was not meant to imply that all mutations led to a decrease in eGFP expression compared to the wildtype sequence. The point we intended to convey is that mutations favoring conformation 1 are associated with higher GFP expression levels compared to those favoring conformation 2. It should also be emphasized that while these mutations are intended to change the conformation with minimal impact on the primary sequence, they nevertheless do change the sequence and may impact other regulatory interactions. Therefore, we expect a general trend but for individual cases, other confounders may need to be considered.

Minor

Comment 2.8: Line 44: How can the authors claim that ribozymes and riboswitches are the oldest known groups of RNA-based regulatory mechanisms? There needs to be a citation or reasoning for this. Are these mechanisms older than transcription per se, or older than the alternative sigma factors or ribonuclease decay pathways? Older than transcription termination? This claim is unfounded.

Response to 2.8: We appreciate the reviewer's attention to detail and agree that our original statement could benefit from more precise phrasing. While our reference to ribozymes and riboswitches as some of the oldest known RNA-based regulatory mechanisms is informed by the RNA world hypothesis, we acknowledge that the comparative age of these mechanisms in relation to others is a complex and debated topic. The RNA world hypothesis suggests that RNA molecules, particularly ribozymes, predate both DNA and proteins in evolutionary history (Gilbert 1986). Riboswitches are also thought to have emerged early, potentially before DNA (Saad 2018). Moreover, Vitreschak et al. (2004), in their review titled "Riboswitches: the oldest mechanism for the regulation of gene expression?", provide substantial evidence supporting the antiquity of riboswitches (Vitreschak et al. 2004). We have revised the manuscript to reflect a more nuanced view and included these citations for clarity and reference. Additionally, we have softened our initial claim in the manuscript (lines 45-47) to better align with the current understanding and ongoing discussions in the field.

Comment 2.9: Line 45: Please change the words "RNA switches" to "Riboswitches." They are not the same.

Response to 2.9: We thank the reviewer for highlighting an area in our manuscript that required clarification. In response, we have included a specific definition of what we mean by "RNA switches" in the current manuscript (lines 49-51) to ensure clear communication of our intended scope.

Riboswitches are traditionally defined as "structured noncoding RNA domains that selectively bind metabolites and control gene expression" (Breaker 2012). However, our study's scope extends beyond this classical definition. We aim to identify RNA domains that regulate gene expression through conformational changes triggered not only by metabolite binding but also by interactions with other trans-acting factors, such as RNA-binding proteins, exemplified by the VEGFA switch (Ray et al. 2009). In the current study, we define an "RNA switch" as an element that adopts two mutually exclusive conformations, each leading to different gene-regulatory outcomes.

Comment 2.10: Line 48: How can the authors claim that bacterial riboswitches are one of the most widely observed mechanisms for gene expression control? The authors either need to present a citation or some experimental data supporting this.

Response to 2.10: We thank the reviewer for their comment. To substantiate our assertion regarding the prevalence of riboswitches in bacterial gene expression control, we have cited (Sun et al. 2013). Figure 7 in this study illustrates the fractions of genes controlled by riboswitches across the metabolic pathways. Notably, riboswitches control a large fraction of genes involved in coenzyme metabolism, uptake, amino acid metabolism across all the studied taxonomic groups. We believe this citation provides adequate support for our claim.

Comment 2.11: Line 67: Please change the word "showed" to "hypothesize."

Response to 2.11: We appreciate the reviewer's suggestion. We have clarified the phrasing in the revised manuscript.

Comment 2.12: Line 150: I am not aware of any data to support the notion that if multiple conformations co-exist, they all contribute the reactivity profile. The DMS modification rate depends on solvent accessibility, protein binding, R-loop formation, reverse transcription read-through, sequence context, and other factors, not on structure only. Many poorly resolved nucleotides in DMS or SHAPE-seq experiments occur genome-wide, they are low probability modification sites which are unable to be accurately called for any number of reasons, not RNA switches.

Response to 2.12: We are grateful for the reviewer's comment and the opportunity it provides to clarify a potential misunderstanding about the SwitchFinder algorithm's functionality.

Firstly, regarding the concept that "if multiple conformations co-exist, they all contribute to the reactivity profile," experimental evidence supporting this has been presented in studies by (Morandi et al. 2021; Tomczko et al. 2020). In these studies, RNAs were experimentally folded into alternative conformations, mixed, and then analyzed using DMS-seq. The authors successfully deconvolved the reactivity profiles into relative proportions of the alternative conformations, demonstrating that multiple conformations can indeed contribute collectively to the reactivity profile.

Secondly, it's crucial to note that SwitchFinder does not solely rely on DMS or SHAPE-seq reactivity profiles for making predictions. As the reviewer rightly points out, reactivity profiles obtained from high-throughput DMS and SHAPE-seq experiments often exhibit poorly resolved signals at individual nucleotides. Therefore, in our approach, DMS reactivity profiles are primarily used to filter out false positive predictions. These include RNA elements predicted *in silico* to occupy multiple conformations but not corroborated by their *in vivo* DMS reactivity profile. We have made revisions in the Results section of our manuscript to better emphasize and clarify this methodological approach.

Comment 2.13: Line 197: The authors need to explain how the mutations were designed. Covariance mutations should be used in this case.

Response to 2.13: We thank the reviewer for this suggestion. We have indeed used the covariant pairs of mutations. We have clarified this point in the revised manuscript.

Comment 2.14: Line 240: Change the word "evinced" to "evidenced"

Response to 2.14: We thank the reviewer for this suggestion. We have corrected this in the revised manuscript.

Comment 2.15: Line 242: The authors claim that the low-resolution models presented are sufficient for making predictions of RNA fold and handedness. How is this possible? The authors need to show their structure probing data in a coherent structural model that also corresponds to the 3D states of the same RNA molecule. How can you state that Class B represents one confirmation or another?

Response to 2.15: We thank the reviewer for this comment and for the opportunity to clarify this aspect of our study. Our claim about these models hinges on their ability to differentiate various RNA conformations and consistently identify these conformations across different samples. While these models do not offer high-resolution structural details, they are adept at discerning distinct conformational states, which is essential for our analysis. For instance, we identified three classes - A, B, and C - in the wildtype sample, while only classes A and C were present in the 77-GA sample. Given that our structural probing data suggests the 77-GA mutation shifts the equilibrium away from conformation 2, we hypothesize that class B likely represents this conformation. To directly address the comment, we employed DRAFFTER to fit our structural model to the observed structures, as shown in **Figure R.9**.

Regarding the handedness of the RNA structures, we should clarify that our approach allows us to determine the relative handedness due to the chiral shapes observed in these low-resolution models. However, we acknowledge that determining the absolute handedness would indeed require a higher level of structural detail, specifically the assignment of strand direction, which is beyond the capability of our current modeling approach.

In summary, while our models are not high-resolution, they are effective in consistently identifying distinct RNA conformations.

Comment 2.16: Line 257: Two-fold difference in relative expression is very low. Most bacterial riboswitches regulate gene expression on the order of 5-10-fold activation or repression. One cannot rule out here that there is a sequence effect on the UTR.

Response to 2.16: We thank the reviewer for pointing out the concern regarding the fold change in relative expression. In response, we extended our analysis to ensure that the expression changes observed are primarily attributed to changes in the RNA structure, rather than sequence effects. This involved examining two regions within the RORC element, termed "box 2" and "box 3" (see Fig. 3A), which are predicted to compete for base pairing with "box 1."

Applying the DMS-seq assay across five different cell lines with diverse genetic backgrounds (LNCaP, MCF-7, HepG2, ZR-75-1, 293T, and LS174T), we assessed whether an

anticorrelation existed between the accessibility of boxes 2 and 3, indicative of competitive base pairing. Our findings confirmed this anticorrelation ($R=-0.81$, **Figure R.8A**), suggesting that these regions are indeed competing for base pairing.

Additionally, we measured the stability of RORC mRNA across these cell lines using RT-qPCR. We observed a very strong correlation ($R=0.96$, $P=0.004$) between the relative proportions of the conformations and mRNA stability (**Figure R.8B**). This finding provides additional evidence that the balance between the RORC switch conformations is directly linked to the stability of RORC mRNA. Notably, the difference in RORC mRNA stability between LNCaP and LS174T cell lines is greater than 5-fold.

Figure R.8. (A) Accessibility of "box 2" (X axis) and "box 3" (Y axis) regions of RORC element across cell lines, as measured by DMS-seq. **(B)** Relationship between the relative conformation ratio of RORC element (X axis) and stability of the endogenous RORC mRNA, as measured by RT-qPCR.

Comment 2.17: Line 260: Since no control data presented from a non-switch 3' UTR, one cannot conclude that the two conformations play divergent functional roles.

Response to 2.17: We thank the reviewer for their comment. It is important to clarify why the suggested analysis involving a non-switch 3' UTR is not applicable in our study context. Our research specifically focuses on a mutational analysis of a bi-stable RNA switch within the RORC element. This involves using sequence mutations to shift the equilibrium between two alternative conformations and then assessing the impact of these shifts on reporter expression.

A non-switch 3' UTR, by its very nature, does not possess multiple conformations, and thus any mutations introduced would not yield interpretable effects on its secondary structure landscape in the same way as they do for a bi-stable switch. Consequently, comparisons with a non-switch 3' UTR would not provide meaningful insights into the divergent functional roles of the two conformations in the RORC element. The experimental design we employed, focusing on mutated and wild-type sequences of the RORC element, is specifically tailored to investigate the distinct regulatory functions of its two conformations, as depicted in Fig. 3A.

Comment 2.18: Line 376: “Both approaches were used to discover the first known RNA switches in bacteria...” Cite here Mironov et al (2002) PMID: 12464185 too, as this paper indeed reports the first two known riboswitches before the term “riboswitch” has been adopted.

Response to 2.18: We thank the reviewer for this comment. We cite (Mironov et al. 2002) in the revised manuscript.

Reviewer #3:

Remarks to the Author:

Summary of the key results

Although ligand-dependent conformational switching is prominent in the transcripts of numerous bacterial RNAs – with >55 validated classes of riboswitches — the extent to which structural switches control gene expression in eukaryotes remains largely unexplored. Here, Khoroshkin and co-workers used a massively parallel approach to identify a large number of human regulatory elements that adopt two conformations, and behave as gene-regulatory RNA switches. The findings suggest widespread conformation-dependent control of gene regulation in the human transcriptome.

Major

Originality and significance

A major innovation is the development of the methodology called Switchseeker, which is likely applicable to many other switches beyond the human transcriptome and test cases presented. The authors also give compelling evidence that specific regulatory elements adopt

two (or more) conformations, along with a plausible mechanism of action by which the upstream gene is regulated.

Clarity and context

Comment 3.1: Introduction. In the Introduction, the authors describe RNA switches as elements that control gene expression by direct binding of a small-molecule ligand or other trans-acting factor. Importantly, this is the definition of a riboswitch. A better working definition for this study is a 'switch RNA' that adopts two mutually exclusive conformations that lead to different gene-regulatory outcomes. This early differentiation of RNA switches is essential because the average reader will be confused that the RNA switches of this study are not actually riboswitches. It is better to define riboswitches as a specialized case of an RNA switch.

Response to 3.1: We appreciate the reviewer's suggestion to refine our definition of RNA switches in the Introduction. As correctly pointed out, riboswitches are traditionally defined as "structured noncoding RNA domains that selectively bind metabolites and control gene expression" (Breaker 2012). However, the scope of our study extends beyond this classical definition, aiming to identify RNA domains that regulate gene expression through conformational changes triggered not only by metabolite binding but also by interactions with other trans-acting factors, such as RNA-binding proteins exemplified by the VEGFA switch (Ray et al. 2009).

Therefore, we agree with the reviewer's recommendation to categorize riboswitches as a specialized subset of RNA switches. This broader perspective acknowledges the diversity of mechanisms through which RNA can regulate gene expression. We have revised the wording in the introduction to emphasize this distinction (lines 45-51), thereby clarifying our approach and ensuring it aligns with the broader definition of RNA switches.

Comment 3.2: Introduction. The authors also state, "The search for such ligand-binding riboswitches in eukaryotes has had limited success to date. Just two human examples are known: the RNA switch in VEGFA and the m6A modification-based switches (Liu et al. 2015; Ray et al. 2009)." This sounds like the VEGFA and m6A switches are ligand-sensing RNA switches, which is not correct based on the accepted definition of a riboswitch. As the authors know, eukaryotes have been shown to possess riboswitches that sense TPP (in plants and fungi), and these must be included if the authors wish to point out ligand-sensing RNA switches in eukaryotes.

Response to 3.2: We thank reviewer for this suggestion. We have updated the introduction, referencing the TPP-sensing riboswitches in plants and fungi, and clarified the description of VEGFA and m6A switches.

Comment 3.3: Results. At the start of the Results, the authors should define what they mean by “RNA structural switches”, which will be the premise of the paper. As worded, this is vague. In particular the authors should define that there are two mutually exclusive conformations that are probably nearly isoenergetic in folding but may be perturbed to adopt a conformation that stabilizes or destabilizes the upstream transcript leading to changes in gene expression. In other words, a cogent working definition is needed here.

Response to 3.3: We thank reviewer for this suggestion. We have revised the Results section and included the working definition of “RNA structural switches” (lines 49-51).

Comment 3.4: Results. Although a minor point, the authors describe DMS-seq as “single-nucleotide resolution.” However, some qualification is needed here because the method does not give a full picture of the modification landscape since only A and C are modified.

Response to 3.4: We thank reviewer for this comment. We have clarified this description in the revised manuscript.

Comment 3.5: Results. The authors state, “The accessibility of a single nucleotide is a population average of multiple RNA molecules that represent different minima in the RNA folding conformation ensemble.” This statement should mention different “Gibbs free energy minima.”

Response to 3.5: We thank reviewer for this comment. We have revised the wording of this sentence as suggested.

Comment 3.6: Results. In the second iteration of SwitchSeeker, how did the application of chemical modification data improve the results that led to a high confidence set of RNA switches? Did the chemical probing data help to better define the accuracy of regions that adopt two mutually exclusive conformations? Or did it help to eliminate problem sequences that do not adopt two clear switch conformations based on RNA structure prediction alone? Or both? Some clarification would be appreciated. A major point for the user is that either approach would succeed in the workflow, but what are the pros and cons of each?

Response to 3.6: We thank reviewer for this comment. *In silico* RNA secondary structure predictions often face limitations, such as a short folding window and overlooking interactions with RNA binding proteins and miRNAs. By incorporating chemical modification data into SwitchSeeker, we significantly enhance the precision of our RNA structure predictions. This improved accuracy primarily aids in eliminating sequences that do not demonstrate two clear switch conformations *in vivo*. We have clarified this point in the revised manuscript (lines 185-189).

Comment 3.7: For the TCF7 RNA switch in Fig 2D, what was the difference in Gibbs free energy for the two predicted conformations? This information will be interesting to the reader.

Response to 3.7: We thank reviewer for this comment. We updated the caption for Figure 2D to reflect the difference in Gibbs free energy.

Suggested improvements/Data & methodology

Comment 3.8: Cryo-EM Results. The reviewer disagrees that the 2D class averages provide sufficient evidence to recognize the RNA fold and handedness. No cogent evidence is provided to support this statement.

Response to 3.8: 2D class averages (Extended Data Fig. 5A) are presented to convey a sense for the nature and quality of the raw cryo-EM data. Therefore we assume the reviewer intended to refer to the 3D reconstructions. The structures of the 3 classes are quite distinct, and are recapitulated by reference-free methods within multiple datasets (Extended Data Fig. 5G-I). This ability to distinguish differences in the overall fold or tertiary structure of the RNA is key to 1) confirming the switch adopts alternate tertiary structures as predicted, and 2) using a mutational analysis to assign structures to the predicted states of the switch.

The relative handedness, e.g. of similar classes from different samples, can be distinguished because the structures have chiral features such as the bending direction of the central protrusion in class B or the angle of the bent stem loop in class C. Although there are specific features that appear to be expected RNA secondary structure elements, assignment of absolute hand requires confident assignment of strand direction which is not currently possible.

In response to the reviewer's comments we have made a number of changes to the cryo-EM section to clarify our use of cryo-EM in this study, reflecting the above points.

Comment 3.9: Specifically, the various particles do not appear to contain the amount of RNA attributed to the folds produced from chemical modification.

Response to 3.9: Although very flexible elements of macromolecules may be partially resolved in cryo-EM maps, and which might be expected for a structurally heterogeneous RNA, it does appear possible for the 143 base constructs to fit in the maps. Please see our discussion of modeling below.

Comment 3.10: Only partial models were docked into the potential maps providing an incomplete analysis of the putative RNA conformation.

Response to 3.10: The partial models (now in Extended Data Fig. 4D-F) are visual aids to show that specific structural features evident in the 3D reconstructions match what is expected for RNA (e.g. the major groove). They are model sequences with crystal structures from PDB, not models of the RORC RNA. Again please see below regarding atomic models.

Comment 3.11: The addition of other evidence is needed to support the composition of the particles, such as SEC-MALS. This is necessary because the class A particle appears much larger than class C and appears to accommodate twice as much RNA. Even at the stated resolution of ~ 10 Å, it should be feasible to model a secondary structure based on chemical modification. This aspect of the study was one of the weakest.

Response to 3.11: Initially, we had the same impression as the reviewer, and we undertook to model the structures using DRRAFTER. Unfortunately, some critical limitations became apparent as we applied this state-of-the-art software that ultimately led to low confidence in the output models such that we lacked sufficient conviction to include them in the first submission of the paper. First is the need for partial starting models, which must include manually positioned strand termini as well as idealized coordinates for secondary structure elements identified by DMS-seq. Another is the use of explicit secondary structure constraints (that is, base pairing constraints), so that for each secondary structure cluster obtained from the DRACO pipeline an independent line of modeling must be conducted. Nevertheless, we attempted to use both manually constructed partial models, as well as complete models obtained from the RNAcomposer server, within DRRAFTER. These efforts led us to discover the final, most severe issue with the pipeline: almost any starting model could be compacted sufficiently to fit into the cryo-EM envelopes.

The **Figure R.9** below shows pairs of example low-energy models from DRRAFTER runs for class B and C structures, with the indicated elements as initial guess. Even assuming class assignments based on the mutational analysis discussed in the paper are correct, the need to consider multiple possible DRACO cluster representatives, ~ 10 possible initial models per structure, and two handedness possibilities, each producing thousands of “decoys” (Rosetta terminology for scored models), represents a very significant effort likely to produce only low-confidence results. The models do have some use, however, in that they show that given varying levels of compaction during RNA folding, both smaller and larger class densities actually can accommodate the full 143 base RNA molecule.

Figure R.9. Pairs of high-scoring models created by DRRAFTER for WT 3D classes B and C with density overlaid. The pre-positioned, idealized RNA structures used as initial models are indicated by a bracket. Although the individual models are of low-confidence, they demonstrate that the class densities likely represent all or the majority of the RNA molecule.

Our experience suggests that the current state-of-the-art tool for RNA structure prediction/modeling remains best suited to stable, conformationally homogeneous samples in the context of relatively high-resolution structural data. This observation is borne out by the recent literature of RNA-only cryo-EM structures, the vast majority of which exhibit sufficient resolution (~3.5 Å) that manual building of the entire molecule is possible even without sophisticated computational approaches.

In contrast, in this study we looked intentionally for RNAs with significant conformational heterogeneity, and we ask the reviewer to consider that given current methods this limits the structural data to the key role described above. In the revised manuscript, the DRRAFTER models shown above have been included in Extended Data Figure 4.

Comment 3.12: For the cryo-EM analysis, a major concern is whether the samples used form a self-complementary dimer. This could be analyzed to examine the free energy difference between the dimer and monomer to support their understanding of the cryo-EM particle compositions. SEC-MALS analysis could be used to support this assignment.

Response to 3.12: The possible formation of multimeric RNA complexes is an important consideration highlighted by the reviewer. As shown above, however, the classes we have identified accommodate the full-length RNA without any room (or need) for dimerization. We also approached this question experimentally through paired denaturing and native gels

(**Figure R.10**). Native PAGE revealed a single major band, which was not dependent on refolding concentration.

Figure R.10. The wildtype RORC element RNA was synthesized *in vitro*, refolded at various concentrations (2.2 μ M and 22 μ M, respectively) as per the cryo-EM samples, and analyzed on denatured and native gels.

Comment 3.13: Notably, information from the cryo-EM section was absent in the reviewer's version of the manuscript: (i) Table S8 described in the Methods was not included in this manuscript submission; there does not appear to be any previous Tables either in the Supplemental Information. (ii) Table S7 described in the Methods was not included in this manuscript submission. (iii) Extended Data does not appear to have been included in this submission. Please correct these call outs and provide the information as needed.

Response to 3.13: We apologize that the uploaded SI was not made available, we have asked the editor to ensure they will have access to the material.

DMS-MapSeq and cryo-EM data suggest that the RORC 3' mRNA element inhabits a shallow energy landscape with two rugged minima linked to two major molecular conformations

Comment 3.14: In the Discussion the authors state that they have provided experimental structures of switch states, which validates the SwitchSeeker approach to identify RNA molecules with of bistable energy landscapes. The reviewer believes this is an overinterpretation of the cryo-EM data. The structure is not really determined until an atomistic model is fit into the potential maps. The authors present small pieces of RNA structure docked into their molecular envelopes in Supp Fig. 4, but the complete models are not shown. With DMS-seq data and the molecular envelopes, it seems reasonable that models could be made even at ~ 10 Å. This would be much more supportive of the claim that experimental structures were determined, and that the RNA adopts the conformations claimed.

Response to 3.14: Some biological macromolecules are not conducive to determination of high-resolution structures, because of their intrinsic compositional or conformation heterogeneity. One strength of single-particle cryo-EM is the ability to solve low resolution structures first, and subsequently progress to higher and higher resolutions, until such intrinsic limits are reached. These structures are consistent with one another; although low resolution structures may lack *precision*, with current image processing methods they are highly likely to be *accurate*.

In the absence of complete, high-resolution structures for the RORC RNA, we employed mutational analysis to probe the relationship between RNA conformation and a functional output. We agree with the reviewer that low resolution structures from a single sample are of little utility aside from suggesting the presence of compact, folded RNA. However, our comparison of the differing gross conformations observed in WT RNA and mutants predicted to bias RNA conformation, along with the litany of additional RNA probing data provided, are strong evidence for our overall model of a multistable energy landscape explaining mutant functional effects.

Comment 3.15: A better approach to interrogate the thermodynamic ensemble is SAXS. In this instance, all three conformations could be identified in solution and compared to the particles restored by cryo-EM. AFM has also seen recent advances in the ability to monitor particles in the thermodynamic ensemble [Ding et al. Wang 2023 Nat Comm 14, 714].

Response to 3.15: SAXS data must be combined with simulated/predicted structures (e.g. from molecular dynamics) via a forward scattering model, in order to potentially derive low-resolution particle shape envelopes. SAXS is also a bulk method, and interpretation would be greatly complicated by the potential for fast exchange as raised by the reviewer below, as well as by the known sample heterogeneity. It may be possible in principle to deconvolute SAXS curves for mixed species in some cases, but there does not appear to be any application of this concept to RNA in the literature.

We do agree with the reviewer that AFM shows promise in RNA structure analysis, however it is important to underline that the structures shown in the cited paper are only previously determined x-ray crystal structures from PDB. The authors thus describe their models, derived from manual sorting of 2D images, as “recapitulated” and not “solved” for that reason.

In contrast to current SAXS and AFM approaches, single-particle cryo-EM is able to produce *de novo* structures of the heterogeneous RORC RNA using well-established analysis methods and without exogenous reference information of any kind. We argue that despite the limitations of SPA, purely empirical, reference independent structure determination is an important strength given the risks of model overfitting described above.

Comment 3.16: A main concern with cryo-EM is that not all particles observed give rise to the class averages used for potential map calculations.

Response to 3.16: Several of us have been thinking deeply on the important issue raised here for many years, and as a rule, the current authors do not treat cryo-EM particle counts as quantitative representations of a free energy landscape. The power of cryo-EM is rather in providing a snapshot of the stable or metastable structures existing in the sample, including qualitative differences such as the complete absence of a particular state.

Conclusions: robustness, validity, reliability

Comment 3.17: One of the hallmarks of riboswitches is the presence a consensus model for a particular switch in multiple species. Are there similar elements like RORC RNA non-human primates or metazoans that suggest comparable switching of the RORC transcript?

Response to 3.17: We appreciate the reviewer's query about the presence of RORC RNA switch-like elements in non-human primates or metazoans. Indeed, the exact sequence of the RORC element (GTCTTCTACCACTAGAAGACCCAAGAGAAGCAGAAGTCGCTCGCACTGGTCAGTCGGAAGGC) is found in the 3'UTR of the RORC gene in non-human primates, including Gorilla gorilla and Pan paniscus. While this sequence conservation across species might imply the existence of a similar RORC switch mechanism, it is important to note that confirming such a hypothesis would require additional experimental verification. This aspect of cross-species functional conservation falls outside the scope of our current study.

Comment 3.18: As noted by the authors, UPF1 has been reported to be part of a structure-mediated decay pathway (Fischer et al. 2020). This mechanism of action is plausible here but is differentiated by the requirement for the dsRNA binding protein G3BP1, which is not associated with NMD factors. Is there any evidence that G3BP1 knockdown restores transcript levels as shown for knockdowns of the core SURF complex? This has implications for the model proposed in Fig. 7. Some discussion of this structure-mediated decay mechanism is needed.

Response to 3.18: We appreciate the reviewer's comment and the opportunity to clarify our position on the mechanism of RORC mRNA degradation. Contrary to suggesting that RORC mRNA undergoes NMD-independent degradation mediated by UPF1, as described in (Fischer et al. 2020), our hypothesis aligns with an Exon Junction Complex (EJC)-independent NMD pathway, as described in (Kurosaki, Popp, and Maquat 2019). To directly test this, we investigated the impact of NMDI14, an inhibitor of NMD that disrupts SMG7-UPF1 interactions, on the equilibrium between the two conformations of the RORC element. Using DMS-seq, we measured the accessibility of regions 2 and 3 of the endogenous RORC mRNA. We observed a significant decrease in the accessibility of region

2 following NMDI14 treatment ($P=0.03$, **Figure R.11**), which aligns with the expected slower decay and accumulation of mRNAs in conformation 2.

Furthermore, we would like to direct the reviewer's attention to **Fig. 6C**, which demonstrates the reduction in RORC degradation following the knockdown of key NMD factors, including UPF2, UPF3B, SMG5, SMG8, and SMG9. This evidence supports our hypothesis that RORC degradation is dependent on the core components of the NMD pathway. We have revised the manuscript to better highlight and clarify this hypothesis.

Figure R.11. The effect of NMDI14 on the accessibility of "box 2" and "box 3" regions of RORC element.

Comment 3.19: As noted by the authors, a lingering question for the switch is what factors influence the ability to partition into each gene-regulatory state? As the authors know, riboswitches provide feedback by adopting one of two mutually exclusive conformations that depend upon the concentration of a cognate cellular effector. Do the authors envision differences in the protein expression landscape that promote one conformation over another? Some discussion of known RNA switches, such as VEGF, would be appropriate.

Response to 3.19: We are grateful for the reviewer's insightful question about the factors influencing the partitioning of RNA switches into different gene-regulatory states. We concur that identifying the specific effectors that trigger riboswitches to adopt one conformation over another is crucial for understanding their biological function.

We hypothesized that variations in the protein expression landscape could influence the balance of conformations, thereby affecting gene expression levels. To test this, we analyzed

the relative proportions of the two alternative conformations of endogenous RORC mRNA across five cell lines representing diverse genetic backgrounds: LNCaP (prostate), MCF-7 (breast), HepG2 (liver), ZR-75-1 (breast), 293T (kidney), and LS174T (colon). We observed that the relative proportions of the conformations varied depending on the genetic background (**Figure R.12A**). We then examined the stability of RORC mRNA across these cell lines. We observed a strong correlation between the stability of endogenous RORC mRNA and RNA conformation ratios ($R=0.96$, $P=0.004$, **Figure R.12B**). This suggests that the RORC element connects its host mRNA stability to changes in the protein expression landscape.

The data presented here strongly support the hypothesis that the RORC element is responsive to changes in the protein expression landscape. While our study lays the groundwork, future research is anticipated to pinpoint the specific effector of this RNA switch.

Figure R.12. (A) Accessibility of "box 2" (X axis) and "box 3" (Y axis) regions of RORC element across cell lines, as measured by DMS-seq. Left: the accessibility was measured in the context of a GFP reporter containing the RORC element sequence. Right: the accessibility was measured in the context of endogenous RORC mRNA. **(B)** Relationship between the relative conformation ratio of RORC element (X axis) and stability of the endogenous RORC mRNA, as measured by RT-qPCR.

Comment 3.20: Are the switches described here under thermodynamic control or kinetic control? The use of DMS-seq and other equilibrium methods suggests the former, but this point is not discussed. The model in figure 7 suggests there is some kinetic barrier in the free-energy landscape that joins conformation 1 and 2. If this is true, the kinetics must be extraordinarily slow to allow identification by cryo-EM.

Response to 3.20: The reviewer raises an important point. We do not know what kinetics characterize the RORC mRNA conformational landscape. The freezing rate during cryo-EM sample prep is believed to be $\sim 10^5$ K/s, with vitrification lasting for some hundreds of microseconds. We submit that this may be slow for a macromolecule, but perhaps not

“extraordinarily” slow. For single particle measurements in a frozen glass such as in cryo-EM, a state can be resolved if sufficient examples exist in solution - even if grossly outnumbered by fast-exchange intermediates. This is in contrast to bulk methods such as SEC-MALS or native PAGE, and even slow single-molecule methods such as mass photometry, where fast exchange would lead to measurement of an average only. A full answer to this question could be an important contribution of a subsequent study focused explicitly on the biophysics of the RORC mRNA.

Comment 3.21: Wouldn't this be better described as a local, stable free energy intermediate? The energy diagram suggests it is high-energy but wouldn't this be separated by a high-energy barrier as shown but with a deeper local minimum?

Response to 3.21: While the energy landscape diagram is intended to be schematic given the absence of quantitative information, we agree with the reviewer that this may be a better description of how at least the monomeric RNA seems to behave. We have edited the figure to reflect the reviewer's suggestion of a deeper minimum for the proposed intermediate.

Comment 3.22: Is there any experimental evidence for the interconversion of any conformation to one of the other folds? For example, if the sample is run on gel filtration or a native gel, does a single isolated species repartition into all three states? At present, there is very little support for the proposed model (except that it explains a potential artifact of cryo-EM).

Response to 3.22: We thank the reviewer for their comment. To address the query about experimental evidence for the interconversion of conformations, we refer to the data presented in **Figure R.12A**. This figure demonstrates a strong anticorrelation in the accessibility of the conflicting stems (Box 2 and Box 3), supporting the hypothesis of interconversion between the two folds and substantiating our proposed model.

Comment 3.23: A major concern is that this paper is not written like a Nature Methods paper. The Methods are difficult to follow and there is no significant consideration that the end user would want to replicate the result or develop derivative approaches.

Response to 3.23: We appreciate the reviewer's feedback regarding the presentation style of our paper and the accessibility of our methods. Understanding the importance of replicability and the potential for further development of derivative approaches, we have taken steps to make our methodology more user-friendly. To this end, we have included a detailed Supplementary Protocol, specifically designed to guide users through the SwitchSeeker pipeline step by step. This protocol aims to provide clear instructions, ensuring that researchers can easily replicate our results and apply the SwitchSeeker framework to

their own studies. We believe that this addition will significantly improve the utility of our paper for end users interested in adopting and adapting our methods.

Minor

Comment 3.24: As a comment, it would be interesting to see if any of the identified sequences in the 3'-UTR show evidence for aptamer homology to known bacterial riboswitches. Riboswitch aptamers are the most conserved elements of these regulatory molecules, and one could envision how ligand-dependent folding could promote decay pathways in metazoans.

Response to 3.24: We appreciate the reviewer's suggestion regarding the exploration of potential homology between identified sequences in the 3'-UTR and known bacterial riboswitch aptamers. However, conducting a direct homology comparison between non-coding regions of metazoans and prokaryotes presents significant challenges. One of the main hurdles is the limited availability of accessible genomes from hominids with a sufficient degree of separation, which is crucial for accurately reconstructing the most conserved structured elements. This limited availability of diverse and separated genomic data, particularly in metazoans, likely contributes to the under-exploration of RNA switches in these organisms.

Comment 3.25: Is it known whether any of the RNA structures have been modified by post-transcriptional modifications that could modulate the structures? (Perhaps beyond the scope of this work)

Response to 3.25: We thank the reviewer for bringing up the potential impact of post-transcriptional modifications on RNA structures. Indeed, the role of such modifications in regulating gene expression, particularly in the context of RNA switches, has been underscored in studies such as (Liu et al. 2015). While it is plausible that the RNA structures we identified could be modulated by post-transcriptional modifications, exploring this aspect falls beyond the scope of our current work. Nonetheless, this remains an important area for future research to further understand the dynamics of RNA switches.

Comment 3.26: Several places in the Methods and Supplemental Information refer to the 'RNA switches' as "riboswitches". This is incorrect because the switches identified here do not conform to the definition of riboswitches, which respond to small molecules or ions as effectors of RNA conformation (e.g., Supple Fig. 2 and Methods section on CRISPRi).

Response to 3.26: We thank reviewer for this comment. We have corrected this in the revised manuscript.

Comment 3.27: There are several minor typos in the document that can be easily identified using the MSWord spellcheck tool. These are flagged by red underscoring.

Response to 3.27: We thank reviewer for this comment. We have corrected the typos in the revised manuscript.

Comment 3.28: Supple Fig 2. The authors state that DMS allows assignment of base pairing flexibility, but this modification is better correlated with WC-face accessibility rather than flexibility (which best describes SHAPE).

Response to 3.28: We thank reviewer for this valuable suggestion. We have updated the wording in the revised supplementary materials.

References: appropriate

- Breaker, Ronald R. 2012. "Riboswitches and the RNA World." *Cold Spring Harbor Perspectives in Biology* 4 (2). <https://doi.org/10.1101/cshperspect.a003566>.
- Chan, Clarence W., Deanna Badong, Rakhi Rajan, and Alfonso Mondragón. 2020. "Crystal Structures of an Unmodified Bacterial tRNA Reveal Intrinsic Structural Flexibility and Plasticity as General Properties of Unbound tRNAs." *RNA* 26 (3): 278–89.
- Fischer, Joseph W., Veronica F. Busa, Yue Shao, and Anthony K. L. Leung. 2020. "Structure-Mediated RNA Decay by UPF1 and G3BP1." *Molecular Cell* 78 (1): 70–84.e6.
- Gilbert, Walter. 1986. "Origin of Life: The RNA World." Nature Publishing Group UK. February 1986. <https://doi.org/10.1038/319618a0>.
- Kauppinen, Sakari, Birte Vester, and Jesper Wengel. 2005. "Locked Nucleic Acid (LNA): High Affinity Targeting of RNA for Diagnostics and Therapeutics." *Drug Discovery Today. Technologies* 2 (3): 287–90.
- Kurosaki, Tatsuaki, Maximilian W. Popp, and Lynne E. Maquat. 2019. "Quality and Quantity Control of Gene Expression by Nonsense-Mediated mRNA Decay." *Nature Reviews. Molecular Cell Biology* 20 (7): 406–20.
- Kurreck, Jens, Eliza Wyszko, Clemens Gillen, and Volker A. Erdmann. 2002. "Design of Antisense Oligonucleotides Stabilized by Locked Nucleic Acids." *Nucleic Acids Research* 30 (9): 1911–18.
- Lee, Flora C. Y., and Jemiej Ule. 2018. "Advances in CLIP Technologies for Studies of Protein-RNA Interactions." *Molecular Cell* 69 (3): 354–69.
- Liu, Nian, Qing Dai, Guanqun Zheng, Chuan He, Marc Parisien, and Tao Pan. 2015. "N(6)-Methyladenosine-Dependent RNA Structural Switches Regulate RNA-Protein Interactions." *Nature* 518 (7540): 560–64.
- Lynch, Sean A., Shawn K. Desai, Hari Krishna Sajja, and Justin P. Gallivan. 2007. "A High-Throughput Screen for Synthetic Riboswitches Reveals Mechanistic Insights into Their Function." *Chemistry & Biology* 14 (2): 173–84.
- Mironov, Alexander S., Ivan Gusarov, Ruslan Rafikov, Lubov Errais Lopez, Konstantin

- Shatalin, Rimma A. Kreneva, Daniel A. Perumov, and Evgeny Nudler. 2002. "Sensing Small Molecules by Nascent RNA: A Mechanism to Control Transcription in Bacteria." *Cell* 111 (5): 747–56.
- Morandi, Edoardo, Iliaria Manfredonia, Lisa M. Simon, Francesca Anselmi, Martijn J. van Hemert, Salvatore Oliviero, and Danny Incarnato. 2021. "Genome-Scale Deconvolution of RNA Structure Ensembles." *Nature Methods* 18 (3): 249–52.
- Nemoto, Kaoru, Andreas Vogt, Tetsuya Oguri, and John S. Lazo. 2004. "Activation of the Raf-1/MEK/Erk Kinase Pathway by a Novel Cdc25 Inhibitor in Human Prostate Cancer Cells." *The Prostate* 58 (1): 95–102.
- Ray, Partho Sarothi, Jie Jia, Peng Yao, Mithu Majumder, Maria Hatzoglou, and Paul L. Fox. 2009. "A Stress-Responsive RNA Switch Regulates VEGFA Expression." *Nature* 457 (7231): 915–19.
- Saad, Nizar Y. 2018. "A Ribonucleopeptide World at the Origin of Life." *Journal of Systematics and Evolution* 56 (1): 1–13.
- Sun, Eric I., Semen A. Leyn, Marat D. Kazanov, Milton H. Saier Jr, Pavel S. Novichkov, and Dmitry A. Rodionov. 2013. "Comparative Genomics of Metabolic Capacities of Regulons Controlled by Cis-Regulatory RNA Motifs in Bacteria." *BMC Genomics* 14 (September): 597.
- Tomezsko, Phillip J., Vincent D. A. Corbin, Paromita Gupta, Harish Swaminathan, Margalit Glasgow, Sitara Persad, Matthew D. Edwards, et al. 2020. "Determination of RNA Structural Diversity and Its Role in HIV-1 RNA Splicing." *Nature* 582 (7812): 438–42.
- Vezeau, Grace E., Lipika R. Gadila, and Howard M. Salis. 2023. "Automated Design of Protein-Binding Riboswitches for Sensing Human Biomarkers in a Cell-Free Expression System." *Nature Communications* 14 (1): 2416.
- Vitreschak, Alexey G., Dimitry A. Rodionov, Andrey A. Mironov, and Mikhail S. Gelfand. 2004. "Riboswitches: The Oldest Mechanism for the Regulation of Gene Expression?" *Trends in Genetics: TIG* 20 (1): 44–50.
- Wang, Xun, Can Fang, Yifei Wang, Xinyu Shi, Fan Yu, Jin Xiong, Shan-Ho Chou, and Jin He. 2023. "Systematic Comparison and Rational Design of Theophylline Riboswitches for Effective Gene Repression." *Microbiology Spectrum* 11 (1): e0275222.

Decision Letter, first revision:

Dear Hani,

Thank you for submitting your revised manuscript "A systematic search for RNA structural switches across the human transcriptome" (NMETH-A51594B). It has now been seen by the original referees and their comments are below. The reviewers find that the paper has improved in revision, and therefore we'll be happy in principle to publish it in Nature Methods, pending minor revisions to satisfy the referees' final requests and to comply with our editorial and formatting guidelines.

While referee 2 still had major concerns, upon discussing your rebuttal we decided editorially that additional benchmarking and validation are not strictly necessary to find the methods convincing. We therefore overrule these experimental requests in this case. We do however ask that you make changes to the text to address the remaining referee concerns (from all refs) as appropriate and provide a full point-by-point rebuttal upon resubmission.

TRANSPARENT PEER REVIEW

Please note: we allow redactions to authors' rebuttal and reviewer comments in the interest of confidentiality. If you are concerned about the release of confidential data, please let us know specifically what information you would like to have removed. Please note that we cannot incorporate redactions for any other reasons. Reviewer names will be published in the peer review files if the reviewer signed the comments to authors, or if reviewers explicitly agree to release their name. For more information, please refer to our FAQ page.

ORCID

Sincerely,
Rita

Rita Strack, Ph.D.
Senior Editor
Nature Methods

Reviewer #1 (Remarks to the Author):

The authors have provided a revised manuscript version, in which they have appropriately addressed all of my previous major concerns. I have noted the following minor points:

- 1) A more accurate description of the statistical tests is needed in several cases. For example, for Fig. 5F, 6E, H-I, one p-value is provided in charts with multiple bars without defining the comparisons. What kind of statistical test has been performed in case of Fig. 6E and Extended Data Fig. 7E? Are the data really based on two replicates each and p-values are $< 2e-16$?
- 2) Statement in l. 235-236 "Even though we did not observe a significant decrease in accessibility of the Box 3 upon the 77-GA mutation...". As no statistical analysis is provided, the word "significant" is misleading in this context.
- 3) Legend to Fig. 2B: shouldn't there be a definition of bin 1 to bin 8 for fluorescence, instead of bin 1 to bin 4?
- 4) Fig. 2C, legend says that candidate switches were mutated to lock them in either of the two conformations, and that "a sequence library is then generated (see Extended Data Fig. 2A)". Extended Data Fig. 2A refers to the structure screen by DMS-MaPseq. Does that mean that the conformation of the mutated switch sequences was tested by DMS-MaPseq or is the reference not correct?
- 5) Fig. 3B: the symbols for matching/non-matching base pairs are hard to distinguish, at least at this size.
- 6) Fig. 5C: the "+" symbol for the box2 mutant is incorrectly placed
- 7) Extended data figure 6: some further information or conclusion what these markers stand for would be helpful for readers who are not experts in Th17 cell differentiation.

Reviewer #2 (Remarks to the Author):

See report attached:

The authors here present an updated work defining meta-stable structures of RNAs in mammalian cell lines. This is an ambitious work including transcriptomic, DMS MapSeq, and Cryo-EM. They approach the problem of defining meta-stable structures elegantly with a novel program which can determine RNA structures with multiple low energy minima, separated by an energy barrier. This kind of combined *in silico*, *in vivo*, and Cryo-EM approach is novel and rarely seen. The authors clearly took on a significant amount of work cloning more than 3000 3'UTRs of mammalian genes. Their approach is sound and they went so far as to define a possible switch in the RORC 3' UTR.

I feel that for publication in Nature Methods, this methodology should be benchmarked rigorously against RNA switches which are known to function in mammalian cells. To use this in my lab, I would like to see that this method is robust enough for synthetic switches or at least UTR regions which are known to function as switches. Without this benchmarking, I cannot be certain that the methodology works as the authors propose. I look forward to seeing this work fully formed in the future.

Below I reiterate and clarify some of my points that have not been adequately addressed (in blue).

Reviewers' Comments:

Reviewer #2:

Comment 2.1: The authors present an original, innovative approach to discover RNA structural switches in eukaryotic cells. They combine a wide array of informatics, RNA structure probing, reporter gene assays, and even cryo-EM, which has not yet been applied to RNA switches to date. The methods are all novel. This kind of work can be a major methodological step forward in the field of RNA biology.

My main concern is that none of the methods presented have been benchmarked against known riboswitches in bacteria or even artificial switches and aptamers. As an example, the theophylline binding aptamer is one case which is widely applied to regulate gene expression in current eukaryotic model systems. I think that proper benchmarking of the methods presented is what is really lacking before publication as a methodological paper. There are a few other issues that the authors should consider:

Response to 2.1: We appreciate the reviewer's point regarding the need for thorough benchmarking of our method. However, unfortunately, we are limited by the lack of a proper dataset for benchmarking.

I think the authors are indeed actually attempting this with the theophylline riboswitch. Surely, there are others such as the GuaM8 aptazyme. Below, the authors even state that they "include a comprehensive set of theophylline riboswitches (Wang et al. 2023) and a collection of artificial protein-sensing riboswitches (Vezeau, Gadila, and Salis 2023)." If the authors cannot

benchmark your algorithm against known riboswitches, why not use a synthetic set as a control?

In our review of numerous studies on natural and artificial riboswitches, we found that detailed characterizations of secondary structures for the two alternative states are often missing. While many studies demonstrate the ability of riboswitch RNAs to modulate expression in response to a ligand, they typically do not provide structural details for both states.

If that is the case, then why not test your approach, which includes structural characterizations like DMS MapSeq and Cryo-EM, on at least one known artificial switch.

Additionally, the variety of experimental methods used across these studies makes compiling a single, high-quality dataset for benchmarking a significant challenge. To address this limitation, we benchmarked SwitchFinder by analyzing its performance on known riboswitch sequences versus their dinucleotide-shuffled counterparts. The premise is that while a shuffled sequence lacks a functional riboswitch, its degree of secondary structure formation remains broadly similar due to preserved dinucleotide content. Our results indicate that SwitchFinder effectively distinguishes real riboswitch sequences from their shuffled versions (Figure R.6A). In contrast, when tested with highly structured non-switching RNAs such as ribosomal RNAs, the algorithm showed limited differentiation capability between real and shuffled sequences (auROC = 0.54). This suggests that SwitchFinder is adept at identifying regions in RNA that likely contain conformational switches.

While I like the SwitchFinder approach, I do not think that the difference between an auROC of 0.54 versus 0.63 is significant.

Acknowledging the reviewer's suggestion, we have now extended our benchmarking to include a comprehensive set of theophylline riboswitches (Wang et al. 2023) and a collection of artificial protein-sensing riboswitches (Vezeau, Gadila, and Salis 2023). In these tests, SwitchFinder demonstrated an ability to differentiate artificial riboswitches from their shuffled counterparts, achieving auROC values of 0.74 and 0.69, respectively. We note that this computational step is intended to enrich for real switch elements that are then investigated through the MPRA measurement.

Why didn't the authors include a Theophylline switch or any control synthetic switch in the initial screen? The authors need to validate that the assay works. I do not think that RNAs in cells fold into meta-stable structures, that is why I have a hard time grasping the approach. The authors need to prove that meta-stable structures exist and function as switches. They compare their approach against bacterial riboswitches, which are NOT meta-stable structures, they are alternative stable conformations of the same RNA sequence.

Comment 2.3: Line 110: The authors must present data demonstrating that SwitchFinder accurately selects against highly structured RNA elements such as transfer RNAs and ribosomal RNAs. Having multiple low energy minima in the landscape is not unique to

riboswitches. I fear the algorithm has overfit the bacterial riboswitch class.

Response to 2.3: In response to the concern about multiple low energy minima not being unique to riboswitches, we agree with this observation. SwitchFinder is designed to identify RNAs with such minima, which encompasses riboswitches and other RNA classes like tRNAs known for conformational switching (Chan et al. 2020).

tRNAs are NOT known to switch between secondary structure conformations like RNAs that you are trying to discover in the paper. The paper you cite here demonstrates structural flexibility in tertiary structure on the order of angstroms, between two different crystal structures of the same tRNA. You need to give a better reason for why you can't benchmark SwitchFinder against other classes of RNAs.

In response to the concerns about potential overfitting to the bacterial riboswitch class and the capability of SwitchFinder to differentiate against highly structured RNAs, we extended our testing as suggested by the reviewer. We evaluated SwitchFinder's performance on different RNA classes including ribosomal RNAs, bacterial riboswitches, eukaryotic riboswitches, and synthetic riboswitches.

As outlined in our response to comment 2.1, we approached this by providing the software with sequences containing the elements of interest and their shuffled counterparts, assessing its ability to distinguish between these two groups. To quantify this performance, we employed the auROC metric. Our results revealed a markedly higher performance for bacterial riboswitches compared to ribosomal RNAs (auROC values of 0.63 and 0.54, respectively, Figure R.6A).

While I like the SwitchFinder approach, I do not think that the difference between an auROC of 0.54 versus 0.63 is significant.

Notably, SwitchFinder demonstrated even better performance on eukaryotic and synthetic riboswitches than on bacterial riboswitches, which is significant considering that the initial training focused solely on bacterial riboswitches (Figure R.6A). An example of this capability is the identification of a theophylline riboswitch by SwitchFinder, highlighted in Figure R.6B. These findings support our assertion that SwitchFinder is not narrowly tailored to bacterial riboswitches but is proficient at identifying riboswitches across different categories. Furthermore, it effectively distinguishes non-switching, highly structured RNAs from actual riboswitches, addressing the reviewer's concern about overfitting to a specific riboswitch class.

I would like to see that the authors' algorithm effectively discriminates against other classes of RNAs. It seems to be highly sensitive in detecting potential riboswitches, but is it specific?

Comment 2.4: Line 167: 14% of the predicted switches demonstrated differences from the scrambled control. This is a low accuracy for the algorithm presented. As a comparison, the authors should present an experiment cloning a SwitchFinder predicted "non-RNA-structural

switch" downstream of eGFP and compare it to its scrambled control, would there be a difference in expression between the two? I assume there would be a difference in expression at the thresholds chosen.

Comment 2.5: Line 233: Which features are suggestive of RNA secondary structure? The authors need to translate the 3D classes into structures using the Leontis-Westhoff classification. As far as this reviewer is aware, no study has ever demonstrated that different 3D Cryo-EM classes translate into different RNA secondary structures.

Response to 2.5: Specific base pairing chemistries unfortunately cannot be resolved in the structures presented in this manuscript, which are limited to identification of certain secondary structure features such as helical grooves or hairpins, and discrimination of overall 3D conformation. Groove and hairpin secondary structure features are highlighted in the braided appearance of the 2D classes in Extended Data Figure 5. A-C, and the idealized 3D structures from PDB shown with the 3D class densities in the new Extended Data Figure 4 D-F. The different 3D structures we present do appear to differ in secondary structure content, for example class A appears to represent a helical segment with a central turn leading to a "paperclip" like shape, while class B is consistent with two short helical segments surrounding the central compacted loop elements.

While it seems that there is a conformational change occurring, I am not sure based on the structures presented here because as you state, the models are incomplete.

Comment 2.6: Line 269: The experiment with antisense oligos needs to be presented in comparison to untransfected cells. Antisense oligos activate the RNase H degradation pathway thus I highly doubt that addition of an antisense oligo that targets this region will activate gene expression by altering the conformation of the mRNA.

Response to 2.6: We appreciate the reviewer's comment regarding the use of antisense oligos (ASOs) in our experiments. To address this concern, we would like to clarify that we utilized 2'-O-methylated (MOE) oligonucleotides and locked nucleic acids (LNAs), rather than gapmers. MOEs and LNAs do not activate the RNase H degradation pathway. They function by base pairing with complementary sequences, thereby specifically blocking the function of targeted regulatory elements (Kauppinen, Vester, and Wengel 2005; Kurreck et al. 2002). In response to the reviewer's suggestion, we assessed the impact of ASOs targeting the RORC switch on the stability of endogenous RORC mRNA. Following the introduction of these ASOs, we observed significant shifts in mRNA stability, as detailed in Figure R.7. These effects were more pronounced in cell lines with a higher occurrence of conformation 2 (LNCaP $P=0.006$, MCF-7 $P=0.005$) than in those with a lower occurrence (LS174T $P=0.71$). This finding aligns with our hypothesis that ASOs can shift the equilibrium between RNA conformations, moving it away from the inhibitory conformation 2 towards conformation 1. Thus, our results support the hypothesis that the relative proportions of these RNA conformations are intrinsically linked to mRNA stability in the endogenous setting.

It is not clear in this figure what we are seeing on the y-axis. Did you measure RNA stability with RNAseq? Please clarify this figure and your approach here.

Comment 2.12: Line 150: I am not aware of any data to support the notion that if multiple conformations co-exist, they all contribute the reactivity profile. The DMS modification rate depends on solvent accessibility, protein binding, R-loop formation, reverse transcription read-through, sequence context, and other factors, not on structure only. Many poorly resolved nucleotides in DMS or SHAPE-seq experiments occur genome-wide, they are low probability modification sites which are unable to be accurately called for any number of reasons, not RNA switches.

Response to 2.12: We are grateful for the reviewer's comment and the opportunity it provides to clarify a potential misunderstanding about the SwitchFinder algorithm's functionality.

Firstly, regarding the concept that "if multiple conformations co-exist, they all contribute to the reactivity profile," experimental evidence supporting this has been presented in studies by (Morandi et al. 2021; Tomezsko et al. 2020). In these studies, RNAs were experimentally folded into alternative conformations, mixed, and then analyzed using DMS-seq. The authors successfully deconvolved the reactivity profiles into relative proportions of the alternative conformations, demonstrating that multiple conformations can indeed contribute collectively to the reactivity profile.

This is a profound and important statement of the paper and it is not backed up by the citations provided. The authors posit that the modification rate by DMS is proportional to the RNA strand's conformational profile. The authors state that the Gibbs free energy profile of a given RNA strand is related to its modification rate. For in vivo probing of RNA this is not the case and you need to provide evidence. Even in the citation which you use the authors state: "the rate of DMS modification per open base is sensitive to the local chemical environment, such that not all open bases are equally reactive to DMS." Thus, the DMS modification rate, which you read out in sequencing, is a product of the local environment (multitude of factors) not the conformation or Gibbs free energy of folding. You need to reconcile this difference.

Comment 2.15: Line 242: The authors claim that the low-resolution models presented are sufficient for making predictions of RNA fold and handedness. How is this possible? The authors need to show their structure probing data in a coherent structural model that also corresponds to the 3D states of the same RNA molecule. How can you state that Class B represents one conformation or another?

Response to 2.15: We thank the reviewer for this comment and for the opportunity to clarify this aspect of our study. Our claim about these models hinges on their ability to differentiate various RNA conformations and consistently identify these conformations across different samples. While these models do not offer high-resolution structural details, they are adept at discerning distinct conformational states, which is essential for our analysis. For instance, we identified three classes - A, B, and C - in the wildtype sample, while only classes A and C

were present in the 77-GA sample.

What is meant by classes? Is conformation the same as a class?

Given that our structural probing data suggests the 77-GA mutation shifts the equilibrium away from conformation 2, we hypothesize that class B likely represents this conformation. To directly address the comment, we employed DRAFFTER to fit our structural model to the observed structures, as shown in Figure R.9.

Riboswitches, which the authors are training your data on rely on conformational changes on the secondary structure level, not tertiary/3D level. I do not see any difference in the secondary structures presented in R.9.

Regarding the handedness of the RNA structures, we should clarify that our approach allows us to determine the relative handedness due to the chiral shapes observed in these low-resolution models. However, we acknowledge that determining the absolute handedness would indeed require a higher level of structural detail, specifically the assignment of strand direction, which is beyond the capability of our current modeling approach. In summary, while our models are not high-resolution, they are effective in consistently identifying distinct RNA conformations.

I do not think the authors addressed this comment adequately.

Comment 2.16: Line 257: Two-fold difference in relative expression is very low. Most bacterial riboswitches regulate gene expression on the order of 5-10-fold activation or repression. One cannot rule out here that there is a sequence effect on the UTR.

Response to 2.16: We thank the reviewer for pointing out the concern regarding the fold change in relative expression. In response, we extended our analysis to ensure that the expression changes observed are primarily attributed to changes in the RNA structure, rather than sequence effects.

This did not address my concern. If the authors measured the same expression level data with an artificial riboswitch in mammalian cells, would you see the same <2 fold change?

This involved examining two regions within the RORC element, termed "box 2" and "box 3" (see Fig. 3A), which are predicted to compete for base pairing with "box 1." Applying the DMS-seq assay across five different cell lines with diverse genetic backgrounds (LNCaP, MCF-7, HepG2, ZR-75-1, 293T, and LS174T)...

They are all human cell lines. How diverse could their genetic backgrounds be and why would they have different expression levels of RORC?

we assessed whether an anticorrelation existed between the accessibility of boxes 2 and 3, indicative of competitive base pairing. Our findings confirmed this anticorrelation ($R=-0.81$, Figure R.8A), suggesting that these regions are indeed competing for base pairing. Additionally, we measured the stability of RORC mRNA across these cell lines using RT-qPCR. We observed a very strong correlation ($R=0.96$, $P=0.004$) between the relative proportions of the conformations and mRNA stability (Figure R.8B). This finding provides additional evidence that the balance between the RORC switch conformations is directly linked to the stability of RORC mRNA. Notably, the difference in RORC mRNA stability between LNCaP and LS174T cell lines is greater than 5-fold.

I appreciate that the authors went so far as to perform RT-qPCR but I think the authors need to explain the data a little more. For HepG2 cells, the box2 DMS signal is approximately 0.85 and the box 3 DMS signal is approximately 0.70. Thus the box2:box3 DMS ratio should be approximately 1.2, but in the figure it is displayed as 0.78. For LNCaP cells, box2:box3 DMS ratio should be approximately 0.94, but it is displayed as 0.82. For MCF7 shouldn't it be 1.34? The same is true for all cell lines. I don't think there is any correlation there unless the data is from a different experimental dataset than that which is presented.

Comment 2.17: Line 260: Since no control data presented from a non-switch 3' UTR, one cannot conclude that the two conformations play divergent functional roles.

Response to 2.17: We thank the reviewer for their comment. It is important to clarify why the suggested analysis involving a non-switch 3' UTR is not applicable in our study context. Our research specifically focuses on a mutational analysis of a bi-stable RNA switch within the RORC element. This involves using sequence mutations to shift the equilibrium between two alternative conformations and then assessing the impact of these shifts on reporter expression. A non-switch 3' UTR, by its very nature, does not possess multiple conformations, and thus any mutations introduced would not yield interpretable effects on its secondary structure landscape in the same way as they do for a bi-stable switch. Consequently, comparisons with a non-switch 3' UTR would not provide meaningful insights into the divergent functional roles of the two conformations in the RORC element.

Right, that is why it is a negative control. The authors should present a negative control to make sure that their assay works as they are presenting.

Reviewer #3 (Remarks to the Author):

A. The revised manuscript by Khoroshkin et al. makes many changes requested in the original review. New data are added that provide more confidence in the approach, as well as the identification of the RORC element as a biconformational switch that functions by RNA decay, although the specific effector remain unknown (a topic for future study). A main problem the reviewer had with the revision was the poor quality of some figure presenting the cryo-EM results. In addition, some statements about the cryo-EM technique are incorrect. These can be corrected readily. However, the authors stated that they already made figure corrections to improve the legibility of the manuscript figures.

B. A remaining concern is that the methods are a tour de force approach that are unlikely to be used by the average reader. This will likely limit the impact as a methodological technique paper.

C-F. Major Points

Lines 280-281. The reviewer still believes it is an overinterpretation to state, "These 3D structures demonstrate RNA-like tertiary features, including apparent double-stranded helical segments with a discernible major groove". Without higher resolution or more convincing evidence, this interpretation is not justified. Moreover, it is not really necessary to claim this level of clarity because the point of cryo-EM here is to discern different conformations observed in the samples, and that these are shifted as a result of rational mutations (lines 282-283). The reviewer would be more circumspect by saying, "the rodlike features in the potential maps are consistent with A-form helical features. The maps also appear to show bends and junctions that are consistent with complex RNA folding."

Extended Data Figure 5. Cryo-EM data. The fonts are too small or pixelated to read in these figures. Labels above and below the boxes comprised of dashed lines are too small to read or too pixelated when expanded. The FSC plots are uninterpretable because the keys are too small to see the sample labels. The axes are also too small to read.

G. References

Lines 425-426. The authors states, "In Eukaryotes, mRNA secondary structure is highly dynamic; multiple studies have shown that RNA structure vastly differs when measured in vitro vs in vivo (Rouskin et al. 2014)." If multiple studies have shown this is true, please provide at least one more reference for the reader. The study cited uses DMS modification and the authors reported in Fig. 1c that most of the modified sites are A (68%) and only 24% C. Therefore, the conclusions about dynamics are based on a somewhat incomplete picture of the RNA folding landscape.

Lines 446-447. The reviewer believes this statement is misleading and demonstrably false. The authors wrote, "Additionally, recent advancements in single-particle cryo-EM and computational modeling have enabled the determination of the 3D folds of some RNA molecules (Kappel et al. 2020), despite their small size and intrinsic flexibility." Importantly, x-ray crystallography has been solving the high-resolution structures of RNA molecules since tRNA in the 1970s. The authors should qualify that "recent advances in single-particle cryo-EM and modeling have allowed this technique to be applied to the rapid determination of RNA structures, despite challenges associated with small particle size and flexibility". It is this breakthrough that the authors want to emphasize in the context of their approach. As the authors know, a small amount of sample, no requirement for crystals, and absence

of a phase problem make cryo-EM much better suited for the high throughput approaches shown here. These should be points of emphasis.

H. Corrections/Typos/Clarity

149-150 “these high-throughput screening strategies [compose] the integrated platform SwitchSeeker”.

154-155. This sentence could be clearer. The reviewer suggests, “DMS preferentially modifies unpaired A and C nucleotides resulting in substitutions during reverse transcription (Extended Data Fig. 2A)”.

181. Typo, “included representative candidates with repressive, neutral, or activating function in [Extended Data] Fig. 2B”

Line 230, Supplemental Table S1 is probably Supplemental Table S2. Titles of Tables in the Excel spreadsheet are not clearly labeled.

Line 230, the authors stated, “We then measured the accessibility of individual nucleotides using the in vitro SHAPE assay”. In SHAPE, the acylation is not interpreted as nucleotide accessibility but is correlated with flexibility, resulting in a conformation conducive to nucleophilic attack of the 2'-OH group on the electrophilic modifier. Nucleotides in WC base pairs are less prone to modification. Another interpretation of a highly acylated site is that the structure adopts a stable conformation that is more prone to acylation, which is based on the ribose sugar pucker. [See McGinnis, J.L., Dunkle, J.A., Cate, J.H. and Weeks, K.M. (2012) The mechanisms of RNA SHAPE chemistry. *J. Am. Chem. Soc.*, 134, 6617–6624]. In the ensuing text, I would replace “accessibility” with “reactivity” or possibly “flexibility”. Figure 3c correctly refers to acylation as SHAPE reactivity. Hydroxy-radical footprinting is better correlated with (solvent) accessibility.

Supplemental Figure 3C. Are these plots reactivity for a specific nucleotide? If so, please label each axis as nucleotide reactivity, replicate x.

Supplemental Figure 3D. Are these Biological replicates or technical replicates? What is the definition of a cluster? Is this a specific conformation from DRACO analysis? This could be explained better; e.g., please add some clarification to the figure legend for panel C.

Line 392. The authors describe “regions 2 and 3” of RORC mRNA. This should be Box 2 or Box 3 for consistency.

Reviewer #3 (Remarks on code availability):

I did not try to install and run the code but I did check its availability. The code does seem to have instructions to run it.

Author Rebuttal, first revision:

Reviewer #1:

Remarks to the Author:

The authors have provided a revised manuscript version, in which they have appropriately addressed all of my previous major concerns. I have noted the following minor points:

We thank the reviewer for their constructive comments, and we have addressed the few remaining comments below.

Reviewer Comment 1.1. A more accurate description of the statistical tests is needed in several cases. For example, for Fig. 5F, 6E, H-I, one p-value is provided in charts with multiple bars without defining the comparisons. What kind of statistical test has been performed in case of Fig. 6E and Extended Data Fig. 7E? Are the data really based on two replicates each and p-values are $< 2e-16$?

Response 1.1. We thank the reviewer for this comment. We have revised the descriptions of statistical tests. In Fig. 5F, P-values were determined using the independent *t*-test, comparing the RORC-targeting and control ASOs, independent of the ASO chemistry. In Fig. 6E and in Extended Data Fig. 7E, P-value was calculated using a ratio-of-ratios test as we described in ¹. The statistical significance is determined not only by the number of replicates but also by the number of unique reads in each sample and condition (after UMI-based deduplication). For Fig. 6H-I, statistical significance was assessed using dose-response modeling with ANOVA for model comparison, specifically focusing on differences in drug potency (ED50 values) between cell lines expressing different variants of the RNA switch.

Reviewer Comment 1.2. Statement in l. 235-236 "Even though we did not observe a significant decrease in accessibility of the Box 3 upon the 77-GA mutation...". As no statistical analysis is provided, the word "significant" is misleading in this context.

Response 1.2. We thank the reviewer for this comment. We have changed the phrasing of this sentence to avoid misleading phrasing.

Reviewer Comment 1.3. Legend to Fig. 2B: shouldn't there be a definition of bin 1 to bin 8 for fluorescence, instead of bin 1 to bin 4?

Response 1.3. We thank the reviewer for this comment. We have corrected this in the manuscript.

Reviewer Comment 1.4. Fig. 2C, legend says that candidate switches were mutated to lock them in either of the two conformations, and that "a sequence library is then generated (see Extended Data Fig. 2A)". Extended Data Fig. 2A refers to the structure screen by DMS-MaPseq. Does that mean that the conformation of the mutated switch sequences was tested by DMS-MaPseq or is the reference not correct?

Response 1.4. We thank the reviewer for this comment. The figure referenced was indeed incorrect, and has now been corrected.

Reviewer Comment 1.5. Fig. 3B: the symbols for matching/non-matching base pairs are hard to distinguish, at least at this size.

Response 1.5. We thank the reviewer for this comment. We have increased the size of Fig. 3B.

Reviewer Comment 1.6. Fig. 5C: the "+" symbol for the box2 mutant is incorrectly placed

Response 1.6. We thank the reviewer for this comment. We have corrected this.

Reviewer Comment 1.7. Extended data figure 6: some further information or conclusion what these markers stand for would be helpful for readers who are not experts in Th17 cell differentiation.

Response 1.7. We thank the reviewer for this comment; we have included additional information on the markers shown.

Reviewer #2:

Remarks to the Author:

See report attached

Below I reiterate and clarify some of my points that have not been adequately addressed

Reviewer Comment 2.1. I think the authors are indeed actually attempting this with the theophylline riboswitch. Surely, there are others such as the GuaM8 aptazyme. Below, the authors even state that they "include a comprehensive set of theophylline riboswitches (Wang et al. 2023) and a collection of artificial protein-sensing riboswitches (Vezeau, Gadila, and Salis 2023)." If the authors cannot benchmark your algorithm against known riboswitches, why not use a synthetic set as a control?

Response 2.1. We thank the reviewer for this comment and for the opportunity to clarify the nature of our platform. SwitchSeeker is composed of "SwitchFinder", which computationally scores RNA switches, and secondary structure and functional screens (as we have clearly shown in our Fig. 2A). The RNA switches cited by the reviewer are indeed switches, but they are not necessarily divergently functional in gene expression, i.e. there is no guarantee that inserting them into the 3'UTR of a reporter would impact expression in a conformation dependent manner. This simply has not been tested for any of the switches listed by the reviewer. Therefore, these suggested switches do not comprise a ground truth for our platform,

and cannot be used to benchmark our approach. In the previous round we sought to be responsive to this reviewer by performing some benchmarks, but only for the SwitchFinder, which as stated above does not constitute the entirety of our platform. The search for functionally divergent switches, not just any switch, is at the core of our study.

Reviewer Comment 2.2. If that is the case, then why not test your approach, which includes structural characterizations like DMS MapSeq and Cryo-EM, on at least one known artificial switch.

Response 2.2. We thank the reviewer for this comment. The aforementioned structural characterization methods, DMS-MapSeq and CryoEM, they are both previously well-established and well-characterized by us and others²⁻⁴ and therefore do not require further characterization.

Reviewer Comment 2.3. While I like the SwitchFinder approach, I do not think that the difference between an auROC of 0.54 versus 0.63 is significant.

Response 2.3. We are grateful for the reviewer's engagement with our approach's statistical metrics. While the improvement from an auROC of 0.54 to 0.63 might seem modest, it is a meaningful advancement within the context of our platform. This computational step is intended as an "enrichment" step for RNA switches, which are then systematically evaluated both in terms of their structure and conformation-dependent function, as shown in Fig. 2A. The increase in sensitivity between ribosomal RNAs and bacterial riboswitches exceeds 17%, which is valuable when applying the method to the vast human transcriptome.

Reviewer Comment 2.4. Why didn't the authors include a Theophylline switch or any control synthetic switch in the initial screen? The authors need to validate that the assay works. I do not think that RNAs in cells fold into meta-stable structures, that is why I have a hard time grasping the approach. The authors need to prove that meta-stable structures exist and function as switches. They compare their approach against bacterial riboswitches, which are NOT meta-stable structures, they are alternative stable conformations of the same RNA sequence.

Response 2.4. We thank the reviewer for this comment. Our platform does not require the switches to be meta-stable, just that the two conformations are divergently functional. Nowhere in the text have we made the claim that we are identifying meta-stable structures. Also, the reviewer states that "The authors need to validate that the assay works", yet, we are not clear which assay they are referring to. If they refer to DMS-MapSeq and MPRA, they are both previously well-established and well-characterized by us and others (as cited above). If instead they mean the entire framework they are referring to, figures 3-7 are entirely dedicated to this very validation.

Reviewer Comment 2.5. tRNAs are NOT known to switch between secondary structure conformations like RNAs that you are trying to discover in the paper. The paper you cite here demonstrates structural flexibility in tertiary structure on the order of angstroms, between two

different crystal structures of the same tRNA. You need to give a better reason for why you can't benchmark SwitchFinder against other classes of RNAs.

Response 2.5. We thank the reviewer for this comment and for the opportunity to clarify the purpose of SwitchFinder. SwitchFinder is not meant to be used as a standalone tool, but rather as a first step in a multi-step SwitchSeeker pipeline. As we have sought to clarify above, it identifies the RNA sequences that have the potential to act as RNA switches - i.e. the sequences that could form two alternative mutually exclusive conformations. In the previous rebuttal, we have provided data benchmarking the SwitchFinder against the class of ribosomal RNAs (Fig. 1D). This comparison demonstrated that SwitchFinder distinguishes RNA switches from non-switching but highly structured RNAs. Transfer RNAs, on the other hand, do demonstrate high degree of structural plasticity. They can adopt a variety of structural conformations in vitro, depending on the salt concentration and other conditions. Maglott et al write: "*Conformation-phase diagrams constructed from these measurements consist of four regions: native, cloverleaf or close variants, coil, and extended forms. With the exception of the true cloverleaf secondary structure, direct evidence that bacterial tRNAs can assume all conformational states defined by Crothers has been obtained*".⁵

Even though tRNAs do not switch the same way as the RNA switches that we are trying to identify, their sequence has a propensity to form alternative conformations, and, therefore, we expect SwitchFinder to highlight their sequences. To illustrate this point, we demonstrate the base pairing probability of human tRNA-His (RNACentral ID URS0000052CEF) as predicted by RNAfold software⁶. Alternative folds, highlighted in red, demonstrate that the sequence of this tRNA has a potential of forming alternative conformations, and will, therefore, be identified by SwitchFinder.

Figure R.1: Base pairing probability of human tRNA-His, as predicted by RNAfold software. The potentially conflicting stem-loops are highlighted in red.

Reviewer Comment 2.6. While I like the SwitchFinder approach, I do not think that the difference between an auROC of 0.54 versus 0.63 is significant.

Response 2.6. We thank the reviewer for this comment. See the reply above.

Reviewer Comment 2.7. I would like to see that the authors' algorithm effectively discriminates against other classes of RNAs. It seems to be highly sensitive in detecting potential riboswitches, but is it specific?

Response 2.7. We thank the reviewer for this comment. To demonstrate the specificity of the algorithm, we have benchmarked it against ribosomal RNAs. As shown in Fig. 1D, SwitchFinder performs significantly worse on non-switching ribosomal RNAs compared to bacterial, eukaryotic, and artificial riboswitches. Acknowledging that no algorithm will be sufficiently specific to identify RNA switches in large transcriptomes such as the human genome, we have extended the SwitchSeeker method to include multiple experimental screening steps that increase the method's overall specificity.

Reviewer Comment 2.8. While it seems that there is a conformational change occurring, I am not sure based on the structures presented here because as you state, the models are incomplete.

Response 2.8. We thank the reviewer for this comment. We do not make the conclusion about the conformational change guided by the CryoEM data alone. Rather, we make this conclusion based on the combination of *in vitro* and *in vivo* RNA chemical probing data, complemented with CryoEM data for both reference and mutated sequences. Taken together, the data consistently indicates the presence of the two conformations we have identified.

Reviewer Comment 2.9. It is not clear in this figure what we are seeing on the y-axis. Did you measure RNA stability with RNAseq? Please clarify this figure and your approach here.

Response 2.9. We thank the reviewer for the opportunity to clarify the presented data. The relative RORC expression was measured by RT-qPCR. We have clarified the approach in the revised manuscript (Fig. 5F).

Reviewer Comment 2.10. This is a profound and important statement of the paper and it is not backed up by the citations provided. The authors posit that the modification rate by DMS is proportional to the RNA strand's conformational profile. The authors state that the Gibbs free energy profile of a given RNA strand is related to its modification rate. For *in vivo* probing of RNA this is not the case and you need to provide evidence. Even in the citation which you use the authors state: "the rate of DMS modification per open base is sensitive to the local chemical environment, such that not all open bases are equally reactive to DMS." Thus, the DMS modification rate, which you read out in sequencing, is a product of the local environment

(multitude of factors) not the conformation or Gibbs free energy of folding. You need to reconcile this difference.

Response 2.10. We thank the reviewer for this comment. We agree that the DMS modification rate is influenced by a variety of factors, including local chemical environment. We do not state that DMS modification rate is directly proportional to the RNA conformational profile. It is important to emphasize that we are solely relying on previously established methods for RNA conformational profiling deconvolution from DMS-MaPseq data, namely DRACO and DREEM, and we are not making any new claims on this front. While DMS-MaPseq reactivity is influenced by a variety of factors, the works by Tomeszco and Morandi^{2,3} demonstrated that the ratio of conformations in the RNA folding ensemble can be inferred quite accurately from DMS-MaPseq data: for example, Morandi et al state that "*DRACO correctly identified the expected number of conformations in nearly 100% of cases, accurately deconvoluted the individual conformation mutational profiles and precisely estimated relative conformation stoichiometries*". We are, therefore, relying on these techniques for DMS-MaPseq data analysis.

Reviewer Comment 2.11. What is meant by classes? Is conformation the same as a class?

Response 2.11. We thank the reviewer for this comment. They are not quite the same, as molecular "conformation" is a general term for the concept of a particular structural microstate, whereas 3D classes are essentially Gaussian mixture model centroids that may or may not have a one-to-one relationship with the true conformational landscape of the sample molecules. For example, multiple classes may represent the same conformation, or a certain class might be dominated by crystalline ice contamination and thus carry no information about the intended sample at all. The process of selecting the classes presented in main Figure 4, that most likely represent unique conformations of the sample molecules, has been loosely standardized in the field and is diagrammed in Extended Data Figure 5.

When multiple conformations are identified in the same sample through orthogonal structural or biochemical/biophysical methods such as cryo-EM and DMS-seq, it becomes necessary to make assignments between them in order to provide a consistent interpretation of all the data. Unfortunately, very high resolution cryo-EM structures that might have allowed comparison of residue-level interactions between the methods were not forthcoming. Instead, we relied on mutational analysis, which is robust even with limited structural detail because the 3D classes have coarse (low resolution) differences.

Reviewer Comment 2.12. Riboswitches, which the authors are training your data on rely on conformational changes on the secondary structure level, not tertiary/3D level. I do not see any difference in the secondary structures presented in R.9.

Response 2.12. We thank the reviewer for this comment. The comment is somewhat unclear, as the "primary/secondary/tertiary/quaternary" terminology for macromolecular structure is intrinsically hierarchical. A change in the secondary structure of a macromolecule, for example the helical state or base pairing of particular residues, necessarily implies a difference of the

atomic positions - which is the definition of tertiary structure. The figures, now in the manuscript as Extended Data Figure G-H, show drastically different structural models with few base pairs in common outside the small segment used to initialize the search.

Reviewer Comment 2.13. I do not think the authors addressed this comment adequately: Line 242: The authors claim that the low-resolution models presented are sufficient for making predictions of RNA fold and handedness. How is this possible? The authors need to show their structure probing data in a coherent structural model that also corresponds to the 3D states of the same RNA molecule. How can you state that Class B represents one confirmation or another?

Response 2.13. Empirical macromolecular structures are not predictions, but are representations of the actual state of the sample molecules when the experiment was conducted (that is, in the past). The structures of the cryo-EM classes are clearly differentiable - class A may be described as resembling a hairpin or paperclip, class B as a letter "T" with elongated arms, and class C as a letter "L." These folds, or overall structures, are robustly evinced by both reference-free and reference-based image processing methods, and may be plainly recognized across wild-type and mutant samples. Relative hand may be assessed because the structures have chiral features visible even at low resolution, for example the position and angle of the junction between the long and short segments of the "T-like" class B structure, or the somewhat finer tip of the apparent hairpin in class C (as shown in Extended Data 4F, left panel).

Our assignment of the cryo-EM classes to conformations identified by orthogonal methods was made using mutational analysis. For example, class B is not observed in cryo-EM images of the 77-GA mutant, and can thus be assigned as the conformation eliminated from this mutant in the (orthogonal) DRACO analysis.

Reviewer Comment 2.14. This did not address my concern. If the authors measured the same expression level data with an artificial riboswitch in mammalian cells, would you see the same <2 fold change?

Response 2.14. We thank the reviewer for this comment. Unfortunately, the observations of bacterial riboswitches do not translate directly to human RNA switches, as they likely act through very different mechanisms. The handful of known examples of human RNA switches, such as the VEGFA switch, lack data on the expression fold change upon their switching. Similarly, the artificial riboswitches likely do not represent the nature of natural human RNA switches. This very lack of prior knowledge motivated us to perform deep characterization of the RORC RNA switch.

Reviewer Comment 2.15. They are all human cell lines. How diverse could their genetic backgrounds be and why would they have different expression levels of RORC?

Response 2.15. We thank the reviewer for this comment. It is important to recognize that even within human cell lines, genetic backgrounds can vary significantly due to their derivation from different individuals, tissues, or conditions, each with unique genetic mutations and epigenetic modifications. This diversity is well documented in large-scale studies such as those cataloged by the Human Cell Atlas and GTEx^{7,8}, where single-cell and bulk RNA sequencing data has shown a wide range of gene expression profiles across various cell lines. As detailed in Figure R2, which presents data from the GTEx project, there is considerable variability in RORC expression across different tissues and cell lines. This variability can be attributed to tissue-specific regulatory networks, differential activation of signaling pathways, and variability in transcription factor binding sites.

Figure R.2: RORC expression across human tissues as reported by GTEx.

Reviewer Comment 2.16. I appreciate that the authors went so far as to perform RT-qPCR but I think the authors need to explain the data a little more. For HepG2 cells, the box2 DMS signal is approximately 0.85 and the box 3 DMS signal is approximately 0.70. Thus the box2:box3 DMS ratio should be approximately 1.2, but in the figure it is displayed as 0.78. For LNCaP cells, box2:box3 DMS ratio should be approximately 0.94, but it is displayed as 0.82. For MCF7 shouldn't it be 1.34? The same is true for all cell lines. I don't think there is any correlation there unless the data is from a different experimental dataset than that which is presented.

Response 2.16. We thank the reviewer for this comment. The Figures 5D and 5E show the relationship of relative RNA stability to the box2:box3 DMS signal ratio of the reporter across the set of cell lines. We defined the state of the RORC switch in each background based on the box2:box3 DMS signal ratio calculated from DMS-MaPseq performed on the reporter. Therefore, this ratio was plotted on X axis of both panels. Given that the box2:box3 DMS signal ratios correlate almost perfectly between reporter-expressing cells and endogenous measurements (Suppl. Fig. 3G, $R=0.93$), we use these measurements interchangeably. The endogenous RORC RNA stability correlates with endogenous DMS signal ratios at $R=0.8$, as

shown in Fig. R3. We have clarified the captions to Fig. 5D,E to avoid any future misunderstanding.

Figure R.3: Scatterplot showing the relationship between the box2:box3 ratio of RORC element, as measured by DMS-MaPseq on endogenous RORC mRNA and stability of the endogenous RORC mRNA, as measured by RT-qPCR.

Reviewer Comment 2.17. Right, that is why it is a negative control. The authors should present a negative control to make sure that their assay works as they are presenting.

Response 2.17. We value the reviewer's perspective on the use of a negative control. However, in the context of our study, the purpose of introducing mutations into the RORC 3' UTR is specifically to perturb the equilibrium between its two naturally occurring conformations and to observe the corresponding functional outcomes. Since a non-switch 3' UTR inherently lacks such conformational variability, mutations would not alter its structure-function relationship in a comparable manner. Thus, it wouldn't serve as an effective negative control for our particular assay, which is designed to measure the functional impact of conformational changes, not the baseline activity of a static RNA element. Introducing a non-switch 3' UTR as a control would not validate the assay's ability to detect differences in gene expression due to structural shifts, as there would be no such shifts to observe. Our findings specifically highlight the functional differences arising from the dynamic nature of the bi-stable switch, a feature absent in non-switch elements.

Reviewer #3:

Remarks to the Author:

Reviewer Comment 3.1. A. The revised manuscript by Khoroshkin et al. makes many changes requested in the original review. New data are added that provide more confidence in the approach, as well as the identification of the RORC element as a biconformational switch that functions by RNA decay, although the specific effector remain unknown (a topic for future study). A main problem the reviewer had with the revision was the poor quality of some figure

presenting the cryo-EM results. In addition, some statements about the cryo-EM technique are incorrect. These can be corrected readily. However, the authors stated that they already made figure corrections to improve the legibility of the manuscript figures.

Response 3.1. We thank the reviewer for their commitment in helping prepare the manuscript for a diverse readership. We hope that additional changes in the new revision allay their remaining concerns.

Reviewer Comment 3.2. B. A remaining concern is that the methods are a tour de force approach that are unlikely to be used by the average reader. This will likely limit the impact as a methodological technique paper.

Response 3.2. We thank the reviewer for this comment. We have simplified the methods section and included a detailed Supplementary Protocol to facilitate easier application of our techniques. Our team is committed to further simplifying the process and enhancing user-friendliness in subsequent updates.

Reviewer Comment 3.3. Lines 280-281. The reviewer still believes it is an overinterpretation to state, "These 3D structures demonstrate RNA-like tertiary features, including apparent double-stranded helical segments with a discernible major groove". Without higher resolution or more convincing evidence, this interpretation is not justified. Moreover, it is not really necessary to claim this level of clarity because the point of cryo-EM here is to discern different conformations observed in the samples, and that these are shifted as a result of rational mutations (lines 282-283). The reviewer would be more circumspect by saying, "the rodlike features in the potential maps are consistent with A-form helical features. The maps also appear to show bends and junctions that are consistent with complex RNA folding."

Response 3.3. We thank the reviewer for their continued engagement here and we have incorporated the suggestion in the main text.

Reviewer Comment 3.4. Extended Data Figure 5. Cryo-EM data. The fonts are too small or pixelated to read in these figures. Labels above and below the boxes comprised of dashed lines are too small to read or too pixelated when expanded. The FSC plots are uninterpretable because the keys are too small to see the sample labels. The axes are also too small to read.

Response 3.4. We apologize that the copy received by the reviewer was of reduced quality. In the original, these are all either vector graphics or high DPI embedded bitmaps that are clear on our screens even when expanded. We will communicate with the editor to ensure that sufficient resolution is maintained in the revised version.

Reviewer Comment 3.5. G. References

Lines 425-426. The authors states, "In Eukaryotes, mRNA secondary structure is highly dynamic; multiple studies have shown that RNA structure vastly differs when measured in vitro vs in vivo (Rouskin et al. 2014)." If multiple studies have shown this is true, please provide at

least one more reference for the reader. The study cited uses DMS modification and the authors reported in Fig. 1c that most of the modified sites are A (68%) and only 24% C. Therefore, the conclusions about dynamics are based on a somewhat incomplete picture of the RNA folding landscape.

Response 3.5. We thank the reviewer for this comment. We have revised the phrasing and have updated the references to improve the clarity.

Reviewer Comment 3.6. Lines 446-447. The reviewer believes this statement is misleading and demonstrably false. The authors wrote, "Additionally, recent advancements in single-particle cryo-EM and computational modeling have enabled the determination of the 3D folds of some RNA molecules (Kappel et al. 2020), despite their small size and intrinsic flexibility." Importantly, x-ray crystallography has been solving the high-resolution structures of RNA molecules since tRNA in the 1970s. The authors should qualify that "recent advances in single-particle cryo-EM and modeling have allowed this technique to be applied to the rapid determination of RNA structures, despite challenges associated with small particle size and flexibility". It is this breakthrough that the authors want to emphasize in the context of their approach. As the authors know, a small amount of sample, no requirement for crystals, and absence of a phase problem make cryo-EM much better suited for the high throughput approaches shown here. These should be points of emphasis.

Response 3.6. The intended meaning was: "Additionally, recent advancements in single-particle cryo-EM and computational modeling have enabled the determination of the 3D folds of some RNA molecules **[by cryo-EM]** (Kappel et al. 2020), despite their small size and intrinsic flexibility."

We thank the reviewer for the comment, and the sentence in question is no longer in the revised manuscript. In concordance with the reviewer's perspective we have emphasized that advancements in secondary structure probing and cryo-EM were instrumental in our systematic (i.e. higher throughput) investigations of RNA function.

Reviewer Comment 3.7. 149-150 "these high-throughput screening strategies [compose] the integrated platform SwitchSeeker".

Response 3.7. We thank the reviewer for this comment; we have incorporated the suggestion in the revised text.

Reviewer Comment 3.8. 154-155. This sentence could be clearer. The reviewer suggests, "DMS preferentially modifies unpaired A and C nucleotides resulting in substitutions during reverse transcription (Extended Data Fig. 2A)".

Response 3.8. We thank the reviewer for this comment; we have incorporated the suggestion in the revised text.

Reviewer Comment 3.9. 181. Typo, “included representative candidates with repressive, neutral, or activating function in [Extended Data] Fig. 2B”

Response 3.9. We thank the reviewer for this comment. We are indeed referring to the representative candidates shown in Fig. 2B.

Reviewer Comment 3.10. Line 230, Supplemental Table S1 is probably Supplemental Table S2. Titles of Tables in the Excel spreadsheet are not clearly labeled.

Response 3.10. We thank the reviewer for this comment. We have incorporated the Titles of Tables into the Excel spreadsheet.

Reviewer Comment 3.11. Line 230, the authors stated, “We then measured the accessibility of individual nucleotides using the in vitro SHAPE assay”. In SHAPE, the acylation is not interpreted as nucleotide accessibility but is correlated with flexibility, resulting in a conformation conducive to nucleophilic attack of the 2'-OH group on the electrophilic modifier. Nucleotides in WC base pairs are less prone to modification. Another interpretation of a highly acylated site is that the structure adopts a stable conformation that is more prone to acylation, which is based on the ribose sugar pucker. [See McGinnis, J.L., Dunkle, J.A., Cate, J.H. and Weeks, K.M. (2012) The mechanisms of RNA SHAPE chemistry. *J. Am. Chem. Soc.*, 134, 6617–6624]. In the ensuing text, I would replace “accessibility” with “reactivity” or possibly “flexibility”. Figure 3c correctly refers to acylation as SHAPE reactivity. Hydroxy-radical footprinting is better correlated with (solvent) accessibility.

Response 3.11. We thank the reviewer for this comment; we have incorporated the suggestion in the revised text.

Reviewer Comment 3.12. Supplemental Figure 3C. Are these plots reactivity for a specific nucleotide? If so, please label each axis as nucleotide reactivity, replicate x.

Response 3.12. We thank the reviewer for this comment; we have updated the axis labels to clearly indicate that nucleotide reactivity is shown.

Reviewer Comment 3.13. Supplemental Figure 3D. Are these Biological replicates or technical replicates? What is the definition of a cluster? Is this a specific conformation from DRACO analysis? This could be explained better; e.g., please add some clarification to the figure legend for panel C.

Response 3.13. We thank the reviewer for this comment; we have clarified the figure legend for the Extended Data Fig. 3D.

Reviewer Comment 3.14. Line 392. The authors describe “regions 2 and 3” of RORC mRNA. This should be Box 2 or Box 3 for consistency.

Response 3.14. We thank the reviewer for this comment; we have incorporated the suggestion in the revised text.

1. Navickas, A. *et al.* An mRNA processing pathway suppresses metastasis by governing translational control from the nucleus. *Nat. Cell Biol.* **25**, 892–903 (2023).
2. Morandi, E. *et al.* Genome-scale deconvolution of RNA structure ensembles. *Nat. Methods* **18**, 249–252 (2021).
3. Tomezsko, P. J. *et al.* Determination of RNA structural diversity and its role in HIV-1 RNA splicing. *Nature* **582**, 438–442 (2020).
4. Kappel, K. *et al.* Accelerated cryo-EM-guided determination of three-dimensional RNA-only structures. *Nat. Methods* **17**, 699–707 (2020).
5. Maglott, E. J., Deo, S. S., Przykorska, A. & Glick, G. D. Conformational transitions of an unmodified tRNA: implications for RNA folding. *Biochemistry* **37**, 16349–16359 (1998).
6. Lorenz, R. *et al.* ViennaRNA Package 2.0. *Algorithms Mol. Biol.* **6**, 26 (2011).
7. Regev, A. *et al.* The Human Cell Atlas. *Elife* **6**, (2017).
8. GTEx Consortium. The Genotype-Tissue Expression (GTEx) project. *Nat. Genet.* **45**, 580–585 (2013).

Final Decision Letter:

Dear Hani,

I am pleased to inform you that your Article, "A systematic search for RNA structural switches across the human transcriptome", has now been accepted for publication in Nature Methods. The received and accepted dates will be Feb 26, 2023 and May 29, 2024. This note is intended to let you know what to expect from us over the next month or so, and to let you know where to address any further questions.

Over the next few weeks, your paper will be copyedited to ensure that it conforms to Nature Methods style. Once your paper is typeset, you will receive an email with a link to choose the appropriate publishing options for your paper and our Author Services team will be in touch regarding any additional information that may be required. It is extremely important that you let us know now whether you will be difficult to contact over the next month. If this is the case, we ask that you send us the contact information (email, phone and fax) of someone who will be able to check the proofs and deal with any last-minute problems.

Please note that *Nature Methods* is a Transformative Journal (TJ). Authors may publish their research with us through the traditional subscription access route or make their paper immediately open access through payment of an article-processing charge (APC). Authors will not be required to make a final decision about access to their article until it has been accepted. Find out more about Transformative Journals

You may wish to make your media relations office aware of your accepted publication, in case they consider it appropriate to organize some internal or external publicity. Once your paper has been scheduled you will receive an email confirming the publication details. This is normally 3-4 working days in advance of publication. If you need additional notice of the date and time of publication,

please let the production team know when you receive the proof of your article to ensure there is sufficient time to coordinate. Further information on our embargo policies can be found here: <https://www.nature.com/authors/policies/embargo.html>

If you are active on Twitter/X, please e-mail me your and your coauthors' handles so that we may tag you when the paper is published.

Best regards,
Rita

Rita Strack, Ph.D.
Senior Editor
Nature Methods